

# Explaining trends and changing seasonal cycles of surface ozone in North America and Europe over the 2000-2018 period: A global modelling study with NO$_X$ and VOC tagging

Tabish Ansari[1*], Aditya Nalam[1,2], Aurelia Lupaşcu[3], Carsten Hinz[4], Simon Grasse[4], and Tim Butler[1,2]

[1]Research Institute for Sustainability - Helmholtz Centre Potsdam (RIFS), Potsdam, 14467, Germany
[2]Institut für Meteorologie, Freie Universität Berlin, Berlin, Germany
[3]European Center for Medium Range Weather Forecasts (ECMWF), Bonn, 53175, Germany
[4]Jülich Supercomputing Centre (JSC), Forschungszentrum Jülich GmbH, 52428 Jülich, Germany

*Correspondence to*: Tabish Ansari (tabish.ansari@rifs-potsdam.de)

**Abstract.** Surface ozone, with its long enough lifetime, can travel far from its precursor emissions, affecting human health, vegetation, and ecosystems on an intercontinental scale. Recent decades have seen significant shifts in ozone precursor emissions: reductions in North America and Europe, increases in Asia, and a steady global rise in methane. Observations from North America and Europe show declining ozone trends, a flattened seasonal cycle, a shift in peak ozone from summer to spring, and increasing wintertime levels. To explain these changes, we use TOAST 1.0, a novel ozone tagging technique implemented in the global atmospheric model CAM4-Chem which attributes ozone to its precursor emissions fully by NO$_X$ or VOC+CO+CH4 sources and perform multi-decadal model simulations for 2000-2018. Model-simulated maximum daily 8h ozone (MDA8 O$_3$) agrees well with rural observations from the TOAR-II database. Our analysis reveals that declining local NO$_X$ contributions to peak-season ozone (PSO) in North America and Europe are offset by rising contributions from natural NO$_X$ (due to increased productivity), and foreign anthropogenic- and international shipping NO$_X$ due to increased emissions. Transported ozone dominates during spring. Methane is the largest VOC contributor to PSO, while natural NMVOCs become more important in summer. Contributions from anthropogenic NMVOCs remain smaller than those from anthropogenic NO$_X$. Despite rising global methane levels, its contribution to PSO in North America and Europe has declined due to reductions in local NO$_X$ emissions.

# 1 Introduction

Ozone near the Earth's surface is primarily formed by the photodissociation of NO$_2$ molecules by sunlight - the NO$_2$ molecule breaks down and furnishes atomic oxygen which combines with molecular oxygen in the air to form ozone. The naturally occurring NO$_2$ concentration in the troposphere is low and cannot alone explain the high ozone observed in the troposphere (Jacobson, 2005; Seinfeld & Pandis, 2016). However, in the modern era especially towards the end of the 20th century, increased industrialization and motorization of society has led to increasing emissions of nitric oxide (NO) (Logan 1983;



Beaton et al., 1991; Calvert et al., 1993). NO can interact with peroxy radicals, chiefly produced from naturally and anthropogenically emitted non-methane volatile organic compounds (NMVOCs), carbon mo$NO_X$ide (CO), and methane ($CH_4$)

in the presence of the hydroxyl radical (OH) to form $NO_2$ which can then produce ozone through the pathway described above (Atkinson 1990, 1994, 1997; Seinfeld & Pandis, 2016). Unsurprisingly, with increasing anthropogenic activities emitting NO, CO, NMVOCs and $CH_4$, the ozone concentrations in the troposphere and at the surface have risen substantially as compared to the pre-industrial or early-industrial times (Logan 1985; Crutzen 1988; Young et al., 2013).

Ozone is a highly reactive pollutant that harms human health, vegetation, and the environment due to its oxidative properties. In humans, it causes respiratory inflammation, exacerbates chronic illnesses, and impairs lung function by generating reactive oxygen species that damage cellular structures (Lippmann 1989; Chen et al., 2007; Devlin et al., 1991; Brook et al., 2004). Ozone disrupts photosynthesis in plants and damages tissues, reducing crop yields and altering ecosystems (Ashmore 2005; Felzer et al., 2007; Grulke & Heath 2019). Moreover, it contributes to climate change by diminishing the carbon sequestration

ability of vegetation and acting as a greenhouse gas (Oeschger & Dutsch 1989; Sitch et al, 2007). In light of these harmful effects, the World Health Organization (WHO) has set safe standards for short-term and long-term human exposure to ozone: on any day, the maximum 8h average ozone concentration (MDA8 $O_3$) which must not exceed 100 $\mu g m^{-3}$ (or ~51 ppb), and annually, the Peak Season Ozone (PSO), i.e., the maximum value of the six-month running average of MDA8 $O_3$, must not exceed 60 $\mu g m^{-3}$ (or ~30.61 ppb) (WHO 2021).

In order to meet these safe health standards, various national governments - particularly in North America and Europe and more recently in China - have acted to reduce their industrial and vehicular emissions by adopting cleaner fuel and technologies and have successfully managed to bring down their national $NO_X$ and NMVOC emissions substantially (Goldberg et al., 2021; Shaw & Heyst 2022; Crippa et al., 2023). However, these national efforts of emission reductions have not fully translated into

commensurate reductions in local ozone concentrations and health impacts (Seltzer et al., 2020; Parrish et al., 2022). This is due to the long-enough atmospheric lifetime of ozone (about 3-4 weeks) which allows it to traverse intercontinental distances and affect the air quality of regions far from the location of its chemical production or the location of the emission of its precursors. Therefore, air quality benefits in regions with declining emissions can be offset by an increasing share of transported ozone from far away regions where emissions are on the rise. Many previous observational-based studies have

reported declining peak-ozone trends in North America towards the final decades of the 20th century and the beginning of the 21st century (Wolffe et al., 2001; Cooper et al., 2014; Cooper et al., 2015; Chang et al., 2017; Fleming et al., 2018; Cooper et al., 2020). However, some of these studies and many others - through novel statistical filtering of observational data - have also pointed out increasing trends in wintertime and background ozone concentrations at many sites in North America, particularly at the US west coast (Jaffe et al., 2003; Cooper et al., 2010; Simon et al., 2014; Parris & Ennis, 2019; Parrish et

al., 2022). Some of these observational studies (e.g., Jaffe et al., 2003) have further correlated the increasing background ozone in western US to increasing emissions in Asia while others (e.g., Cooper et al., 2010) have also employed air mass back



trajectory analysis to support their claims. A number of observational studies have also reported changes in the ozone seasonal cycle in North America, with shifting peaks from summer to springtime (Bloomer et al., 2010; Parrish et al., 2013; Cooper et al., 2014), a reversal of the spring-to-summer shift in peak ozone during mid-twentieth century which was reported in earlier

studies (e.g., Logan 1985) when anthropogenic emissions were increasing in North America. Similarly, for Europe, many studies have observed declining ozone trends since 2000 (Cooper et al., 2014; Chang et al., 2017; Fleming et al., 2018; EEA report 2020; Sicard 2021). For Europe too, there have been attempts of statistical filtering and analyses of observational data in innovative ways to highlight the increasing share of intercontinental transport and the consequent changes in ozone seasonal cycle in recent decades (Carslaw 2005; Parrish et al., 2013; Derwent & Parrish, 2022).


Reliable, long-term, and publicly accessible monitoring stations across different continents form the backbone of an international consensus on ozone distributions, trends, and health impacts on various populations. These observational networks provide essential data for advanced statistical analyses, which can estimate both transported and locally produced ozone (as seen in many observational studies mentioned earlier). However, such statistical interpretations can be subject to

dispute and must be corroborated by well-evaluated atmospheric chemical transport models which simulate atmospheric transport processes explicitly. The hemispheric-scale transport of "foreign" ozone is a phenomenon peculiar to longer-lived pollutants such as ozone. While short-lived pollutants like PM2.5, which are regional in nature, can be largely controlled through domestic policies, effective ozone mitigation requires international engagement and cooperation. Developing such cooperation requires a high-trust international dialogue, underpinned by confident estimates of ozone transport between

regions on which there is international consensus. These estimates are vital to implementing effective policies in a world where "foreign" ozone contributions are significant.

Atmospheric chemical transport models simulate the emission, chemical production and loss, transport, and removal of various coupled species within the atmosphere and allow us to assess theory against observational evidence. Atmospheric models can

also enable us to quantify various source contributions to concentrations of a particular chemical species in a given location or region. This is achieved by using, broadly, one of the two methods - *perturbation* or *tagging*. In the *perturbation* method, several runs are conducted where certain emission sources are removed or reduced and the resulting concentration fields are subtracted from the baseline run with full emissions to yield the contribution of the removed source. In the *tagging* method, generally a single simulation yields source contributions from different tagged regions or emission sectors. The contributions

derived from the perturbation method are not the true contributions operating under baseline conditions. Instead, they represent the response of all other sources to the removal of a particular source, which may be different from their contribution when all sources are present (Jonson et al., 2006; Burr & Zhang, 2011; Wild et al., 2012; Ansari et al., 2021). Therefore, perturbation experiments are best-suited to evaluate air quality policy interventions, when certain emission sources are actually removed (or reduced) or are planned to be removed in the real-world as part of policy. On the other hand, *tagging* techniques allow us

to assess the contribution of various sources under a baseline scenario when no policy intervention has been made. We refer



the reader to Grewe et al. (2010) for a first-principles discussion on perturbation versus tagging methods and to Butler et al. (2018) for a review of different tagging techniques.

With growing observational evidence of the increasing importance of "foreign" transported ozone, there have been many
attempts at confirming and quantifying these contributions using both perturbation-based and tagging-based model simulations for both North American and European receptor regions in recent years. For example, Reidmiller et al. (2009) used results from an ensemble of 16 models which conducted several regional perturbations for the year 2001, to report that East Asian emissions are the largest foreign contributor to springtime ozone in western US while European emissions are the largest foreign contributor in eastern US. Lin et al., (2015) disentangled the role of meteorology from changing global emissions in
driving the ozone trends in the US by performing sensitivity simulations with fixed emissions over their simulation period of 1995-2008. Strode et al. (2015) conducted a perturbation experiment where they only allowed domestic US emissions to vary over time but keep the remaining global emissions fixed at an initial year to better quantify the effect of changing foreign emissions on ozone in the US. Similarly, Lin et al. (2017) performed global model simulations with several perturbation experiments where emissions were fixed at the initial year over Asia and where US emissions were zeroed-out. They used the
difference between the simulated concentrations in their perturbation and base simulations to quantify the influence of local and foreign emission changes on the ozone concentrations in the US. Mathur et al. (2022) calculated emission source sensitivities of different source regions for the year 2006 using a sensitivity-enabled hemispheric model and applied these sensitivities to multi-decadal simulations to compute the influence of foreign emissions on North American ozone levels. They found a declining influence of European emissions and an increasing influence of East- and Southeast Asian emissions along
with shipping emissions on the spring- and summertime ozone in North America. Derwent et al. (2015) used an emissions-tagging method in a global Lagrangian model for the base year 1998 to explain the changing ozone seasonal cycle in Europe. Garatachea et al. (2024) performed three-year long regional model simulations with emissions tagging to calculate the import and export of ozone between European countries. Building on previous work, Grewe et al. (2017) introduced a new tagging method which assigns different ozone precursors into a limited number of chemical 'families' and attributes ozone to multiple
sources within each family. Mertens et al. (2020) used this tagging technique at a regional scale to calculate the contribution of regional transport emissions on surface ozone within Europe.

As pointed out earlier, perturbation-based estimates are more suited to evaluate an emissions policy intervention rather than to quantify baseline contributions of various sources (Grewe et al., 2010, 2017; Mertens et al., 2020). Tagging techniques, in
calculating baseline source contributions, can also have limitations. For example, they often tag combined $NO_X$ and VOC emissions over a tagged region or attribute ozone to the geographic location of its chemical production rather than the original location of its precursor emissions (as in Derwent et al., 2015) which can complicate policy-relevant interpretation of the model results. Some tagging techniques (as in Garatachea et al., 2024) tag ozone only to its limiting precursor in each grid cell thereby complicating detailed chemical interpretation of the computed contributions. While others (e.g., Grewe et al., 2017; Mertens





et al., 2020) attribute ozone molecules to tagged NO$_X$ and VOC depending on their abundances relative to the total amount of NO$_X$ and VOC present in each grid cell at each time step.

In this study, we use the TOAST tagging technique as described in Butler et al. (2018) which separately tags NO$_X$ and NMVOC emissions in two model simulations to provide separate NO$_X$ and VOC contributions from different regions and sectors to

simulated ozone in each model grid cell. The results from NO$_X$- and VOC-tagging can be compared side-by-side and the total contributions of all sources from both simulations add up to the same total baseline ozone. The TOAST tagging technique has been previously applied in both global (Butler et al., 2020; Li et al., 2023; Nalam et al., 2024) and regional models (Lupascu & Butler, 2019; Lupascu et al., 2022; Romero-Alvarez 2022; Hu et al., 2024) to calculate tagged ozone contributions over US, Europe, East Asia as well as the global troposphere.


We describe our model configuration, simulation design, input emissions data, and observations from the TOAR-II database used for model evaluation in section 2. In section 3.1, we present region-specific model valuation for the policy-relevant MDA8 O$_3$ metric. Key results on attribution of trends and seasonal cycle to NO$_X$ and VOC sources are presented in sections 3.2 for North America and section 3.3 for Europe. We finally summarise our key findings along with potential future directions in

section 4.

## 2 Methodology

### 2.1 Model description, tagged emissions, and simulation design:

We perform two 20-year long (1999-2018) global model simulations, with 1999 used as a spin-up year, using a modified

version of the Community Atmosphere Model version 4 with chemistry (CAM4-Chem) which forms the atmospheric component of the larger Community Earth System Model version 1.2.2 (CESMv1.2.2; Lamarque et al., 2012; Tilmes et al., 2015). The two simulations are identical in simulating the baseline chemical species including the total ozone mixing ratios, however, they are used to separately tag region- or sector-based NO$_X$ and VOC ozone precursor emissions respectively which ultimately allow us to break down ozone mixing ratios into their tagged NO$_X$ or VOC sources separately.


The model is run at a horizontal resolution of 1.9º×2.5º, a relatively coarse resolution which essentially allows us to compensate for the added computational burden due to the introduction of many new chemical species in form of tags and to effectively carry out two multi-decadal simulations. Vertically, the model was configured with 56 vertical levels with the top layer at approximately 1.86 hPa and roughly the bottom half of the levels representing the troposphere. The model is run as an offline

chemical transport model with a chemical time-step of 30 min and is meteorologically driven by prescribed fields from the



MERRA2 reanalysis (Molod et al., 2015) with no chemistry-meteorology feedback. The model is meteorologically nudged towards the MERRA2 reanalysis fields (temperature, horizontal winds, and surface fluxes) by 10% every time step.

We use the recently released Hemispheric Transport of Air Pollution version 3 (HTAPv3) global emissions inventory (Crippa

et al., 2023) to supply the temporally varying anthropogenic emissions input for $NO_X$, CO, $SO_2$, NH3, OC, BC and NMVOCs over 2000-2018 for our model runs. These include multiple sectors including several land-based sectors but also domestic and international shipping as well as aircraft emissions. We break down the global aircraft emissions spatially to denote three different flight phases based on EDGAR6.1: landing & take-off, ascent & descent, and cruising. Based on this spatial disaggregation of flight phases, we vertically redistribute the aircraft emissions at appropriate model levels for each flight

phase following the recommended vertical distribution in Vukovich & Eyth (2019). We also speciated the lumped NMVOCs as provided by the HTAPv3 emissions dataset, first, into 25-categories of NMVOCs as defined by Huang et al. (2017). This was done by using the regional (North America, Europe, Asia, and Other regions) speciation ratios specified for each sector by        Crippa        et        al.        (2023)        (see        table        here:        https://jeodpp.jrc.ec.europa.eu/ftp/jrc-opendata/EDGAR/datasets/htap_v3/NMVOC_speciation_HTAP_v3.xls). After obtaining the 25-category region- and sector-

based NMVOC speciation, we further speciated them into the appropriate NMVOC species as required by the MOZART chemical mechanism, which included merging as well as bifurcation of certain species. Biomass burning emissions are taken from GFED-v4 inventory (van der Werf et al., 2010) which provide monthly emissions for boreal forest fires, tropical deforestation and degradation, peat emissions, savanna, grassland and shrubland fires, temperate forest fires, and agricultural waste burning. The biogenic NMVOC emissions are taken from CAMS-GLOB-BIO-v3.0 dataset (Sindelarova et al., 2021),

while biogenic (soil) $NO_X$ is prescribed as in Tilmes et al. (2015). While we spatially interpolate the emissions from HTAPv3 high-resolution (0.1°×0.1°) dataset to our coarser model resolution (1.9°×2.5°), it leads to some land-based emissions at coastal areas to spill into the ocean grid cells and vice versa, thereby creating a potential for misattribution of tagged emissions. To correct this, we move these wrongly allocated land-based emissions over ocean grid cells back to the nearest land grid cells (and similarly, wrongly moved oceanic emissions to coasts back into the ocean) to make sure that the emissions are allocated

to the correct region for the source attribution. We also ensure that small islands which are smaller than the model grid cell area are preserved and their emissions are not wrongly attributed as oceanic or shipping emissions.

Our simulations do not resolve the full carbon cycle and do not have explicit methane emissions. Instead, methane concentration is imposed as a surface boundary condition. These methane concentrations are taken from the 2010–2018

average mole fraction fields from the CAMS $CH_4$ flux inversion product v18r1 (https://ads.atmosphere.copernicus.eu/cdsapp#!/dataset/cams-global-greenhouse-gas-inversion?tab=overview)        and        is specified as a zonally and monthly varying transient lower boundary condition. For upper boundary conditions, annually varying stratospheric concentrations of $NO_X$, $O_3$, $HNO_3$, $N_2O$, CO and $CH_4$ are prescribed from WACCM6 ensemble member of CMIP6 and are relaxed towards climatological values (Emmons et al., 2020).






Following the methodology of Butler et al. (2018 and 2020), as per the TOAST tagging system, we modify the MOZART chemical mechanism (Emmons et al., 2012) to include extra tagged species for the $NO_X$ tags and VOC tags, respectively, for the two simulations. This system allows us to attribute 100% of tropospheric ozone fully in terms of its $NO_X$ (+ stratosphere) sources and fully in terms of its VOC (+ methane + stratosphere) sources in two separate simulations. In the two simulations,

aside from the full baseline emissions, we additionally provide regionally- and sectorally-disaggregated $NO_X$ and VOC emissions, respectively, which undergo the same chemical and physical transformations in the model as the full baseline emissions. The regional tags are based on the HTAP Tier1 regions (Galmarini et al., 2017; see Figure 1). Since the focus of this study is to study ozone trends and its sources in North America and Europe, and because ozone is primarily a hemispheric pollutant (with little inter-hemispheric contributions), we explicitly tagged the land-based $NO_X$ emissions in the northern

hemisphere regions, namely, North America, Europe, East Asia, South Asia, Russia-Belarus-Ukraine, Mexico & Central America, Central Asia, Middle East, Northern Africa and Southeast Asia, while the southern hemisphere regions of South America, Southern Africa, Australia, New Zealand and Antarctica are tagged together as "rest-of-the-world". The ocean is also divided into many zones and tagged separately. In case of the VOC emissions, we use fewer explicitly tagged regions and some of the explicitly tagged $NO_X$ regions are aggregated with the "rest-of-the-world". This is done to ensure computational

efficiency given that tagging NMVOC means tagging several speciated NMVOCs within the MOZART chemical mechanism (as opposed to a single NO species in case of $NO_X$ tagging). In addition to the regional tags which carry anthropogenic emissions, we also tag other, mainly non-anthropogenic, global sectors separately: biogenic, biomass burning, lightning, aircraft, methane and stratosphere.

We specify an additional tag for $NO_X$ emission generated from lightning parameterization (Price and Rind, 1992; Price et al., 1997) in our $NO_X$-tagged simulation, and for methane in our VOC-tagged simulation. We refer the reader to Figure 1 for the geographic definitions of the various source regions and to Table 1 for more details on the regional and global tags for the $NO_X$ and VOC-tagging runs. Based on these tags changes were made to the model source code following Butler et al. (2018) which allows for physical and chemical treatment of all tagged species within the model.

Figure 2 shows the trends in $NO_X$ and VOC emissions for North America (NAM) and Europe (EUR) tagged source regions and for the northern hemisphere along with the global lightning $NO_X$ emissions and prescribed methane concentrations over the study period. We see a consistent decline in North American anthropogenic $NO_X$ emissions (Fig 2a) from ~250 Kg (N) s$^{-1}$ in 2000 down to ~100 Kg (N) s$^{-1}$. We also see a decline in European anthropogenic $NO_X$ emissions (Fig 2c), although starting

from a lower base in 2000, from ~140 Kg (N) s$^{-1}$ down to 80 Kg (N) s$^{-1}$. Similarly, the anthropogenic NMVOCs, or AVOCs, in the two regions (Figs 2b and d) have also declined substantially. These large emission changes reflect the strict and effective emission control policies implemented in these regions (Clean Air Act 1963, Clean Air Act Amendments 1990; Council Directive 1996, 2008). The biogenic $NO_X$ emissions peak in summertime for both regions but remain much lower (up to 40





Kg (N) s⁻¹ in North America and 20 Kg (N) s⁻¹ in Europe) than the anthropogenic NOₓ emissions and exhibit no long-term
trend. NOₓ emissions from fires remain extremely small. The biogenic NMVOCs, or natural VOCs, also peak during
summertime for both regions. This is due to the larger leaf area in the summer season (Guenther et al., 2006; Lawrence and
Chase, 2007). The natural VOCs for North America are higher than the AVOCs and show an increasing trend since 2013. The
natural VOC emissions in Europe are comparable to the AVOC emissions especially in recent years. The biomass burning
NMVOC emissions are the smallest but they show an increasing trend in North America. We have also plotted the total
northern hemispheric (NH) NOₓ and NMVOC emissions which can provide some context in understanding foreign
contributions to ozone in North America and Europe. Here, we see the NH anthropogenic NOₓ increasing from 2000 until
2013 after which it declines to below 2000 levels. This increasing trend is primarily driven by increasing Chinese emissions,
while the decline is driven by a decline in Chinese, North American and European emissions (not shown). We see a similar
trend for NH AVOC as well. Summertime NH natural VOC emissions exceed the AVOC emissions. NH biomass burning
NMVOC emissions are also significant, up to 5000 Kg C s⁻¹, but they are lower than natural VOC and AVOC emissions and
do not show any significant trend. Global lightning NOₓ emissions show a declining trend from ~100 Kg (N) s⁻¹ in 2000 to
~90 Kg (N) s⁻¹ in 2014 after which they increase to 95 Kg (N) s⁻¹ in 2018. The global methane concentration remains consistent,
around 1780 ppb, for 2000-2006 but rises steadily since 2007 reaching around 1880 ppb in 2018. Understanding these trends
in regional emissions of different ozone precursors allows us to better interpret tagged contributions to simulated ozone in later
sections.

**2.2 Model runs and initial post-processing:**

We perform two separate 20-year long simulations for 1999-2018. The first year, 1999, is discarded as a spin-up year and only
the outputs for 2000-2018 are used for further analyses. For the VOC-tagged run, the spin-up time was two years, such that
the 1999 run was restarted with the conditions at the end of the first 1999 run. Introducing extra tagged species with full
physical and chemical treatment in the model leads to a substantial increase in computational time (approx. 6x-8x) as compared
to a basic model run without tagging. Therefore, such a model configuration typically needs a large number of CPU cores
spread over multiple parallel nodes. We run our tagged simulations on 6-nodes with 72 Intel Icelake cores each (432 cores in
total) with a memory of 2048 GB per node. It takes approximately 24h and 36h wallclock time to complete a single year of
simulation with NOₓ- and VOC-tagging, respectively, with our model configuration. The VOC-tagged simulations take longer
despite having fewer land-based and oceanic tags because, unlike NOₓ-tagging, VOC-tagging involves all speciated NMVOCs
to be tagged separately thereby increasing the total number of chemical species to be treated in the model.

We configure the model to write out key meteorological and chemical variables, including tagged O₃ variables, as 3D output
at monthly average frequency but also write out the tagged O₃ variables at surface at an hourly frequency which allows us to
assess key policy-relevant ozone metrics for further analyses. Before we proceed to analyses of the results, we convert the



model output into global MDA8 $O_3$ (maximum daily 8h average) values along with its tagged contributions for each grid cell in the model. The model writes-out the hourly ozone values in Universal Time Coordinates (UTC) for all locations. Therefore, we first, consider different time-zones (24 hourly zones based on longitude range) and select the 24 ozone values by applying
the appropriate time-offset to reflect a "local day" for each grid cell. Once a 24h local-day has been selected, we perform 8h running averages spanning these 24 values and pick the maximum of these 8h averages as the MDA8 $O_3$ value for that grid cell on a given day. We then use the selected time window for the MDA8 $O_3$ value for the grid cell to also calculate the 8h-average tagged contribution over this window. Using this methodology, we prepare global NetCDF files which contain daily MDA8 $O_3$ values along with tagged contributions for each grid cell. We use these files for further analyses.


Figure 3 shows the geographic definitions of various HTAP-Tier2 regions (Galmarini et al., 2017), out of which nine regions, five in North America and four in Europe, shown in various shades of magenta and green, are used as receptor regions to perform further analyses of trends and seasonality in section 3. We use these receptor regions to perform area-weighted spatial averaging of MDA8 $O_3$ values before analysing the trends and contributions.


## 2.3 TOAR Observations and related data processing:

For model evaluation, we utilize ground-based observations of hourly ozone from many stations over North America and Europe which are part of the TOAR-II database of the Tropospheric Ozone Assessment Report (TOAR). We use the newly developed TOAR gridding tool (TOAR Gridding Tool 2024) to convert the point observations from individual stations into a
global gridded dataset which matches our model resolution of 1.9°×2.5°. The TOAR gridding tool allows for data selection including the variable name, statistical aggregation, temporal extent and a filtering capability according to the station metadata.

We extract the Maximum Daily 8h Average (MDA8) metric for ozone from the TOAR-II database analysis service (TOAR-II 2021) for the years 2000 to 2018 (as available until May 2024). The MDA8 values are only saved if at least 18 of the 24
hourly values per day are valid (see, *dma8epa_strict* in TOAR-analysis 2023). Also, since our model resolution is coarse, we only include rural background stations in our analyses to avoid influences of urban chemistry which may not be resolved in our model.

We use the *type_of_area* field of the station metadata to select the rural stations; this information is provided by the original
data providers (see Acknowledgements for an exhaustive list of data providers). They cover about 20% of all stations in North America and Europe. We note that roughly a similar fraction of stations in these regions remains unclassified. In the final gridded product, which contains daily MDA8 $O_3$ values over North America and Europe a grid cell has non-missing value if there is at least one rural station present within it. We obtain large parts of NAM and EUR regions with valid TOAR grid cells, although the number of these valid grid cells changes day-to-day and year-to-year. In North America, the number of valid



stations varies from 2-4 for Eastern Canada, 17-44 for NW US, 52-140 for SW US, 134-235 for NE US, 100-114 for SE US. In Europe, the number of rural stations varies from 201-236 for Western Europe, 57-223 for Southern Europe, 45-100 for Central & Eastern Europe, and 1-20 for SE Europe, with a general increase in the number of stations in each region with time, except for 2012 when there is an anomalous drop in the number of stations. Furthermore, the number of valid TOAR stations within each grid cell also varies for certain locations. To better understand the changes in the TOAR station network in each
of the 9 receptor regions considered here, we have plotted a time-series of annual average number of stations within each receptor region. This is shown in Figure S8.

### 3. Results:

### 3.1 Model Evaluation:

The CAM4-Chem model has been evaluated for its ability of simulating the distribution and trends of tropospheric ozone by many previous studies (Lamarque et al., 2012; Tilmes et al., 2015) including its modified version with ozone tagging (Butler et al., 2020; Nalam et al., 2024). Generally, many atmospheric models including CAM4-Chem have been shown to overestimate surface ozone in the Northern Hemisphere (Reidmiller et al., 2009; Fiore et al. 2009; Lamarque et al., 2012; Young et al., 2013; Tilmes et al., 2015; Young et al., 2018; Huang et al, 2021). In a recent study that utilized the same model
simulations as those presented in this study, Nalam et al. (2024) evaluated model simulated monthly average surface ozone against gridded observations from the TOAR-I dataset (Schultz et al., 2017) over various HTAP Tier 2 regions (Galmarini et al., 2017) in North America, Europe and East Asia for 2000-2014 and found a satisfactory performance, albeit with a general high bias of 4-12 ppb, similar to a reference CMIP6 model CESM2-WACCM6 (Emmons et al., 2020); see Figure 1 in Nalam et al., 2024 for more details. Furthermore, Nalam et al. (2024) have also evaluated the model simulated monthly mean ozone
against the ozone sonde-based climatology compiled by Tilmes et al. (2012) for different latitude bands in the northern hemisphere at different pressure levels over the same period and found generally high correlations and low biases - see Figure 2 in Nalam et al. (2024) for further details.

One reason for a high bias as seen in Nalam et al., (2024) and other studies could be the use of all available stations (including
many urban stations) for evaluating the model performance. Given the coarse model resolution, we expect the model not to resolve high $NO_X$ concentrations around the urban and industrial centres and therefore suffer from the lack of ozone titration. Therefore, here, we only evaluate the model against data from rural stations, wherever available. Also, in this study, we only work with policy-relevant metrics such as Maximum Daily 8h Average (MDA8) Ozone at the surface or other metrics derived from it, e.g., Peak Season Ozone (PSO). These metrics generally include only the daytime ozone, especially over land.
Therefore, evaluating the model for these metrics also allows us to exclude nighttime ozone and avoid any large nighttime



biases which often arise due to improper simulation of the nighttime boundary layer which has been a persistent issue in both global and regional models (Houweling et al., 2017; Du et al., 2020; Ansari et al., 2019).

For model evaluation, we derive regionally averaged monthly mean MDA8 $O_3$ for all HTAP tier 2 receptor regions for North
America, Europe and Asia but sample the MDA8 $O_3$ values only from those gridcells where rural TOAR observations were available. Figure 4 shows the time-series of monthly mean MDA8 $O_3$ from the model and TOAR observations for the entire simulation period. We ask the reader to refer to the geographic extent of the receptor regions discussed here in Figure 3.

In Eastern Canada (Figure 4a), the model reproduces the $O_3$ seasonal cycle very well, especially between 2007-2018. It
overshoots the maxima and undershoots the minima for the earlier years of 2000-2006. This could be due to inaccurate (higher) $NO_X$ emissions over the region in the HTAPv3 inventory for the earlier years which leads to higher summertime production and lower wintertime levels due to increased titration. The model also reproduces the flattening annual cycle well which is consistent with decreasing $NO_X$ emissions over this region (see Figures 3 and S3). For the Northwestern United States (Figure 4b), the model reproduces the annual cycle very well, although it systematically overestimates the MDA8 $O_3$ during peak
season by up to 5 ppb. For the Northeastern United States (Figure 4c), the model captures the structure of the annual cycle of MDA8 $O_3$ very well for recent years but overestimates the summer peak and underestimates wintertime ozone for earlier years, similar to Eastern Canada, again pointing to high $NO_X$ emissions in the emission inventory over this region in the initial years. The model shows an extremely skilful simulation of MDA8 $O_3$ in the Southern United States. In Southwestern US (Figure 4d), the model reproduces the gradual and steady decline in MDA8 $O_3$ over time, albeit with a slight overprediction (~2ppb) in
later years. Similarly, in the Southeastern US (Figure 4e), we note a very good reproduction of trends, with a decreasing summertime peak. For all North American regions, we see a high correlation between observed and modelled values with correlation coefficient r ranging from 0.86 to 0.98. Mean bias is positive for all regions but small, ranging from 0.68 ppb to 3.65 ppb.

The model reproduces the MDA8 $O_3$ for Europe extremely well with very small mean biases (-1.54 ppb to 1.25 ppb) and very high r values ranging from 0.94 to 0.97 for various regions, except Southeastern Europe. For Western Europe (Figure 4f), it captures both the trends and the structure of the seasonal cycle extremely well, for example, note the near-stagnant maxima and increasing minima over time in both observations and model output. Similarly for Southern Europe (Figure 4g), we again see a very skilful simulation of monthly mean MDA8 for the entire simulation period - this includes capturing the slightly
decreasing summer maxima and increasing winter minima and an overall flattening of the seasonal cycle post 2006. We see a very good reproduction of MDA8 $O_3$ for Central & Eastern Europe (Figure 4h) particularly for the summer months. We see a small underprediction for the winter months in years up to 2012. However, it is the summertime MDA8 $O_3$ values that constitute the peak season ozone metric which are ultimately utilized in our further policy-relevant analyses. Finally, for Southeastern Europe (Figure 4i), we notice an overprediction of MDA8 $O_3$ for early years, until 2006, after which the model





captures the trends and particularly the summer peaks very well. The mean bias is 7.63 ppb and r value is 0.62. We have also

included the Belarus & Ukraine region (Figure 4j) in our evaluation and here too we see a good simulation of MDA8 $O_3$ for

the entire period (with a small mean bias of 0.56 ppb and r value of 0.83), barring a couple of years (2014 and 2017) when the

model overestimates the values.

We have also evaluated the model for MDA8 $O_3$ against observations from the TOAR-II database in other regions including

Mexico, North Africa, Southern Africa, Latin America, and Eastern Russia, where the model has also captured the trends well,

however, since we do not discuss these regions in further analyses, they are presented in the supplement (see, Figure S1).

Overall, we obtain very good model-observations agreement, with low biases and high correlations, better than previous studies

(e.g., Butler et al., 2020; Li et al., 2023; Garatachea et al., 2024). The possible reasons for such improved performance could

be 1) the use of the newly developed HTAPv3 emissions inventory 2) using only rural stations for evaluation which avoids

urban titration which may be in the observations but not in model output 3) improved treatment of spatial and temporal

representativeness (including the treatment of missing values) of the stations through the TOAR gridding tool 4) evaluating

the policy-relevant MDA8 $O_3$ metric which avoids nighttime $O_3$ which may not be well-simulated due to improper estimation

of the nighttime boundary layer.

After a satisfactory performance of the model across different world regions and, in particular, excellent performance in the

simulation of MDA8 $O_3$ against rural stations from the TOAR-II database, we proceed to further analyses of trends and source

contributions to ozone in different receptor regions. First, to explain the year-to-year trends, we present the full 19-year time

series of Peak Season Ozone (PSO) for North America and Europe along with their $NO_X$- and VOC- source contributions

derived from our two tagged simulations. After explaining the year-to-year trends in ozone in terms of the $NO_X$ and VOC

contributions, we further calculated a 19-year month-centered average MDA8 $O_3$ and its source contributions for each receptor

region. This allows us to interpret the leading sources of ozone in each receptor region on a monthly basis averaged over the

entire simulation period. We then break down this 19-year month-centered average MDA8 $O_3$ seasonal cycle into a past (first

five years) and recent (last five year) averaged seasonal cycle and explain the shifts in terms of tagged contributions for all

receptor regions during these periods. In the next subsections, we present these results for North America and Europe.

**3.2 Ozone in North America:**

**3.2.1 Peak Season Ozone in North America: Trends and Source Contributions:**

In this section we discuss the trends in and contributions to PSO in North America. The Peak Season Ozone for any location

is defined as the highest of the 6-month running average of monthly mean MDA8 $O_3$ values. In order to compute PSO, we

performed the averaging over 6-month windows (Jan-Jun, Feb-July, Mar-Aug and so on) over the TOAR observations and the



same time window was imposed over the modelled values for calculating the 6-month averaging (instead of independently selecting the peak 6-month time window for the model). This approach ensures temporal consistency between the observations and modelled values. Furthermore, for spatial consistency, the model values were sampled only from those grid cells where at least one TOAR-II station was present. Finally, these values from multiple grid cells were spatially averaged over various receptor regions after weighting them with the grid cell areas to derive a single PSO value per region per year for observations and the model along with tagged contributions.

Figure 5 shows the observed versus model-simulated time series of Peak Season Ozone along with its $NO_X$- and VOC-source contributions for five different receptor regions within North America (see Figure 2 for geographic definitions). We note that, for all regions in North America, the observed PSO exceeds the WHO guidelines throughout the 2000-2018 period. For each row, the left and right panels show the same observed and model-derived PSO in dotted lines for a given receptor region but break it down in terms of the $NO_X$ contributions and VOC contributions respectively, thereby providing us two distinct perspectives of seeing ozone in terms of its contributors. In terms of $NO_X$ contributions, PSO is broken down in terms of local anthropogenic $NO_X$ contribution, foreign anthropogenic $NO_X$ contribution which also includes global aircraft contribution, natural $NO_X$ contribution which is a sum of biogenic, fire and lightning $NO_X$ contribution, global shipping $NO_X$ contribution, and stratospheric intrusion, regardless of the origin of the VOCs that interacted with them. It describes 100% of ozone at any given receptor wholly in terms of its $NO_X$ sources only. Similarly, the right-hand panels describe the same ozone in terms of its VOC sources + global methane irrespective of its $NO_X$ sources. Here, the different contributors are, local anthropogenic VOC sources, foreign anthropogenic VOC sources (including global aircraft VOC), natural VOC which is a combination of global biogenic VOCs and fire VOCs, global shipping VOC contribution, methane contribution, and stratospheric intrusion, which again explains the entire 100% of ozone abundance for any given receptor region. Analysing these contributions side-by-side can also provide qualitative insights into possible interactions between different $NO_X$ and VOC sources along with some insights into plausible regional control measures.

Figures 5 a & b show PSO for Eastern Canada in terms of $NO_X$ and VOC contributions respectively. Overall, we see a slight negative trend in the observed PSO (-0.24 ppb/yr, (1.0)) with magnitudes in the range of 40-45 ppb. The model captures the PSO magnitude well but overestimates the trend (-0.35 ppb/yr, (0.99)). From the $NO_X$ source perspective (Fig 5a), we see that the largest contribution is from local anthropogenic $NO_X$ sources although with a declining trend of (-0.75 ppb/yr, (1.0)) over the 19-year period. The declining trend in the local $NO_X$ contributions is sharper than the trend in overall PSO because all other sources show a small positive trend (see table S1 for details) which partially compensates the negative trend in local $NO_X$ contributions. Despite declining trends, local $NO_X$ remains the largest contributor (~15ppb) to PSO while each of the remaining contributions, though increasing, remain below 10 ppb. In terms of VOC contributions (Fig 5b), methane contribution is largest, at around 15 ppb. This is followed by natural VOC and local AVOC contributions. The declining trend in overall PSO is explained by declining trends in local AVOC contributions (-0.32 ppb/yr, (1.0)) and natural VOC contributions (-0.17 ppb/yr,

(0.89)), partially offset by an increasing trend in stratospheric (0.12 ppb/yr, (0.99)) and foreign AVOC contributions (0.09 ppb/yr, (0.99)). It is worth noting that the year-to-year peaks and troughs in the local $NO_X$ contributions correspond neatly with the natural VOC contributions and are also reflected in the overall shape of the PSO time series. This suggests a large interaction between local anthropogenic $NO_X$ and local natural VOCs in the region.


Figures 5 c and d show PSO time series for the Northwestern United States. We see PSO values around 45 ppb throughout the period but with a slight decreasing trend (-0.11 ppb/yr, (0.82)). The model overestimates the magnitude, for reasons discussed in the previous section, but reproduced the small declining trend very well (-0.11 ppb/yr, (0.97)). In the early years, local anthropogenic $NO_X$ contribution remains the largest but declines steadily (-0.38 ppb/yr, (1.0)) to become comparable to foreign
$NO_X$ contributions by 2011. In recent years, the foreign $NO_X$ contribution exceeds the local $NO_X$ contributions. The declining contribution of local $NO_X$ can be linked with the large decline in local $NO_X$ emissions in this region along with a steady increase in northern hemispheric anthropogenic $NO_X$ emissions (see Figure 3). In terms of VOC contributions, methane remains the largest contributor with a steady contribution at around 20 ppb. This is followed by natural VOC contributions (10-12 ppb). Here, the overall decline in PSO is almost single handedly associated with the declining trend in local AVOC
contributions (-0.15 ppb/yr, (1.0)). This decline can be linked to a combination of the decline in the North American AVOC and $NO_X$ emissions (see Figure 3). There is also a small declining trend (-0.03 ppb/yr, (0.97)) in natural VOC contributions.

Figures 5 e and f show PSO for Southwestern US. Here we see the highest PSO of any other region considered in our analysis, with concentrations reaching 60 ppb in the early years declining at (-0.34 ppb/yr, (1.0)) to reach 55 ppb in 2018, still well
above the WHO guideline of 31 ppb. The model slightly overestimates the magnitude but captures the decreasing trend reasonably well (-0.25 ppb/yr, (1.0)). There is a very sharp downward trend (-0.71 ppb/yr, (1.0)) in local $NO_X$ contributions which is partially offset by an increasing trend in foreign anthropogenic $NO_X$ contributions (0.2 ppb/yr, (1.0)). These two together explain the decreasing trend in overall PSO. This region has seen a dramatic reduction in local $NO_X$ emissions such that they were the single largest contributor to ozone in the initial years (up to 27 ppb) with more than double the contributions
of foreign $NO_X$, but in recent years the local $NO_X$ contributions have declined to 16 ppb which is comparable to foreign $NO_X$ and natural $NO_X$ contributions. In terms of VOC contributions, methane remains the largest contributor, at around 25 ppb albeit with a very small decreasing trend (-0.09 ppb/yr, (1.0)). This is remarkable given the rapidly increasing background concentration of methane, but is consistent with the lower availability of local $NO_X$ during methane oxidation for producing ozone. Given the arid climate and sparse vegetation of this region, natural VOC contribution is much lower, at around 14-18
ppb. Similar to Eastern Canada, the stratosphere contributes up to 6-8 ppb, while foreign and local AVOC contributions remain low, beginning at equal strengths at around 5 ppb but followed by a steady decline in local AVOC contribution (-0.25 ppb/yr) as also seen in other parts of the United States.





Figures 5 g and h show PSO time series for Northeastern United States along with its NOₓ and VOC contributions respectively.
Here, we see a substantial decline in observed PSO (-0.43 ppb/yr, (1.0)) from around 50 ppb in early 2000s down to 45 ppb in 2018. The model overestimates the magnitude but reproduces the declining trend well (-0.52 ppb/yr, (1.0)). From a NOₓ source perspective, the PSO decline in this region is driven by a dramatic decline in local NOₓ contributions from ~40 ppb to ~20 ppb (-0.94 ppb/yr, (1.0)) which is partially offset by a steadily increasing foreign NOₓ (0.17 ppb/yr, (1.0)) and natural NOₓ contribution (0.13 ppb/yr, (1.0)). It is notable that the natural NOₓ contribution is increasing despite no increase in natural
NOₓ emissions (see Figure 2) which is consistent with the natural NOₓ emissions becoming more productive due to overall lower NOₓ levels (Liu et al., 1987). Stratospheric contribution remains low between 4-7 ppb and the ship NOₓ contribution is the lowest, 0-2 ppb, albeit with an increasing trend consistent with the increasing shipping NOₓ emissions. In terms of VOC contributions, we see comparable contributions from methane and natural VOC, around 18 ppb. The higher natural VOC contribution in this region suggests ample availability of natural VOC through vegetation and also ample local NOₓ nearby
the natural VOC sources. The peaks and troughs in local NOₓ contributions and natural VOC contributions are coincident which also points to their interaction in this region. The declining trend in PSO can be explained by the declining local AVOC contribution (-0.37 ppb/yr), natural VOC (-0.25 ppb/yr) and methane contributions (-0.11 ppb/yr) which shows that the ozone produced through the oxidation of VOCs is responding to the declining local NOₓ emissions, especially because the natural VOC emissions and methane concentrations, themselves, are rising (Figure 2).


Finally, Figures 5i and j show the PSO time series for the Southeastern US. This region shows the sharpest decline in observed PSO than any other receptor region in North America (-0.47 ppb/yr, (1.0)). The contributions are similar to those in Northeastern US: a sharp decline in local NOₓ contribution (38 ppb to 20 ppb; -1.07 ppb/yr, (1.0)) which remains the largest contributor even after the decline, and modest increases in foreign NOₓ, natural NOₓ and ship NOₓ contributions (see Table
S1 for quantitative trends). Natural NOₓ and foreign NOₓ contributions remain around 10 ppb while stratospheric and ship NOₓ contributions are under 5 ppb. In terms of VOC contributions, methane and natural VOC contributions are comparable and explain part of the declining trend in PSO (-0.16 ppb/yr, (1.0)) and (-0.33 ppb/yr, (0.9)). The remaining part of the declining trend is captured by a steady decline in local AVOC contribution which reduces from 10 ppb to under 4 ppb over the 19-year period; (-0.33 ppb/yr, (1.0)). Again, the peaks and troughs in natural VOC contribution coincide with those in the local NOₓ
contribution suggesting their interaction in ozone formation in this region. The quantitative Thiel-Sen trends for observed and modelled PSO in all receptor regions and their tagged contributions are included along with their significance in Table S1.

### 3.2.2 Ozone seasonal cycle in North America: Trends and Source Contributions:

Figure 6 shows the 2000-2018 average seasonal cycle of MDA8 O₃ over the different receptor regions within North America
along with its source contributions. We see elevated levels of MDA8 O₃ in spring and summer and lower levels in winter, in line with the scientific understanding of ozone photochemistry (Logan 1985; Seinfeld & Pandis 2016). The model reproduces the 19-year average seasonal cycle over different parts of North America very well. For western regions, we see a consistent



systematic positive bias of 2-4 ppb. For eastern regions we see a very good reproduction of the seasonal cycle during winter and spring but a notable overestimation during summertime.


Figure 6a and b shows the average seasonal cycle of MDA8 $O_3$ in Eastern Canada along with its $NO_X$ and VOC source contributions respectively. The MDA8 $O_3$ seasonal cycle in this region is characterized by a springtime peak (Mar - Apr; ~44 ppb) and a decline in the summertime (Jul - Sep; ~35 ppb). The springtime peak is driven by peaks in foreign anthropogenic $NO_X$ contribution and stratospheric intrusion along with high local $NO_X$ contribution. The summertime peak in the model (not seen in observations) is composed of peaks in local $NO_X$ and natural $NO_X$. This modelled but not observed peak is likely the reason for the high model bias in this region seen in figure 4. And since the model performs well for springtime, this summertime high bias points to nearby emissions being too high, or alternatively, an overactive photochemistry (see also NE and SE US). In terms of VOC contribution, the springtime peak is composed of methane contribution (12-14 ppb), stratospheric intrusion (up to 10 ppb), foreign AVOC contribution peak (8 ppb) and an increasing share of local AVOC contribution (~6ppb). natural VOC contribution peaks in the summertime, when there are more leaves and emissions of natural VOC - this also drives the summer peak in modelled PSO which is not seen in observations. The summertime model-observations gap warrants further investigation into uncertainties in local anthropogenic $NO_X$ as well as local natural VOC emissions to further attribute this mismatch. Methane remains the highest overall contributor in terms of VOCs with slightly higher levels than foreign $NO_X$ contributions suggesting its substantial interaction with both local and foreign $NO_X$ in production of local as well as transported ozone in this region.

Figures 6c and d show the average seasonal cycle of MDA8 $O_3$ in Northwestern US in terms of $NO_X$ and VOC source contributions respectively. Here, we see high MDA8 $O_3$ from spring through summer in observations. The shape of the seasonal cycle is skilfully captured by the model albeit with a high bias. In contrast to the eastern regions, the bias here is high all year. This points to an overestimation of the background ozone rather than a high bias in local emissions and photochemistry. The spring peak is primarily driven by peaks in foreign $NO_X$ and stratospheric intrusion. Ship $NO_X$ contributions, although small, peak during springtime. Summer highs are driven by highs in local $NO_X$ and natural $NO_X$ contributions. A peculiar feature is a sustained high foreign anthropogenic $NO_X$ contribution throughout the year which only dips in the summertime. This summertime dip in foreign contribution is likely because ozone lifetime is reduced at higher temperatures due to the increased ability of air to hold water vapour (Stevenson et al., 2006) and long-range transported ozone can be destroyed when it encounters moisture (Real et al., 2007). Thus, the overall long-range transport efficiency of ozone is reduced during summertime. In terms of VOC contributions, springtime peak is primarily composed of methane, stratospheric, foreign AVOC contributions with smaller contributions from natural VOC and local AVOC. Summertime peak is composed of a peak in methane contribution and natural VOC peak. A natural VOC peak is expected during summertime due to a high leaf area during this time of the year. All other VOC contributions are very small during summertime in this region.





Figures 6e and f show the average seasonal cycle of MDA8 $O_3$ in Southwestern US which is similar to that for the Northwestern US and is well reproduced by the model. Springtime peak is dominated by foreign $NO_X$ and stratospheric contributions but also composed of an increasing local $NO_X$ and natural $NO_X$ component. Summertime peak is driven by local $NO_X$ and natural

$NO_X$ contributions. In terms of VOC contributions, methane, stratosphere and foreign AVOC drive the springtime peak while methane and natural VOC contributions drive the summertime peak.

Figures 6g and h show the average seasonal cycle of MDA8 $O_3$ in Northeastern US which is characterized by a major springtime peak which declines over the summer until the winter months. The model skilfully captures the seasonal cycle for

the first five months but overestimates the summertime ozone. Both the $NO_X$ and VOC contributions show a similar cycle as in western US regions except for a very large local $NO_X$ peak (and a corresponding natural VOC contribution peak) which drives the modelled summertime peak not seen in observations. Unlike the western regions, the summertime natural VOC contribution exceeds the methane contribution by a large margin and reaches up to 25 ppb. The higher natural VOC and lower methane contributions broadly correspond with the higher local $NO_X$ and lower remote $NO_X$ contributions. The accuracy of

natural VOC emissions in this region is a matter of further investigation. There is also a sustained higher local AVOC contribution (> 5ppb) than in western US.

Figures 6i and j show the average seasonal cycle of MDA8 $O_3$ in the Southeastern US which is very similar to that in the Northeastern US. The model reproduces the observed seasonal cycle well although with an overestimation of the summer

peak. The shape of the seasonal cycle is primarily driven by local anthropogenic $NO_X$ contributions from a $NO_X$ perspective and by methane and natural VOC contributions from the VOC perspective. Foreign anthropogenic $NO_X$ contributions are high, up to 10 ppb, during spring and winter but dip to around 5 ppb in the summer.

**3.2.3 Changes in seasonal cycle of ozone in United States: Role of Local vs Remote contributions**
A careful analysis of the dominant contributors to MDA8 $O_3$ seasonal cycle for different months alongside the changing dominant contributors to PSO over the two decades suggests that the seasonal cycle as well as its composition must be changing significantly over the years. This led us to plot full envelopes of MDA8 $O_3$ cycles (instead of averages) to fully assess the changes in the shape of the $O_3$ seasonal cycle for different receptor regions. These envelope plots are shown in the supplement

(Figure S2) which reveal the changes in the MDA8 $O_3$ seasonal cycle year-to-year. In Figure S2, we note that generally the spring and summertime MDA8 $O_3$ is decreasing while the wintertime $O_3$ is increasing for many regions which is consistent with decreasing local anthropogenic $NO_X$ emissions reducing titration in winter and local production in summer. The wintertime ozone increase could also be partly due to increasing transported ozone from foreign contributions.



To better understand how the seasonal cycle has changed over these two decades, we present the initial and final 5-year averaged MDA8 $O_3$ seasonal cycles (over 2000-2004 and 2014-2018, respectively) along with their $NO_X$ and VOC contributions. Figures 7 a and d show the observed and modelled initial 5-year and final 5-year average seasonal cycles for Northwestern US. We see that between these two periods, the spring and summertime ozone has decreased while the wintertime ozone has increased. The model reproduces these seasonal changes reasonably well but with a high bias of up to 4

ppb. We see (in Figs 7b and e) that these changes in the seasonal cycle are driven by a substantial drop in local $NO_X$ contributions especially in the summer along with an increase in summertime natural $NO_X$ contribution which partially compensates for the drop in local anthropogenic $NO_X$ contributions. As noted in the previous sections, there is no increase in the natural $NO_X$ emissions in these two decades (Figure 2) however, under lower $NO_X$ conditions, the same natural $NO_X$ becomes more productive in forming ozone during summer (see Liu et al., 1987). The wintertime increases are primarily driven

by an increased foreign $NO_X$ contribution along with a small increase in ship $NO_X$ contribution. From a VOC perspective (Figs 7c and f), the biggest changes occur in the local AVOC contributions which have declined throughout the year (probably in response to the declining local $NO_X$ emissions but also a decline in their own emissions; see figure 2). Wintertime increase in MDA8 $O_3$ is composed of increases in methane and foreign AVOC contributions in winter. Summertime decrease is associated with a decrease in methane and local AVOC contributions in the summer.


Figure 8 shows a similar analysis but for Northeastern US. Here, in the observed seasonal cycle, we see a small decrease in springtime ozone (~48 ppb to ~45 ppb), a large decrease in summertime (~48 ppb to ~40 ppb), and an increase in wintertime ozone (28-32 ppb to 30-36 ppb). For the initial 5-year period, the model overestimates the summertime peak by a large margin (~10 ppb) and underestimates the wintertime levels by 4-8 ppb. This is likely due to high anthropogenic $NO_X$ over this region

in the HTAPv3 emissions dataset and has also been discussed in the model evaluation section (see section 3.1 and Figure 3c). The model captures the seasonal cycle for the final 5-year period much better over the winter and spring seasons but the summertime overestimation remains. However, the model is able to capture the directional changes in the seasonal cycle: small decrease in spring, large decrease in summer and an increase in winter. These changes can be understood in terms of decreasing summertime local $NO_X$ contributions and increasing wintertime and springtime foreign $NO_X$ contributions (Figs 8b and e).

From a VOC point of view, the summertime drop is primarily due to a large drop in local AVOC contributions and to a lesser extent in natural VOC contributions, while the wintertime increase is due to an increase in methane contributions. We have performed a similar analysis for other receptor regions within North America which can be found in the supplement (Figures S3-S5). Our results are in agreement with observational studies which have also reported a decline in summer peaks and a shift in peak ozone to the springtime in North America (Bloomer et al., 2010; Parrish et al., 2013; Cooper et al., 2014). For the first

time, through our tagging technique, we are able to explain these changes in terms of $NO_X$ and VOC source contributions from local and remote regions. It is crucial to note the increased share of foreign $NO_X$ contributions to springtime ozone which coincides with the growing season and highlights the increasing impact of transported ozone on crop yields (Dingenen et al., 2009; Avnery et al., 2011).



### 3.3 Ozone in Europe:

Here, we present the observed and model-derived results for different sub-regions in Europe: Western Europe, Southern Europe, Central & Eastern Europe, and Southeastern Europe (see Figure 2 for geographical extents). We first present trends in PSO along with their $NO_X$ and VOC contributions and then show the 19-year average seasonal cycle of MDA8 $O_3$ and its source contributions, and finally present changes in the seasonal cycle between initial and the final 5-years. Europe has undergone significant reductions in $NO_X$ emissions over the past decades (see Figure 3), particularly in Western and Southern Europe. However, some countries in Central and Eastern Europe have not yet achieved the same level of reductions, suggesting potential variability in ozone trends across the continent. This raises important questions about how these uneven $NO_X$ reductions might influence ozone formation dynamics in different sub-regions, which we will explore in detail in this section using our tagged model results.

### 3.3.1 Peak Season Ozone in Europe: Trends and Source Contributions:

Figure 9 shows the observed and modelled PSO in different regions of Europe along with the corresponding $NO_X$ and VOC source contributions. We note that despite the large decline in European anthropogenic $NO_X$ and NMVOC emissions (Figure 2) over the two decades, the observed PSO values exceed the WHO guidelines in all regions.

Figures 9 a and b show the observed and modelled PSO for Western Europe along with its $NO_X$ and VOC contributions respectively. The model does a near-perfect job of reproducing the magnitude and trend of the observed PSO (see Table S1 for quantitative trends). It also captures the high PSO for 2003 and 2006 which were associated with summertime heatwaves in Europe (Vautard et al., 2005; Solberg et al., 2008; Struzewska & Kaminski, 2008). We do not see any significant trends in the PSO for this region over the 19-year period. There is a decline in the local $NO_X$ contribution (-0.26 ppb/yr, (1.0)) but it is partially compensated by small increasing trends in foreign $NO_X$ (0.06 ppb/yr, (0.99)) and ship $NO_X$ (0.12 ppb/yr, (1.0)) contributions. These results demonstrate that the local $NO_X$ emission controls did not translate into the local air quality improvement in this region, at least in terms of the policy-relevant PSO metric. Although, other studies have highlighted that summertime ozone extremes have been reduced in recent decades (Yan et al., 2018; Crespo-Miguel et al., 2024). From the VOC perspective, methane is the largest contributor at around 18 ppb with an increasing trend (0.08 ppb/yr, (1.0)) which is followed by natural VOC, stratosphere, foreign AVOC and local AVOC contributions in that order, all contributing between 4-10 ppb. The small declining trend in PSO is mainly captured by a declining trend in the local AVOC contributions (-0.16 ppb/yr, (1.0)) while other VOC contributions show modest trends (see Table S1 for details).

Figures 9 c and d show the observed and model-derived PSO for Southern Europe along with its $NO_X$ and VOC contributions respectively. The model captures the magnitude and trend of PSO extremely well. Here, we see a gentle decline in observed





PSO (0.04 ppb/yr, (0.52)) from ~50 ppb in early years to ~46 ppb in 2016, albeit with an uptick in the final two years. The model captures the trend well for a large part of the time series but overestimates the overall decline (-0.17 ppb/yr, (0.98)). This declining trend is driven by a noticeable decline in local $NO_X$ contribution (-0.51 ppb/yr, (1.0)) partially compensated by

increasing trends in foreign $NO_X$ (0.07 ppb/yr, (0.98)) and ship $NO_X$ (0.16 ppb/yr, (1.0)) contributions. Despite the decline, local $NO_X$ remains the largest contributor throughout the year, at 25 ppb in 2000 and 19 ppb in 2018. The large gap between the local $NO_X$ and foreign $NO_X$ contributions in early years has narrowed in recent years - and foreign anthropogenic $NO_X$ contributions are becoming an important source of transported ozone in this region. In terms of VOC contributions, methane and natural VOC remain the largest contributors. The variability in the PSO time series corresponds with the variability in

local $NO_X$ contributions in the left panel and natural VOC contributions in the right panel, suggesting their interaction. From a VOC perspective, the declining PSO trend is mainly associated with declining local AVOC contributions (-0.22 ppb/yr, (1.0)).

For Central & Eastern Europe (Figures 9 e and f), we see a noticeable negative trend of -0.43 ppb/yr (1.0) in the observed PSO.

The model captures the PSO magnitude well but with a small underestimation for the early years and overestimation for the later years, which leads to a smaller negative modelled trend of -0.04 ppb/yr (0.47). Similar to Southern Europe, local $NO_X$ contributions are the largest contributor but with a consistent decline (-0.27 ppb/yr, (1.0)) while foreign $NO_X$ contributions are increasing (0.1 ppb/yr, (0.99)). Other contributions remain small. The VOC contributions are very similar to those seen in Southern Europe where the declining PSO trend is primarily captured by a decline in local AVOC contributions (-0.18 ppb/yr,

(1.0)) which is consistent with both decreasing emissions of AVOC and decreasing availability of $NO_X$ for ozone production (Figure 2).

Finally, Figures 9 g and h show PSO time series for Southeastern Europe along with its $NO_X$ and VOC contributions respectively. Here, we see a large model-observations gap for the early years which narrows and closes towards the later years.

There is considerable year-to-year variability in the observations which is not reproduced in the modelled results. This could be due to the complicated nature of model sampling from TOAR-valid grid cells which are changing from year-to-year while the number of stations within a grid cell are also changing rapidly in the region. We have explored the TOAR station-network for each of the receptor regions and plotted the number of valid stations per region as a time series in Figure S8. We see that Southeastern Europe only had 1-3 rural stations in the initial years which increased to up to 20 stations towards the end. Such

a rapidly changing station network, especially when happening within a model grid cell, can complicate the model-observation agreement and interpretation. Due to these sampling issues, we do not overinterpret the results for this region. Instead, we refer the reader to Lin et al., (2015) for a discussion on the dependence of the modelled ozone trends on the co-sampling with observations.





### 3.3.2 Ozone seasonal cycle in Europe: Trends and Source Contributions:

Figure 10 shows the 19-year average seasonal cycle of MDA8 $O_3$ for different sub-regions of Europe along with its $NO_X$ and VOC source contributions. The observed seasonal cycle is distinct in each receptor region: we see a major spring peak in Western Europe, a sustained spring-to-summer peak in Southern Europe and Central & Eastern Europe, and a major summer peak in Southeastern Europe. The model reproduces the average seasonal cycles in these regions extremely well, particularly in Western and Southern Europe. The model underestimates the MDA8 $O_3$ for Central & Eastern Europe in winter months and systematically overestimates the full seasonal cycle for Southeastern Europe.

For all regions, we see that, in the left panels, the local anthropogenic $NO_X$ and natural $NO_X$ contributions peak in the summertime, along with methane and natural VOC contributions in the right panels. The foreign $NO_X$ and stratospheric contributions peak in the springtime. In all sub-regions, the springtime peak is composed of a peaking contribution from foreign $NO_X$ and stratosphere along with an increasing local $NO_X$ contribution. Methane remains the highest contributor throughout the year in terms of VOC contributions for all sub-regions. The lack of a summer peak for Western Europe is explained by lower local $NO_X$ contributions as compared to other regions. For all regions, the wintertime MDA8 $O_3$ levels are sustained by high foreign $NO_X$ contributions, mostly greater than 10 ppb. Ship $NO_X$ contribution remains low, but can reach up to 5 ppb in spring and summer. Foreign AVOC contributions remain low, below 10 ppb, much lower than the foreign $NO_X$ contributions, pointing to their low interaction and potentially a higher interaction of foreign $NO_X$ with natural VOC and methane globally.

### 3.3.3 Changes in seasonal cycle of ozone in Europe: Role of Local vs Remote contributions

A long-term average of the ozone seasonal cycle as shown in the previous section provides us with a general sense of monthly contributions from various sources but it may conceal the (possibly large) year-to-year variations within the cycle. Therefore, in this section we compare the early 5-year average seasonal cycle with the recent 5-year seasonal cycle to understand the changing shape of the cycle and its contributing factors in terms of $NO_X$ and VOC sources. Figure 11 presents the observed and modelled 5-year averaged MDA8 $O_3$ seasonal cycles for the initial (2000-2004) and final (2014-2018) periods along with their $NO_X$ and VOC contributions for Western Europe. The model captures both the spring and summer peaks and their changes in this region extremely well. Between these initial and final periods, we see a significant drop in the summer peak (from 44 ppb to 40 ppb) along with an increase in the wintertime ozone levels. The summertime drop is due to a drop in local $NO_X$ contributions while the wintertime increase is due to an increase in foreign $NO_X$ contributions (Figs 11b and e). It is noteworthy that the summertime drop in the local $NO_X$ contributions is larger than the overall drop in summertime PSO. For example, for the month of August, the observed PSO dropped by 3.86 ppb between the two periods. This drop is 4.63 ppb in the model. However, the drop in the local $NO_X$ contributions is larger (7.06 ppb) and there is also a drop in foreign $NO_X$ contribution (0.25 ppb). These combined decreases in local and foreign $NO_X$ contributions (7.31 ppb) are offset by increases in contributions from shipping $NO_X$ (1.65 ppb), natural $NO_X$ (0.96 ppb) and stratosphere (0.06 ppb) such that the overall drop in PSO in August is smaller. While the increase in shipping $NO_X$ contribution is consistent with an increase in the northern hemispheric shipping

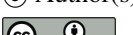



$NO_X$ emissions (Figure 2e), there is no significant increase in natural $NO_X$ between the two periods, which shows the increasing
ozone producing efficiency of natural $NO_X$ when overall $NO_X$ emissions are decreasing. In terms of VOCs, the summertime
drop is associated with a drop in local AVOC contributions and the wintertime increase is primarily due to increased share of
methane contribution as well as some foreign AVOC contribution.

Figure 12 presents the changes in the MDA8 $O_3$ seasonal cycle for Southern Europe. The model reproduces the seasonal cycles
for both the initial and final periods extremely well. We broadly see a flattening of the ozone seasonal cycle in this region
between the two periods, with the summertime peak coming down (due to reduced local $NO_X$ contribution partially offset by
increases in natural and ship $NO_X$ contributions) and wintertime levels rising due to increase in wintertime foreign $NO_X$
contributions, same as in Western Europe. From a VOC perspective, the summertime drop is associated with a decrease in
local AVOC contributions and a small drop in methane contributions. The wintertime increase is associated with an increase
in methane and foreign AVOC contributions but also stratospheric intrusion.

## 4. Conclusion, Limitations and Future Outlook:

In this study we explain the trends and changes in the seasonal cycle of surface ozone in Europe and North America through
the use of an ozone tagging system in a global chemical transport model for the period 2000-2018. While both regions have
experienced rapid reductions in locally-emitted ozone precursors in recent decades, we note that the Peak Season Ozone (PSO)
in both regions exceeds the WHO guidelines for the entire study period.

Our model is generally in good agreement with ground observations from rural stations in the newly-developed TOAR-II
database, allowing us to attribute the observed trends in terms of the changing contributions from local and foreign emission
sources of $NO_X$ and VOC. While anthropogenic NMVOC emissions contribute a relatively small fraction of the total PSO,
anthropogenic $NO_X$ emissions have a much stronger influence. The decreasing trend in $NO_X$ emissions in both North America
and Europe leads to a lower fraction of the PSO attributable to these local $NO_X$ emissions towards the recent years, however
the total modelled decrease in PSO in both regions is partially offset by increasing contributions from natural $NO_X$, foreign
anthropogenic $NO_X$, and international shipping.


While the increasing trend in ozone attributable to international shipping is consistent with increasing emissions from this
sector, the increasing contribution of natural $NO_X$ emissions we find in our study, especially during the summertime, is most
likely due to the increasing ozone productivity of these emissions. The decreases in local $NO_X$ emissions in both regions lead
to strong reductions in summertime ozone, but have a smaller effect in the springtime, when long-range transport of ozone
produced from foreign anthropogenic $NO_X$ emissions is more important. All regions show a modest increasing trend in the

foreign anthropogenic $NO_X$ contribution to the peak season ozone over the study period. Especially in the western sub-regions of Europe and North America, the foreign anthropogenic $NO_X$ contribution to PSO has become comparable in magnitude to the local $NO_X$ contribution.

Due to the nature of our ozone tagging system, we perform two separate source attributions, one for $NO_X$ emissions, and another for VOC emissions. When attributing ozone to VOC emissions, we note the strong contribution of BVOC emissions to the summertime peak ozone, which is clearly linked with the strong contribution of local anthropogenic $NO_X$ emissions to summertime ozone. In all of the sub-regions in our study except for the eastern parts of the United States, the contribution of methane to ozone is greater than that of BVOC. While global methane concentrations have risen from 1787 ppb to 1875 ppb

during our study period (an increase of about 5%), this has only led to a modest increasing trend in methane contributions to PSO in Europe. In all regions of the US except NW US, the methane contribution to PSO has slightly decreased over this time. This is consistent with the large reductions in local $NO_X$ emissions, leading to a lower efficiency of ozone production during methane oxidation over both regions.

While our ozone source attribution is capable of determining the contributions of different local and remote emission sources to the ozone as simulated in our model, it is only of limited usefulness in predicting the response of ozone levels to any future emission reductions. For such an assessment, it is necessary to perform model sensitivity studies reflecting the actual policy interventions aimed at reducing ozone. Studies like ours can however identify the major contributing emission sources. We have shown here that local anthropogenic $NO_X$ emissions still contribute significantly to PSO in both Europe and North

America. As an emission source which can be controlled with policy interventions, future policy should continue to target these emissions. Given the strong role of methane as an ozone precursor, as noted in this study and consistent with previous work, targeted reductions of methane along with other anthropogenic NMVOC can also be expected to contribute to the reductions in PSO needed to comply with the WHO guideline value.

**Author Contributions:**

TA and TB together designed the study. TA performed the model simulations. AN and AL provided support in model setup, source code changes and generating the tagged chemical mechanisms. TA performed all the analyses and produced visualizations with inputs from TB. CH and SG created and provided the gridded TOAR dataset for model evaluation. TA wrote the manuscript with inputs from TB. All co-authors provided their inputs to the discussion and the final manuscript.





**Acknowledgements:**

This work utilized high-performance computing resources made possible by funding from the Ministry of Science, Research and Culture of the State of Brandenburg (MWFK) and are operated by the IT Services and Operations unit of the Helmholtz Centre Potsdam. TA would like to thank Dr. Edward Chan at RIFS for his technical assistance in setting-up the model on the GFZ GLIC supercluster.


The following programs, organizations and persons contributed to the TOAR gridded dataset: **programmes:** Global Atmospheric Watch Program; Research and Development Programme of Ministry of Education Culture Sports Science and Technology Japan and Environmental Research and Technology Development Fund of Ministry of Environment Japan; **organizations:** AECOM; Aethon Energy; Air Quality Services; Air Resource Specialists, Inc; Akron Regional Air Pollution

Control Agency; Al Dept Of Env Mgt; Albuquerque Environmental Health Department, Air Quality Division; Anadarko, WY; Arizona Department Of Environmental Quality; Arkansas Department of Energy and Environment; Bay Area Air Quality Management District; Bayerisches Landesamt für Umwelt; Boulder City-County Health Department; CH2M Hill; Cabazon Band of Mission Indians, CA; California Air Resources Board; Canton City Health Department Air Pollution Control; Chattanooga-Hamilton County Air Pollution Control; Cherokee Nation, Oklahoma; Cherokee Nation/ITEC

(consortium); Choctaw Nation of Oklahoma; City of Huntsville, Div of Natural Resources; City of Jacksonville Environmental Quality Division; City of Toledo, Environmental Services Division; Clark County, NV DAQEM; Colorado Department of Public Health And Environment; Confederated Tribes of Umatilla Reservation of Oregon; Connecticut Department of Environmental Protection; Dames & Moore, Inc; Dayton Regional Air Pollution Control Agency; Delaware Dept Natural Resources and Environmental Control; Delaware Nation, OK; EBAS; Eastern Band Of Cherokee Indians of

North Carolina; Eastern Shoshone and Northern Arapaho Tribes of Wyoming; Enefit; Environment and Climate Change Canada / Government of Canada; Environmental Monitoring Company (Emc); European Environment Agency; Evansville Division Of Air Pollution Control; FDEP Ambient Monitoring Section; Florida Department of Environmental Protection (FDEP); Florida Dept of Environmental Protection, Central District; Florida Dept of Environmental Protection, Northeast District; Florida Dept of Environmental Protection, Northwest District; Florida Dept of Environmental Protection, South

District; Florida Dept of Environmental Protection, Southeast District; Florida Dept of Environmental Protection, Southwest District; Forest County Potawatomi Community, WI; Forsyth County Environmental Affairs Department; Georgia Air Protection Branch Ambient Monitoring Program; Gila River Indian Community of Gila River Indian Reservation, AZ; Goodyear Tire Obion County, TN; Great Basin Unified APCD; Hamilton County Department Of Environmental Services; Hessisches Landesamt für Umwelt und Geologie; Hillsborough County Environmental Protection Commission; Idaho

Department Of Health And Welfare-Environment Division; Illinois Environmental Protection Agency; Imperial County APCD; Indiana Depart Of Environ Management/Office Of Air Quality; Indianapolis Office of Environmental Services; Institut für Hygiene und Umwelt Hamburg; Inter-tribal Council of MIchigan, Inc.; Japan Agency for Marine-Earth Science



and Technology; Jefferson County, AL Department Of Health; Kansas City Health Department, Air Quality Section; Kansas Department Of Health And Environment; Kentucky Division For Air Quality; KNO$_X$ County Department Of Air Pollution

Control; La Posta Band of Dieguento Mission Indians of La Posta Indian Reservation, CA; Lake County Health Department Division Air Pollution Control; Landesamt für Natur, Umwelt und Verbraucherschutz Nordrhein-Westfalen; Landesamt für Umwelt, Naturschutz und Geologie; Landesamt für Umwelt, Wasserwirtschaft und Gewerbeaufsicht; Landesamt für Umweltschutz Sachsen-Anhalt; Landesanstalt für Umwelt, Messungen und Naturschutz Baden-Württemberg; Lane Regional Air Pollution Authority; Las Vegas Tribe of Paiute Indians of the Las Vegas Indian Colony, NV; Lincoln-Lancaster County

Health Department; MT Dept Of Environmental Quality, Air Quality Division; Mactec, Inc; Mahoning-Trumbull Air Pollution Control Agency; Maine D.E.P. Bureau Of Air Quality Control, Augusta; Manatee County Environmental Management Department; Maricopa County Air Quality ; Maryland Department of the Environment; Mass Dept Environmental Protection-Div Air Quality Control; Mecklenburg County Air Quality; Memphis-Shelby County Health Department; Meteorological Solutions, Inc.; Metropolitan Health Department/Nashville & Davidson County; Miami-Dade

County Department of Environmental Resources Management ; Michigan Dept Of Environment, Great Lakes, and Energy - Air Quality Division; Ministerium für Energiewende, Landwirtschaft, Umwelt und ländliche Räume; Ministerium für Umwelt und Verbraucherschutz Saarland; Minnesota Chippewa Tribe, MN (Fond du Lac Band); Minnesota Pollution Control Agency, Division Of Air Quality; Mississippi DEQ, Office Of Pollution; Missouri Laboratory Services Program; Mojave Desert AQMD; Monterey Bay Unified APCD; Morongo Band of Mission Indians, CA; National Institute for

Environmental Studies; National Park Service; Navajo Nation, AZ, NM, UT; Nevada Division Of Environmental Protection; New Hampshire Air Resources Agency; New Jersey State Department Of Environmental Protection; New Mexico Environment Department; New York State Department Of Environmental Conservation; Niedersächsiches Ministerium für Umwelt, Energie und Klimaschutz; North Carolina Dept Of Environmental Quality; North Coast Unified Air Quality Management District; North Dakota DEQ; Northern Sonoma County APCD; Northwestern Band of Shoshoni Nation of

Utah (Washakie); Ohio EPA Central Office; Ohio EPA, Central District Office; Ohio EPA, Northeast District Office; Ohio EPA, Northwest District Office; Ohio University, Athens, OH; Oklahoma Dept. Of Environmental Quality Air Quality Division; Olympic Air Pollution Control Authority; Omaha-Douglas County Health Department; Open Air Quality; Oregon Department Of Environmental Quality; Paiute-Shoshone Indians of Bishop Community of Bishop Colony, CA; Pala Band of Luiseno Mission Indians of Pala Reservation, CA; Passamaquoddy Tribe of Maine (Pleasant Point); Pennsylvania

Department Of Environmental Protection; Pennsylvania State University (Penn State); Penobscot Tribe of Maine; Picayune Rancheria of Chukchansi Indians of California; Pima County Department of Environmental Quality; Pinal County APCD; Pinellas County Department Of Environmental Management; Polk County Physical Planning; Ponca Tribe of Indians of Oklahoma; Portsmouth City Health Dept Division Air Pollution Control; Pueblo of Jemez, NM; Quapaw Tribe of Indians, OK; Quebecor; RR Donnley, IN; Sac and Fox Nation, OK; Sacramento County APCD; Salt River Pima-Maricopa Indian

Community of Salt River Reservation, AZ; San Diego County Air Pollution Control District; San Joaquin Valley Unified Air Pollution Control District; San Luis Obispo County APCD; Santa Barbara County APCD; Santa Rosa Indian Community of





Santa Rosa Rancheria, CA; Senat für Umwelt, Bau und Verkehr, Bremen; Senatsverwaltung für Stadtentwicklung und Umwelt; Senatsverwaltung für Umwelt, Verkehr und Klimaschutz Berlin; Servicio Meteorologico Nacional; Shasta County APCD; South Carolina Department Health And Environmental Control; South Coast Air Quality Management District;

South Dakota Department of Agriculture and Natural Resources; South East Texas Regional Planning Commission (SETRPC); Southern Ute Indian Tribe of Southern Ute Reservation, CO; St Louis County Health Department Air Pollution Control; St. Regis Mohawk Tribe, New York; State Of Louisiana; Swinomish Indians of Swinomish Reservation, WA; Sächsisches Landesamt für Umwelt, Landwirtschaft und Geologie; Table Mountain Rancheria; Taiwan Environmental Protection Agency; Tallgrass Energy Partners; Tata Chemicals; Tennessee Division Of Air Pollution Control; Tennessee

Valley Authority; Texas Commission On Environmental Quality; Thüringer Landesanstalt für Umwelt und Geologie; Torres-Martinez Cahuilla Indians, California; Toyota; Tracer Technologies; US Forest Service; USEPA - Clean Air Markets Division; Umweltbundesamt; United States Environmental Protection Agency; University Hygenic Laboratory (University of Iowa); Utah Department Of Environmental Quality; Ute Indian Tribe of Uintah & Ouray Reservation, UT; Vandenberg AFB; Ventura County APCD; Vermont Agency Of Environmental Conservation; Vigo County Division Of Air Pollution

Control; Virginia Department of Environmental Quality; WAUPACA Foundry; Wampanoag Tribe of Gay Head (Aquinnah) of Massachusetts; Warren Energy Services, LLC; Washington State Department Of Ecology; West Virginia Division of Air Quality; Weston Solutions, TX; Wisconsin Dept Of Natural Resources, Air Monitoring Section; Wyoming Air Quality Division, Dept Of Environmental Quality; Wyoming Bureau of Land Management; **persons**: Maria; Adela Holubova; Anne-Cathrine Nilsen; Aude Bourin; Christine F Braban; Christoph Hueglin; Christopher Conolly; Chrysanthos Savvides;

Erik Andresen; Erzsebet Gyarmatine Meszaros; Gabriela Vitkova; Gerardo Carbajal Benitez; Gerardo Carbajal Benítez; Hiroshi Tanimoto; Indriksone Iveta; Iveta Indriksone; Jan Silhavy; Jaroslav Pekarek; Jasmina Knezevic; Juan Martinez; Karin Sjoberg; Karin Sjöberg; Karin Sjøberg; Keith Vincent; Lino Fabian Condori; Magdalena Bogucka; Maj-Britt Larka; Marcin Syrzycki; Maria Barlasina; Marijana Murovec; Mateja Gjere; Milan Vana; Ming-Tung Chuang; Monistrol Jose Antonio Fernandez; Muinasmaa Urmas; Murovec Marijana; Nikolova Yana; Robert Gehrig; Roman Prokes; Rune Keller;

Stefan Reimann; Truuts Toivo; Ursul Gina; Usin Eve; Veronika Minarikova; Wenche Aas; Willis Paul; Yugo Kanaya; Zdzislaw Przadka (2024). Ozone data obtained from TOAR Database for rural stations between 2000 and 2018.

**Competing Interests**

TB is a member of the editorial board of Atmospheric Chemistry and Physics.


**Data Availability**

Please contact tabish.ansari@rifs-potsdam.de for availing the model output of our simulations.

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

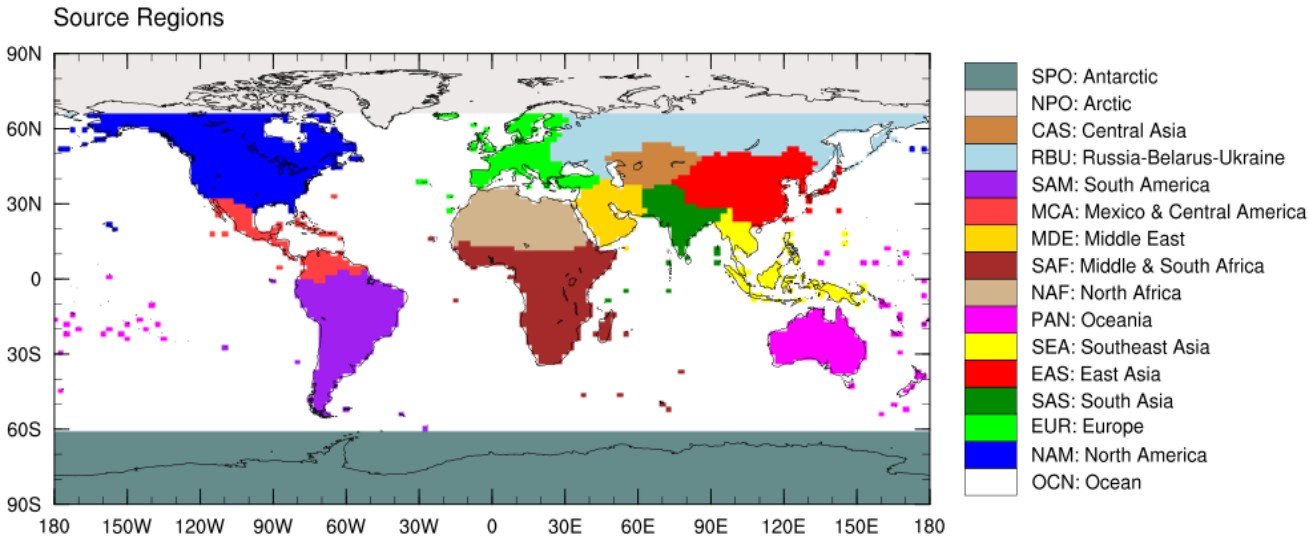


**Figure 1: HTAP Tier 1 regions which form the basis for source regions NO$_X$ and VOC tagging. See Table 1 for more details on tagged regions.**





**Figure 2: Time-series of NO$_X$- (left panels) and VOC-emissions (right panels) for North America (a, b), and Europe (c, d) source regions along with Northern Hemispheric totals (e, f) and global totals of lightning NO$_X$ and background CH$_4$ concentrations over the study period.**





**Figure 3: HTAP Tier 2 receptor regions.**





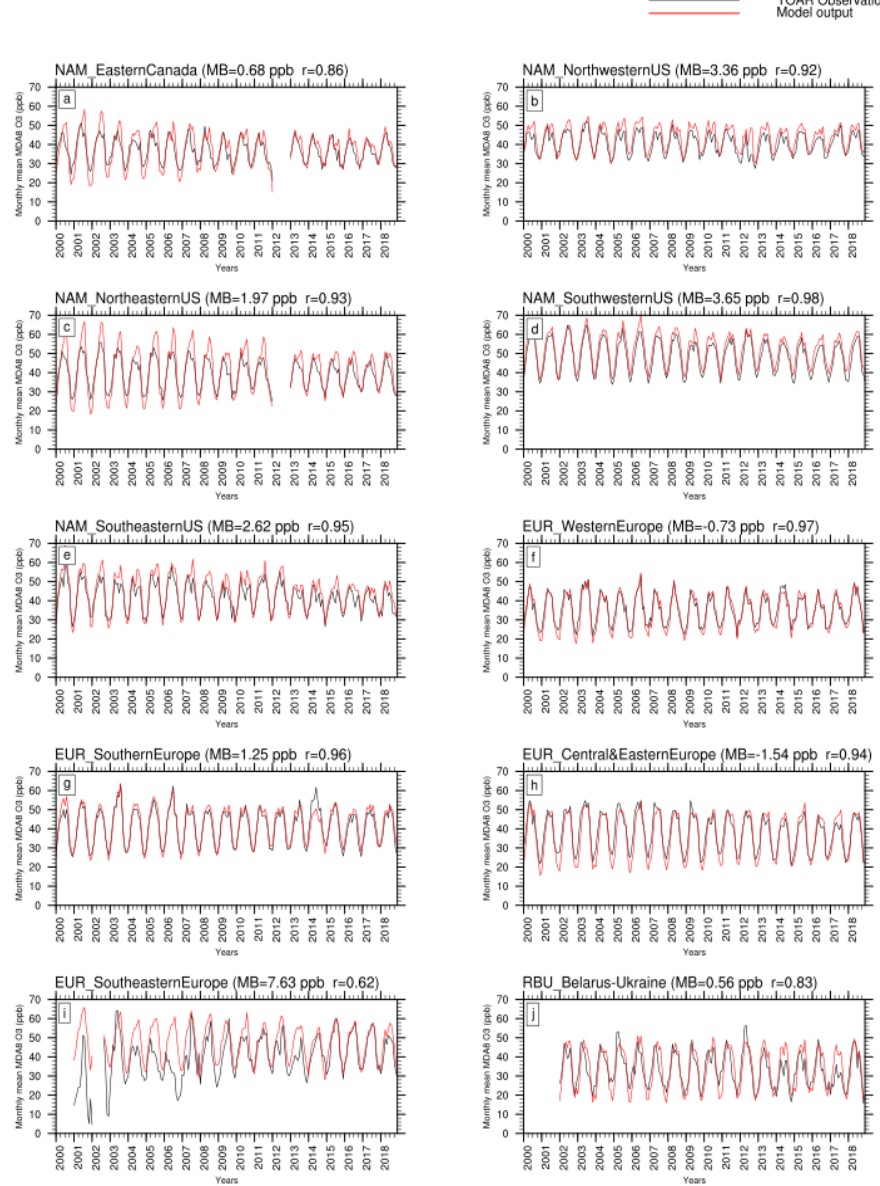

**Figure 4: Time series of observed versus simulated monthly mean MDA8 O₃ along with mean bias and correlation coefficients for various receptor regions. Only rural stations data were utilized from the TOAR database and model output was fetched only for those grid cells where observations were available.**





**Figure 5: Time-series of observed and model-derived Peak Season Ozone for various receptor regions in North America for 2000-2018 and its source contributions in terms of NOₓ sources (left panels) and VOC sources (right panels). Model output was sampled from TOAR-valid grid cells only.**




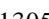

**Figure 6: Month-centered average MDA8 O₃ over the 2000-2018 period for various receptor regions in North America and its source contributions in terms of NOₓ sources (left panels) and VOC sources (right panels). Model output was sampled from TOAR-valid grid cells only.**




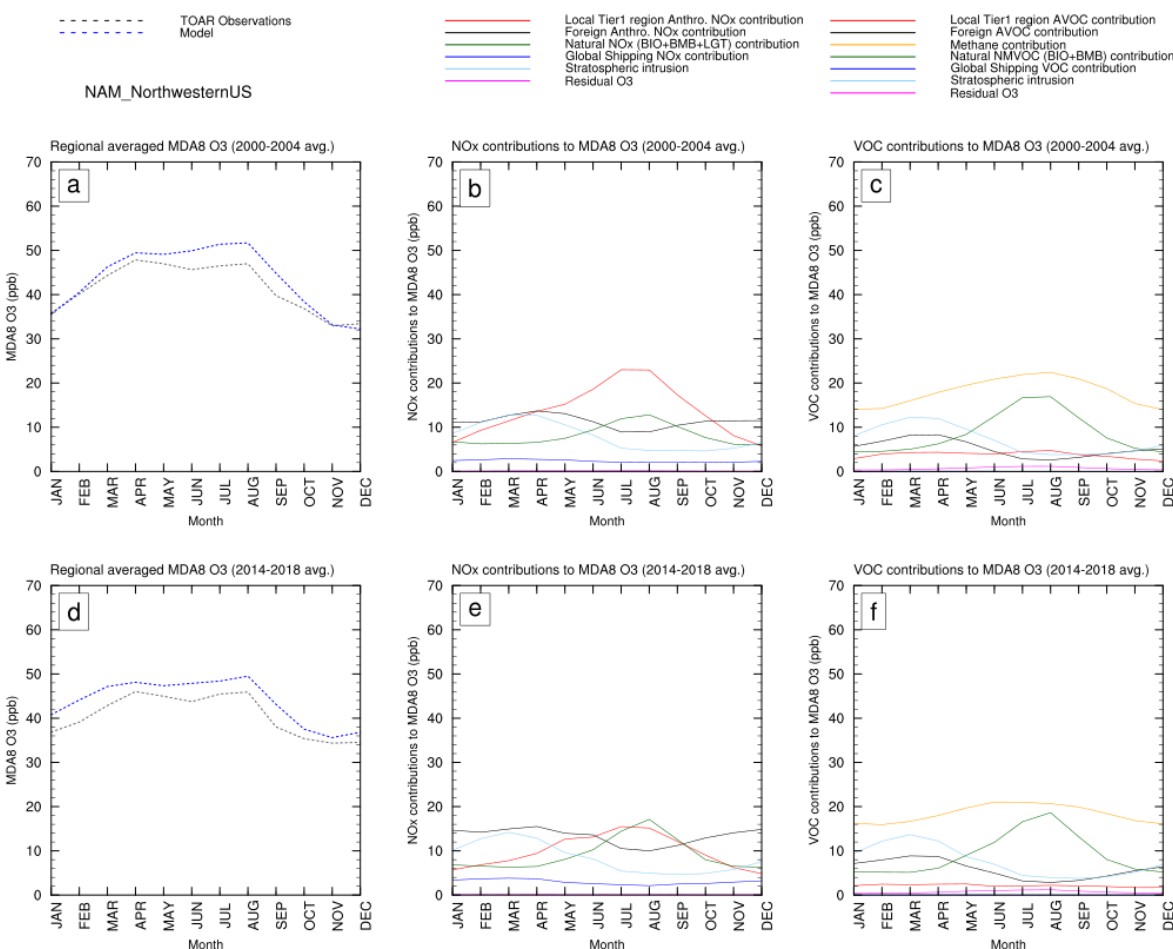

**Figure 7: 5-year average MDA8 O₃ seasonal cycles for Northwestern US for 2000-2004 (a) and 2014-2018 (b) along with their NOₓ (b,e) and VOC contributions (c,f).**





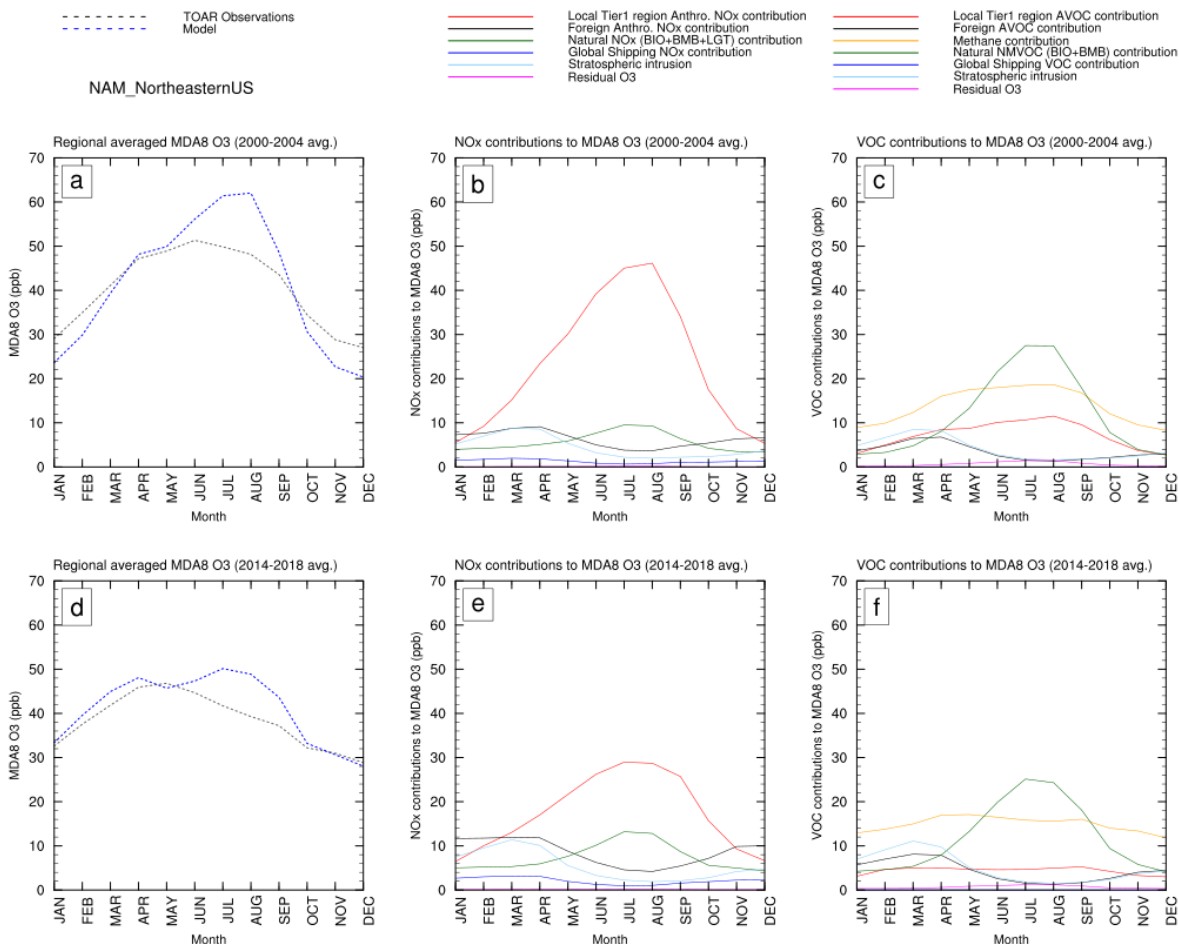

**Figure 8: 5-year average MDA8 O₃ seasonal cycles for Northeastern US for 2000-2004 (a) and 2014-2018 (b) along with their NOₓ (b,e) and VOC contributions (c,f).**





**Figure 9: Time-series of observed and model-derived Peak Season Ozone for various receptor regions in Europe for 2000-2018 and its source contributions in terms of NO$_X$ sources (left panels) and VOC sources (right panels). Model output was sampled from TOAR-valid grid cells only.**






**Figure 10: Month-centered average MDA8 O₃ over the 2000-2018 period for various receptor regions in Europe and its source contributions in terms of NOₓ sources (left panels) and VOC sources (right panels). Model output was sampled from TOAR-valid grid cells only.**




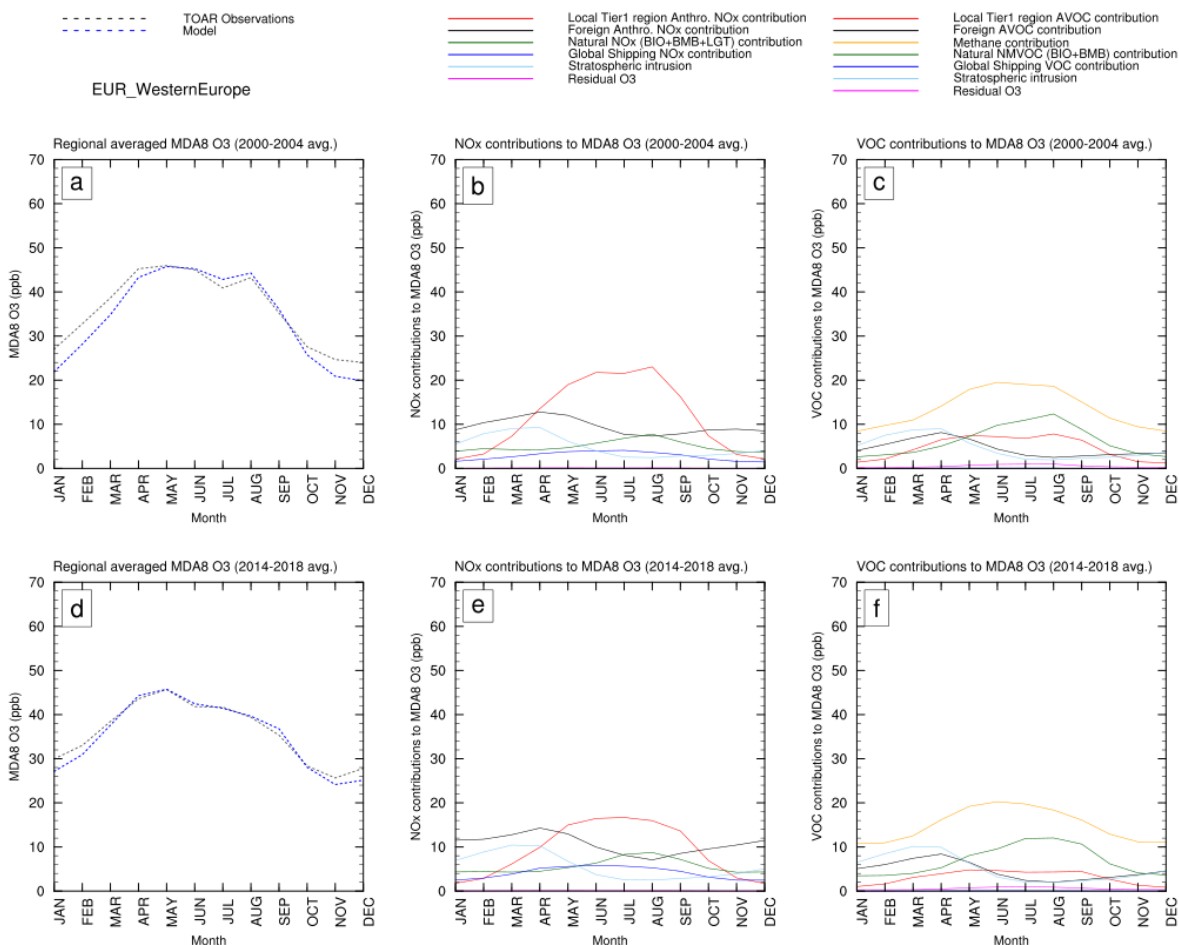

**Figure 11: 5-year average MDA8 O₃ seasonal cycles for Western Europe for 2000-2004 (a) and 2014-2018 (b) along with their NOₓ (b,e) and VOC contributions (c,f).**






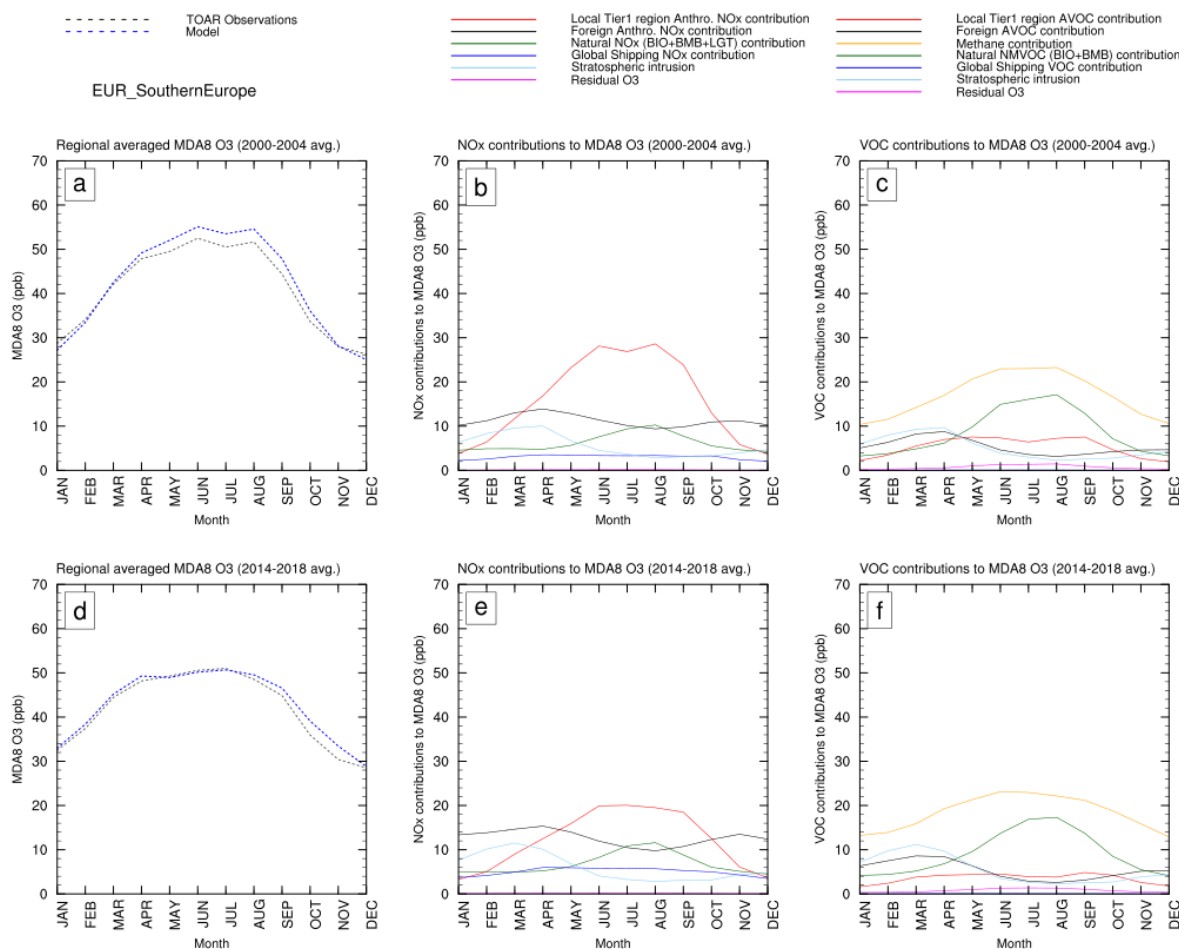

**Figure 12: 5-year average MDA8 O₃ seasonal cycles for Southern Europe for 2000-2004 (a) and 2014-2018 (b) along with their NOₓ (b,e) and VOC contributions (c,f).**








**Table 1: Various emission tags for NO<sub>X</sub>- and VOC-tagged simulations. The geographic definition of the land-based tags corresponds to the HTAP tier 1 regions as shown in Figure 1. For NO$_X$-tagging, "Rest of the World" corresponds to the tier 1 regions of South America, Oceania, and Middle & Southern Africa combined. For VOC-tagging, the regions: Arctic, Central Asia, Mexico & Central America, North Africa, and Southeast Asia were also combined into the "Rest of the World". The regional oceanic tags are only applicable for NO$_X$-tagging and their geographic definitions can be seen in Figure 3. For VOC-tagging we use a single oceanic tag representing NMVOCs from shipping and natural DMS emissions. Lightning tag is only applicable for NO$_X$-tagging.**

| Regional land-based Tags | Regional oceanic tags | Global sector/process-based tags |
|---|---|---|
| Arctic | North Atlantic | Aircraft |
| Central Asia | Eastern North Atlantic | Biogenic |
| East Asia | North American East-Coastal zone | Biomass Burning |
| Europe | North American West-Coastal zone | Lightning |
| Mexico & Central America | North Pacific | Stratosphere |
| Middle East | Baltic and North Seas | |
| North Africa | Hudson Bay | |
| North America | Indian Ocean | |
| Russia-Belarus-Ukraine | Mediterranean, Black, and Caspian Seas | |
| South Asia | Southern Hemisphere Oceans | |
| Southeast Asia | | |
| Rest of the World | | |