# Peer review of "Explaining trends and changing seasonal cycles of surface ozone in North America and Europe over the 2000-2018 period: A global modelling study with NOx and VOC tagging"

_EGUsphere, 2024_

## Author Response (AR1)

**Responses to reviewers**

In this document we have presented a point-by-point response to the reviewers' comments. The original comments by the reviewer are shown in black while our responses are shown in blue.

**Reviewer 1:**

This manuscript attributes trends and seasonal cycles in ozone metrics (PSO and MDA8) across Europe and North America to NOx and VOC sources. Some key findings include that declining local emissions of NOx contribute to decreased O3 production in summertime but increasing import of foreign precursors leads to increasing winter and springtime O3; that natural VOC and local anthropogenic NOx are often linked in their formation of O3; and that foreign NOx is now of similar magnitude to local NOx in terms of contribution to O3 formation in several locations.

These findings are achieved using a model in which the precursor molecules are tagged by source and location. This is a considerable amount of work, which has yielded novel results. The conclusions are of interest to the community and the work fits well into the scope of the journal and special issue.

We thank the reviewer for their positive comment - it was indeed a lot of work but we are glad that it has produced some novel results that are of interest to the community.

Although long, the text is clearly written and the conclusions are well supported by the data. I would recommend this manuscript for publication with only a few amendments/suggestions.

**General comments**

The methodology is very clear on several aspects such as the emissions inventories but no detail on the chemical model itself – how many species and reactions, how are VOC and oxidation products treated / lumped? The text refers to MOZART but it's never explicitly introduced that you are using this scheme.

We have now added a few sentences on the chemical mechanism and treatment of VOCs and also pointed the reader to the relevant literature for further details on the MOZART chemical mechanism: L183-190:

"The gas-phase chemical mechanism employed in this study is based on the Model for Ozone and Related chemical Tracers, version 4 (MOZART-4) (Emmons et al., 2010) which includes detailed  $O_x$ - $NO_x$ - $HO_x$ -CO- $CH_4$  chemistry, along with the oxidation schemes for a range of non-methane volatile organic compounds (NMVOCs). Specifically, MOZART-4 treats 85 gas-phase species involved in 39 photolytic and 157 gas-phase reactions. NMVOCs are represented using a lumped species approach, where, for example, alkanes larger than ethane are lumped as a single species (e.g., BIGALK for C4+ alkanes), and alkenes larger than ethene are lumped (e.g., BIGENE), with specific treatments for aromatics, isoprene, and terpenes. The oxidation products of these lumped

and explicit VOCs are also tracked. Further details on the MOZART-4 chemical mechanism, including the full list of species and reactions, can be found in Emmons et al. (2010)"

The structure of the manuscript is ok as is but perhaps could be improved. Some of the results sections begin with several large paragraphs of introduction before any results are given. Some of this text may be better placed in, or is perhaps a repeat of, the methods sections. Similarly, descriptions of figure 2 would work in the results rather than the methods.

We considered including the emissions trends in the results section but felt that it was more appropriate to cover all descriptive aspects of model input, including regional and sectoral emission trends, in the model description section. Since emissions are not an output from the model runs, we do not treat them as part of results. However, we do refer back to these emission trends in relation to modelled ozone in the results section when necessary. We have now included new spatial maps of anthropogenic NOx emissions for North America and Europe in the result section (Figures 5 and 11).

The individual regions are described independently but never compared. Are there any key differences between regions that could be highlighted in the conclusion? If not, do all the regions need to be described in such detail?

Thank you for your comment. This point has also been raised by reviewer 2 and we have substantially rewritten sections 3.2 and 3.3 with more emphasis on intercomparison of same source contributions across all receptor regions for a given continent. We have also made the discussion less qualitative and redundant.

The difference in emissions reductions between western and eastern Europe is highlighted as a reason why PSO trends may differ regionally but this hypothesis is never confirmed. There don't seem to be large differences in PSO changes over time between the regions. Is that correct? If so, can you suggest why? A follow-up on this introductory point is needed somewhere.

We have now included new spatial maps of the peak season anthropogenic NOx emissions, PSO, and local NOx contribution to PSO for the initial (2000) and final (2018) year for both North America and Europe. See Figures 5 and 11. Figure 11a and 11d clearly show the relatively small change in anthropogenic NOx emissions in Central & Eastern Europe over the 19-year period as compared to Western Europe. Section 3.3 has been rewritten now and section 3.3.1 discusses this issue more specifically in lines L1109-1118

"To understand the geographical backdrop of PSO changes, Figure 11 presents a spatial map of local anthropogenic NOx emissions (panels a, d), total PSO (panels b, e), and the modeled contribution of local anthropogenic NOx to PSO (panels c, f) for the initial (2000) and final year (2018). In 2000 (Figure 11a), prominent NOx emission hotspots were evident (e.g., Benelux, Germany, Po Valley), parts of the UK, and major urban agglomerations across the continent. By 2018 (Figure 11d), substantial emission reductions occurred, particularly in Western and Central Europe. However, this decline is not obviously reflected in the spatial patterns of total PSO (Figure 11b, e), which generally decreased in the southern regions but not in northern regions, especially over areas with the largest emission cuts, as also seen in bias-corrected PSO maps by Becker et al. (2023). The direct contribution of local anthropogenic NOx to PSO (Figure 11c, f) mirrors these emission reductions

more closely, with clear reductions from 2000 to 2018. This suggests the role of other contributions in offsetting the expected decline in PSO, especially in northern European regions."

The conclusion could be developed somewhat, for example by comparing the different regions and Europe vs the US (the US emissions reductions seem more successful at reducing O3?) as stated above. There could also be greater discussion on some of the uncertainties – are emissions inventories the greatest source of uncertainty in your study? I would finally also be interested in your thoughts on how this novel tagging approach could be used in future studies – could it play a role in bias correction and identification?

We have now substantially updated the Conclusion, Limitations, and Future Outlook section. It now covers some comparative aspects of North American and European changes in precursor emissions and observed ozone, a couple of statements on the potential sources of biases (model resolution, emission inventory, treatment of ship plumes, deposition schemes), and future directions for the use of this tagging approach.

Data should be made publicly available according to Copernicus guidelines

We will make the data produced in this study publicly available before the final publication of the manuscript.

Units and trend analysis should follow the TOAR special issue guidelines. Just check if any of those are relevant to this work.

We have updated our trend analysis to include p-values and 95% confidence intervals based on the TOAR statistical guidelines.

Specific comments:

L26: A concluding sentence could be useful here.

We have now added a concluding sentence "Our results highlight the evolving drivers of surface ozone and emphasize the need for coordinated global strategies that consider both regional emission trends and long-range pollutant transport."

L45: In case you would like the more recent publication: Cheesman, Alexander W., et al. "Reduced productivity and carbon drawdown of tropical forests from ground-level ozone exposure." *Nature Geoscience* (2024): 1-5.

We have now included this citation.

L56: It is worth being clear that the O3 precursors can also be transported, since this is the key part of the study.

Thank you for this suggestion. We have added a new sentence in L70-72: "Moreover, some ozone precursors (e.g., CO and less reactive NMVOCs) also possess atmospheric lifetimes sufficient for

intercontinental transport, subsequently contributing to ozone formation in downwind regions far from their original emission sources."

L210-214: This is slightly hard to follow and explained much better in Table1 and the associated table caption. I would suggest you direct readers to this table earlier. I would also suggest on L213 to replace 'many zones' with the specific number of zones.

We have now directed the reader to Table 1 earlier, and have modified L252-253 to "The ocean is also divided into multiple zones and tagged separately (see Figure S12)." Figure S12 is a new figure which shows these oceanic zones.

L230: if you plan to refer to anthropogenic VOCs as AVOCs this could be introduced earlier and then used consistently throughout (such as in the figure labels)

We have now replaced subsequent instances of "anthropogenic NMVOCs" with "AVOC"

L226-250: This reads as results rather than methods to me. Certain aspects such as AVOCs showing an increasing trend from 2013 and global lightning NOx showing a decreasing trend are interpretations of the figure. Consider moving this to the results or giving its own subheading in the methods section.

We decided not to move this description of emissions since it is not a model output. However, we have revisited emissions distribution vis-a-vis PSO distribution in the results section (3.2.1 and 3.3.1) through the addition of new figures (Figures 5 and 11).

Section 2.2 could be condensed. The definition of MDA8 is essential but the wallclock time in your specific setup could be left out.

We decided to retain this piece of technical information because we feel it allows other researchers to compare the computational advantage of using a tagging-based system (such as TOAST 1.0) against, say, a set of multiple perturbation simulations which may be used to derive similar source contributions. We believe documenting such computational requirements is scientifically valuable and particularly helpful for new researchers entering the field who often struggle to find such information in published literature.

In figure 3, is it possible to just show the source regions being considered? It is a bit unclear which 5 regions of N America and Canada are being selected at this stage.

Thank you for this suggestion. We have now updated Figure 3 such that only the receptor regions discussed in this study are shown. The map has been cropped out to enlarge the discussed regions and the colour scheme has also been updated to emphasize the different sub-regions within a continent.

L305: Inhomogeneous measurements combined with changing numbers of stations over time can lead to errors in mean and trend identification. An acknowledgement of the uncertainty in

the observational trends would be helpful here. Christoph Frei has done a lot of work on this for temperature and precipitation fields.

We have now added the following sentence with an appropriate reference L360-361: "We note that sparse spatiotemporal sampling can introduce uncertainty in identifying true long-term trends of ozone and refer the reader to a technical note by Chang et al. (2024) for more details on this issue."

L382: I agree the simulation of MDA8 against rural stations is very nice. I am not sure what the satisfactory performance across different world regions refers to.

We have now made this evaluation more holistic by also adding annual averaged correlations (to remove the influence of O3 seasonal cycle in driving up the r-values) and mean absolute bias (MAB) to reduce the compensatory effects of high and low biases. These numbers compare quite favorably to other studies (also added now).

L387: What is a 19 year month centered average MDA8 O3 broken down into 5 years?

Thanks for pointing this out. We have now made this sentence clearer: "We also present the first five year (2000-2004) and last five year (2014-2018) month-centered average MDA8 O3 seasonal cycle and explain the shifts in terms of tagged contributions for all receptor regions during these periods."

L406: A reminder of the PSO guideline could be useful here.

We have now included the WHO PSO guideline values (31 ppb) here.

L444-446: Is it possible for the PSO to be 'single handedly' linked with local AVOC and then linked to declining NOx emissions in the following sentence?

We have removed "almost single handedly" from this sentence.

L525: I would pedantically argue that all ozone is equally destroyed by water, but long-range transport ozone is more likely to encounter it.

This section is entirely rewritten but we have made this point clearer in the new text in lines L881-885: "The summertime dip in foreign NOx contribution (also seen in other sub-regions) is likely due to shorter lifetime of ozone at higher temperatures, which is associated with increased water vapor content in the atmosphere (Stevenson et al., 2006). Water vapor promotes ozone loss via photochemical pathways involving  $HO_x$  radicals, and transported ozone is more likely to be destroyed under moist conditions (Real et al., 2007). Consequently, the efficiency of long-range ozone transport decreases in summer"

L558: perhaps refer back to the figures here

This section is rewritten.

L738: Can you say why local NOx and BVOCs are so interlinked? This is an interesting finding.

We have added more text to emphasize this finding: L1410-1421

"Due to the nature of our ozone tagging system, we perform two separate source attributions, one for NOX emissions, and another for VOC emissions. When attributing ozone to VOC emissions, we note the strong contribution of BVOC emissions to the summertime peak ozone, which is clearly linked with the strong contribution of local anthropogenic NOX emissions to summertime ozone. The co-variability of these two sources is also apparent in the PSO time series for all regions and emphasizes the interaction of anthropogenic NOx with BVOC in rural and background regions. This is an emerging finding made possible due to our dual-tagging approach; a relatively recent regional modelling study (Lupascu et al., 2022) focusing on two high ozone episodes in Germany that also utilized the TOAST1.0 system also noted the interaction of local anthropogenic NOx and BVOC in driving ozone peaks.and emphasizes the interaction of anthropogenic NOx with BVOC in rural and background regions. This finding highlights that, at least for rural and background regions, the interaction of anthropogenic NOx with BVOC exceeds its interaction with AVOC which might be contained within the urban centres. It is noteworthy that BVOC emissions also either match or exceed AVOC emissions in North America and Europe during the peak season."

Technical points:

L22: 'productivity' could be replaced with 'O3 production' or similar for clarity.

We have now made this amendment.

Figure 7 b/d is mislabelled in the figure caption. Same for other figures of this layout.

Thanks for pointing this out. We have now corrected this in the captions.

Figures 2 and 3 are sometimes referred to incorrectly in the text (e.g. Fig 2 written instead of Fig. 3 and vice versa)

| We have now corrected this issue. |
|-----------------------------------|
|                                   |

**Reviewer 2**

This paper presents a global modelling study with an innovative dual tagging analysis; the paper's focus is on surface ozone over North America and Europe, particularly with regard to the causes of long-term changes and the seasonal cycle and its changes. In my view, the work appears to employ a state-of-the art modeling system and addresses an interesting scientific issue.

We thank the reviewer for their positive comment on the scientific relevance of our research question and the innovative nature of our methodology.

However, ACP aims to publish studies with important implications for our understanding of the state and behavior of the atmosphere; I believe that the paper at present fails to advance this aim. Thus, I recommend that this submission be rejected and the paper returned to the authors with encouragement to resubmit if the authors can address 3 critical issues:

Significant questions remain both regarding the evaluation of the modeling system and the possible corruption of observational data upon which the model evaluation and analysis are based.

A more quantitative approach to analysis would improve the paper, both with regard to discussion of present results and discussion of comparisons and contrasts with previously published analyses.

A clear summary is required of what new understanding of the atmosphere has emerged from this study, including the added value of the dual tagging approach, and when and for what issues that approach is required in such modeling studies.

We thank the reviewer for seriously engaging with our work and pointing out these caveats which we have now addressed in the comprehensively revised manuscript.

More detailed discussions of these issues follow. In addition, several related and unrelated, major and minor issues are discussed that may be of use to the authors for their revision.

**Critical issues:**

Section 3.1 is devoted to evaluation of the CAM-Chem model used in the chemical-transport simulations. In my view a much more robust evaluation is required as outlined in the following paragraphs; I believe that, in general, such evaluation is necessary before model simulations can be relied upon to provide robust results.

One aim of the authors is to explain the seasonal cycles and their changes observed in surface ozone; thus the model evaluation should move beyond an overall statistical comparison of monthly mean concentrations between model and observations; it is necessary to specifically and quantitatively evaluate how well the model reproduces the phenomena of interest, in this case the seasonal cycles and their changes.

We have now adopted a more quantitative approach to model evaluation with a particular focus on the reproduction of seasonal cycles (of individual years as well as multi-year averages) over various receptor regions as captured by fourier parameters (amplitude and phase angles of constituent harmonics).

The three paragraphs on lines 339-372 discuss aspects of such an evaluation, but only qualitatively; this evaluation should be placed on a quantitative basis. Of concern are the findings in previously published intercomparisons of model simulations with observed seasonal cycles of ozone. Parrish et al. (2016) discuss such intercomparisons for marine boundary layer (MBL) sites and in the overlying free troposphere above one site. Three chemistry climate models, including a version of the CAM-Chem model used in the present study, approximately reproduced many features of the measured seasonal cycles within the MBL, with some notable quantitative disagreements, but gave divergent results that do not agree with measurements above the MBL. Bowman et al. (2022) discuss a similar intercomparison that considers both the seasonal cycles and their systematic changes at northern midlatitude baseline locations. The available observational data were compared with simulations by 6 Coupled Model Intercomparison Project Phase 6 (CMIP6) Earth system models, again including a version of the CAM-Chem model. Substantial differences were found between the different model simulations and between the simulations and the observations. To my knowledge, the model disagreements revealed in these intercomparisons have not been addressed in later model development. A quantitative evaluation such as presented in these two papers should be included in the present manuscript.

We have now adopted the approach of Parrish et al. 2016 and Bowman et al. 2022 and broken down the modelled and observed seasonal cycles over all the receptor regions considered in our study into two harmonics using a fourier transform, thus yielding values such as the detrended y-intercept and the amplitudes and phases of the two harmonics which allow for a more quantitative comparison of various features of the observed and modelled seasonal cycles. We applied this approach to the seasonal cycles for each individual year (2000 to 2018) as well as time-averaged seasonal cycles over the first five (2000–2004) and last five (2014–2018) years and also the full 19-year (2000–2018) averaged seasonal cycles. These parameters are listed in tables S1-S9 for different receptor regions. We have discussed the model performance in terms of these parameters both in the model evaluation section (section 3.1) as well as in the analysis sections (3.1, 3.2.2, 3.2.3, 3.3.2, 3.3.3)

The authors primarily rely on an overall statistical analysis of the agreement between model simulations and observations of monthly mean ozone in specific regions. Their final conclusion (line 374) is: "Overall, we obtain very good model-observations agreement, with low biases and high correlations, ...." Three issues must be addressed regarding this conclusion. First, this evaluation is limited to comparison between model results and observations of time series of MDA8 ozone that are highly averaged, both temporally (monthly) and spatially (first over model grid cells and then over receptor regions); it should be discussed if this averaging of model result is appropriate in the context of the model results that the authors employ in the following discussion of results.

Since all our subsequent analyses and conclusions depend on Peak Season Ozone values averaged over the defined receptor regions, which in turn depend on monthly mean MDA8 O3 values and not on any high frequency extremes, evaluation of monthly mean MDA8 O3 over the different receptor regions is adequate and appropriate in this context. We have now added a line in our evaluation section highlighting this point.

L493-496: "We note that our model evaluation is based on model results and observations of time series of MDA8 O3 that are averaged, both temporally (monthly) and spatially (first over model grid cells and then over receptor regions) but such an evaluation is valid because all our further analyses and conclusions depend on the same spatial and temporal scales."

Second, for the 10 receptor regions discussed in the paper (shown in Figure 4) and 5 additional receptor regions (shown in Figure S1) the average of the mean biases is indeed small (1.75 ppb), but more context is required for these mean biases. The regional mean biases range from -9.57 to 8.84 ppb, giving an overall regional mean absolute bias twice as large (3.50 ppb) as the overall mean bias. These statistics provide some evaluation of model performance in simulating average ozone concentrations, but the paper aims to quantify ozone "trends and changing seasonal cycles". It is clear from examination of Figures 4 and S1 that there are significantly larger deviations between the observations and model results for the individual monthly means than are reflected in the regional mean biases. The authors should give a more detailed view of the time series comparison including discussion of an additional statistic (I suggest mean absolute difference), which would more relevantly quantify the differences of monthly means between model results and the observations – it is these differences that are of most importance with regard to comparison of seasonal cycles and their changes.

We have now also calculated the Mean Absolute Bias (MAB) which are printed over each panel in Figures 4 and S1. We have also discussed the MAB in the text in section 3.1.

Second, the high correlations the authors cite (generally r > 0.9) are a) expected and b) not universal (3 of 15 r values are 0.62 or smaller); large r values are expected simply because the variability in both model results and observations is dominated by large seasonal cycles that are approximately in phase; if annual averages were compared, the correlations would be much lower. Examination of the figures strongly suggests that the smaller r values very likely indicate spurious observational data; the origin and influence of these spurious data must be assessed as discussed further below.

We have now also included correlation coefficients between annual averaged observed versus modelled MDA8 O3 (shown in brackets in figures 4 and S1). These are indeed smaller than

r-values between observed vs modelled monthly mean MDA8 O3 because they essentially denote interannual variability between modelled and observed ozone which to a large extent depends on year-to-year meteorological changes. These are also the same as correlation coefficients between the y0 which are further presented in Figures S3 and S4. Section 3.1 has been extensively rewritten with a focus on the model's skill in reproducing the observed Fourier parameters (y0, A1, A2,  $\varphi$ 1 and  $\varphi$ 2).

Third, from a skeptical viewpoint, we must be mindful of a subtle issue associated with model-observation comparisons such as the authors present in Section 3.1 (and that are also utilized in many such comparisons in the published literature). Chemical transport models do not treat all relevant processes from first principles of physics and chemistry; rather there are a great many parameterizations embedded within the computer code of the models. Over past decades those parameterizations were developed and tuned so that the models reproduce mean observed ozone concentrations as closely as possible. Consequently, attempting to evaluate the performance of models simply from comparison of means of observations with model results involves a degree of circular reasoning – the models were developed to agree with observations, so such agreement cannot be taken as independent confirmation that models perform properly for the correct reasons.

We acknowledge the general concern that parameterizations within models are often tuned over time to improve agreement with observed mean concentrations, which could introduce a degree of circularity in model evaluation. However, we believe several aspects of our specific evaluation mitigate this concern and demonstrate the robustness of our findings for CAM4-Chem using TOAR-II gridded observations: First, our evaluation focuses on the Maximum Daily 8-hour Average (MDA8) O3, which is a health-relevant peak metric. Reproducing MDA8 O3 accurately requires the model to capture not only mean ozone levels but also the diurnal cycle of ozone, which is influenced by complex interactions between precursor emissions, photochemistry, boundary layer dynamics, and short-term meteorological variability. This is a more stringent test than matching simpler mean ozone concentrations, which are often the primary target for broader model tuning efforts. It is less likely that model parameterizations were specifically tuned to universally optimize MDA8 O3 across diverse global regions.

Second, we use the recently released TOAR-II gridded surface ozone product for evaluation. While CAM4-Chem, like other global models, has undergone development and validation against various observational datasets over its lifetime, it is unlikely that its parameterizations were specifically and extensively tuned to reproduce the regional monthly mean MDA8 O3 values derived from this particular TOAR-II gridded dataset. The TOAR-II dataset is based on a comprehensive collection of global observations, processed and gridded, offering a relatively independent benchmark for models developed prior to or in parallel with its creation. The version of CAM4-Chem used in this study has a development history that would have drawn on a broader range of observational inputs, likely not focused explicitly on optimizing for the TOAR-II MDA8 O3 metric.

Third, our analysis demonstrates high correlations and low biases for regionally averaged monthly mean MDA8 O3 values across all evaluated receptor regions. Achieving such consistent performance across multiple diverse regions, each with unique emission profiles, chemical regimes, and meteorological conditions, suggests that the model captures key underlying processes governing ozone production, transport, and loss with reasonable fidelity. This goes

beyond simply matching a global mean and indicates skill in representing regional-scale phenomena and their seasonal variability.

Finally, the treatment of ship emissions and possibly the MBL structure constitute one apparent shortcoming in the modeling methodology that must be addressed; I believe this is a wide-spread shortcoming of chemical transport models, and has significant impact on this study's results. The model treatment of ship emissions is not discussed in detail in Section 2, but the authors do quantify ozone produced from this emission source. Unless the model includes some sort of plume-in-grid treatment for ship plumes, it must be expected that the influence of ships is significantly exaggerated (e.g., Kasibhatla et al., 2000). I am aware of only a single observational study that provides a detailed characterization of ship plume evolution (Chen et al., 2005). This study shows that the photochemical perturbation of the marine boundary layer (MBL) due to a ship plume is largely limited to the first few hours of plume evolution, while the plume is still narrowly confined (FWHM  $\sim$  3 km). Thus, an effective model resolution < 1 km is required to accurately treat ship emissions. However, I suspect that the model utilized in this study (as well as many other studies) immediately distributes the ship emissions throughout the  $1.9^{\circ}\times 2.5^{\circ}$  model cell; this is expected to lead to a large overestimate of ozone production from ship emissions.

We thank the reviewer for this insightful comment regarding the treatment of ship emissions and their impact on ozone production. The reviewer is correct that our CAM4-Chem configuration, consistent with many global chemical transport models, does not employ a plume-in-grid parameterization for ship emissions. Instead, these emissions are instantaneously diluted within the 1.9° x 2.5° model grid cell. We acknowledge that this approach can lead to an overestimation of ozone production efficiency from ship NOx compared to the more localized, high-NOx conditions within a concentrated young plume where titration effects and NOx self-reactions can be more dominant. The immediate dilution into a larger volume with potentially lower background NOx can artificially enhance the per-molecule ozone production from the emitted ship NOx. Therefore, while our study quantifies the contribution of ship emissions as resolved by the model, we recognize this as a potential uncertainty, and the true ship NOx contribution might be somewhat lower than simulated. Addressing this accurately would indeed require sub-grid scale treatments, which are computationally intensive and not yet standard in all global modeling frameworks. We have added new text to our discussion of limitations to reflect this point:

L472-485: "We also evaluate the model in the context of potential overestimation of ozone production from ship plumes. This is because in our modelling setup, ship NOx emissions are instantaneously diluted within the  $1.9^{\circ} \times 2.5^{\circ}$  model grid cell which can lead to an overestimation of ozone production efficiency from ship NOx. In the real world, the more localized, high-NOx conditions within a concentrated young plume, the titration effects and NOx self-reactions can be more dominant and the true ship NOx contribution might be somewhat lower than simulated (Kasibhatla et al., 2000; Chen et al., 2005; Huszar et al., 2010). Such overestimated ship NOx contribution to ozone shows up, for example, in terms of a lower simulated vertical gradient than the observed vertical profile of ozone especially at remote coastal locations. To assess this, we plot observed and model simulated ozone vertical profiles at Trinidad Head, off the coast of California, for the month of July (a representative month for peak season) for all 19 years (see Figure S5). The

monthly mean modelled vertical O3 profile over Trinidad Head generally falls within the envelope of daily observational profiles within the MBL (say, below 850 hPa). Although, for multiple years, the vertical drop in modelled O3 concentration towards the surface is less sharp than that seen in observations, thereby suggesting a potential overproduction of O3 near the ocean surface in the model due to instantaneous distribution of ship NOx emissions in the model gridcell. This particular feature of our modelling system can partly explain the positive bias in simulated ozone."

**One zeroth-order check that the authors should perform is a comparison of their total marine ozone production from ship plumes with their total ship emissions NOx during the photochemical active season of the year; approximate agreement with a 4.25:1 mole/mole ratio is expected (i.e., production of 10 O3 molecules per NOx emitted during the day, based on the Chen et al., 2005 study), and loss of 1.5 O3 molecules per NOx emitted during the night, assuming that NOx is lost as N2O5 at night). Duncan et al. (2008) further discuss this issue. This comparison should be limited to ozone production from ship emissions within the MBL at northern mid-latitudes.**

We tried performing this sanity check with the output data available to us. However, there were a number of obstacles to ascertaining the suggested 4.25:103-from-ship: ship-NOx molar ratio as suggested by the reviewer. First, we only have monthly mean emissions from each source in the input and therefore cannot explicitly distinguish daytime and nighttime NOx emitted from the ships. Similarly, due to storage constraints, we do not output 3D ozone from different tags, including ships, at the hourly frequency. We only output surface ozone from each tag at the hourly level and 3D ozone from all tags at the monthly mean level. Therefore, we could only perform this MBL-wide analysis at the monthly mean scale. Using July 2018 as a representative month for the peak season, we calculated the total NO emitted from the shipping sector over northern hemisphere midlatitudes (30°N-60°N) to be 1.138e+10 moles. We then utilized the monthly mean 3D ozone concentration field from the ship NOx tag (raw output being in mol/mol-of-dry-air) to calculate moles of O3 attributed to ship NOx over July 2018. We found the total global O3 attributed to global ship NOx to be 5.308e+11 moles, O3 within the MBL attributed to global ship NOx to be 8.703e+10 moles, and O3 within the northern hemisphere midlatitude (NHML) band MBL attributed to global ship NOx to be 8.019e+10 moles. The ratio of NHML MBL ozone produced from ship NOx and the ship NOx emissions over NHML turns out to be **8.019e+10/1.138e+10 = 7.04** which is higher than the expected ratio of 4.25 as suggested by the reviewer. However, we can immediately see the issue with this approach: here we have a large number of moles of NO in the denominator which were arrived at by multiplying the per second emission rate with the number of seconds in a month (a large number), while the numerator was governed by the monthly average O3 concentration. If we perform the same calculation on a daily scale, the numerator won't change much (a daily O3 concentration maybe quite similar to a monthly mean O3 concentration) but the denominator (i.e. moles of NO emitted by ships in a day) would be substantially smaller, thereby making the ratio much larger. Therefore, a reliable sanity check could only be performed if we had tagged ozone production rates rather than tagged ozone concentrations. Unfortunately, our current tagging system does not provide ozone production rates from different tags so it is not feasible to estimate the molar ratio as suggested by the reviewer.

A second zeroth-order check that the authors should perform is a comparison of their simulated vertical profile of ozone within and directly above the MBL with observations. Ozone sonde data from Trinidad Head (e.g., Fig. 15, Oltmans et al., 2008, Fig. 12, Parrish et al. 2016 or Fig. 1, Parrish et al., 2022) are available; these represent the marine environment, so they should be compared with model simulations from a grid cell offshore of northern California. Note that Fig. 12 of Parrish et al (2016) compares the measured vertical gradient of ozone with simulations from 3 global models, one of which is a version of CAM-Chem model that was used in the present paper. Importantly, the observed strong near-surface vertical gradient of ozone over the ocean clearly indicates that the MBL is a region of strong domination by ozone loss; a spuriously large ozone source within the MBL, such as overestimate of the ship emission source, would be expected to disrupt the relatively strong vertical gradient through that level. Notably, none of the 3 models reproduced the observed ozone gradient within the MBL.

We obtained the ozone sonde data over Trinidad Head for the 2000–2018 period and were able to perform this sanity check. As mentioned earlier, we only store 3D ozone data at the monthly mean level from the model output. The sonde data were available at a daily level for a varying number of days per month over the 19 year period. We plotted the daily observed ozone profiles over Trinidad Head for the peak season representative month of July for the 2000–2018 period along with the corresponding model-derived monthly averaged ozone profiles (see Figure S5). We found that the monthly mean modelled vertical O3 profile over Trinidad Head was generally within the envelope of daily observational profiles within the MBL (say, below 850 hPa). Although, for multiple years, the vertical drop in O3 concentration towards the surface was less sharp than that seen in observations, thereby suggesting an overproduction of O3 near the ocean surface in the model due to instantaneous distribution of ship NOx emissions in the model gridcell. We have now discussed this in lines 472–485 and have qualified our conclusions relating to the contribution of the shipping sector to surface ozone (lines 1393–1394).

There are also indications of possible corruption of observational data relied upon in this paper. As noted earlier, Figures 4 and S1 include data that disagree strongly with model simulations and are simply physically unreasonable (e.g., too large in Fig. S1d and too small in Fig. 4i). Further, the authors discuss an anomalous drop in 2012 in the number of rural TOAR stations. Figure S8 shows that this drop was only in the US, not Europe, although southern Europe shows a peak in 2012 and western Europe shows a drop in 2017, both of which are unexplained. Northeastern US also shows a rather large increase in site number after 2015. The US ozone data are available from the US EPA data archive; that archive does not include the TOAR site classification, but the total number of US sites reporting ozone observations increased from 1241 in 2000 through 2010 and remained relatively constant at about 1450 for the three 2011-2013 years, followed by an accelerating decrease to 1231 by 2024. There is no indication of a drop in the number of stations in the US in 2012. Since the US monitoring network remains relatively constant from year-to-year with only small numbers of stations coming on line or closing, and with no systematic movement of sites to or from rural areas, it is clear that the data that the authors extracted from the secondary TOAR archive does not accurately reflect the parent EPA archive from which the TOAR data were obtained. These two features of the observational data make it imperative that the source of these problems be tracked down, and the observational analysis included in this paper be thoroughly evaluated and revised as necessary before included in a submitted manuscript.

We thank the reviewer for pointing this out. We further investigated the anomalous drop in the no. of stations in the US for 2012. It turned out that the data retrieval scripts for the TOAR-II gridding database had returned errors for many US stations for 2012 which were ignored by the automated retrieval system ultimately leading to many missing US stations for 2012. We now use an updated database which includes all valid stations for 2012. For SE Europe, the data for initial years comes from a single station, Aliartos, in Greece, which is a coastal location and may not be representative of the larger gridcell sampled. For southern Africa too, the data comes from a single station and there is lack of clarity regarding units (ppb or ug/m3) for the year 2014 from the original data providers. We have decided to exclude the observed data for 2014 for southern Africa. The slight increase seen in the number of stations in NE US is due to the inclusion of new non-EPA stations from the OpenAQ portal (https://openag.org). We have also updated the station weighting mechanism in the TOAR gridding system. For certain stations, there were multiple (two or three) sensors placed at the same site which provided independent ozone observations that led to over-weighting of those stations. Now, these time-series are first averaged to produce a single time-series per station. Thirteen new observational time-series were included in the updated dataset which were earlier ignored due to a special character in their filename. We have created a new time series plot of the number of valid stations per receptor region (Figure S11) using the updated TOAR-II dataset which now turns out to be free from any dramatic peaks and drops.

Since the observed data has changed significantly, we have also updated the model output used for further analyses by performing a fresh co-sampling using the new TOAR-II dataset. Regional averaged MDA8 O3 and PSO values were recalculated and re-plotted for all receptor regions. All figures in the new manuscript now use these new model-extracted values co-sampled against the updated TOAR-II dataset.

Sections 3.2 and 3.3, which comprise nearly half of the paper, present and discuss the results of the study. However, the discussion is largely a qualitative catalog of features apparent in the observations or model results. That discussion should be extensively revised to replace that qualitative catalog with more systematic and quantitative analysis approaches of the model results, comparisons between model results and observations, and between results in different receptor regions. The separation of long-term changes from the seasonal cycle (i.e., detrending the data before analyzing the seasonal cycle) is often very important, but the trends in ozone over the 2000-2018 period are so weak that this is not essential in this study. A Fourier series or spectral analysis, such as used by Parrish et al. (2016), Bowdalo et al. (2016) and Bowman et al. (2022) is recommended.

We thank the reviewer for their valuable comment. We have now entirely rewritten sections 3.2 and 3.3 with more emphasis on intercomparisons of contributions from the same source across different receptor regions. We have also discussed the observed and modelled seasonal cycles in terms of their fourier parameters.

Sections 3.2.1, 3.2.2 and 3.3.1 are primarily successive, isolated discussions of the ozone contributions in successive receptor regions, and much of this discussion is repetitive between regions. The authors provide little context for this discussion, so the reader is faced with unconnected qualitative descriptions and numbers; an improved organizing context is needed. I suggest that the authors successively discuss each of the ozone contributions over all receptor

regions in these sections, much as done in Section 3.3.2. For example, it would be informative to compare and contrast the anthropogenic NOx contributions to ozone in the receptor regions.

We have now rewritten these sections with more emphasis on contrasting contributions from the same sector across different regions.

As expected, Figure 5 shows that contribution decreases along western North America from the Southwestern US (with many large urban areas) to the Northwestern US (with few large urban areas); in this regard, it would be useful to include the western Canada receptor region, which is not similarly discussed in the paper.

We have not included discussion for Eastern Canada due to the unavailability of TOAR-II data from rural stations in this region which prevents model evaluation and co-sampling for this region.

It would also be of interest to quantitatively examine the correlation between that ozone contribution and the total local anthropogenic NOx emissions over all receptor regions.

We agree with the reviewer's valuable suggestion. We have now performed new analyses and created maps for North America and Europe showing the spatial distribution of the correlation coefficient r between total PSO and the local NOx contribution to PSO over these two continents versus the local NOx emissions. (see figures 7 and 13). We have now discussed this new analysis in sections 3.2.1 and 3.3.1 respectively for North America and Europe.

Insightful comparisons and contrasts with previously published analyses are essential. The quantitative results derived by the authors should be compared and contrasted with published results obtained by similar or differing analytical approaches to those quantifications; this further discussion must be based upon an in-depth literature review of published analyses of ozone trends and seasonal cycles and their changes over North America and Europe. Two specific examples of potential literature comparisons are summarized below; these examples should be considered as illustrative, but not as a comprehensive list of needed discussion topics.

We have now cited some previous studies (eg., Becker et al., 2023; Simon et al., 2024) to put our PSO results in context. We have also cited studies such as Bowdalo et al., 2016; Parrish et al., 2016 and Bowman et al., 2022 when presenting the quantitative results from the Fourier analyses.

Of great interest would be to fit the temporal evolution of the local anthropogenic contribution in each receptor region to an exponential function, rather than the linear analysis the authors employ. Parrish et al. (2025) and papers cited therein have shown that local anthropogenic enhancements of surface ozone in North American regions have decreased exponentially with a time constant of  $21.8 \pm 0.8$  years. From similar analyses, Derwent and Parrish (2022) report exponential time constants for the local anthropogenic contribution of  $18 \pm 4$  years over the United Kingdom and  $37 \pm 11$  years over continental Europe. Comparison of the present model results to those observationally derived results would be quite useful.

We decided not to fit exponential functions to the local anthropogenic contribution to PSO because we concluded that while the exponential fit as used in Parrish et al. (2022) and Parrish et

al. (2025) was better suited for capturing the the longer, fuller, time series of ozone which begins in the 1970s with an initial increase, peaks and then declines, a linear fit sufficiently captures the broadly secular decline in ozone over the shorter 19-year period considered in our study. However, we have now contrasted our tagged local NOx and foreign contributions to ozone to similar contributions derived through statistical approaches in published literature (see L626-645).

"Our model-based findings of declining local anthropogenic contributions to PSO in North America differ quantitatively with recent observation-based studies such as Parrish et al. (2025), which also document a significant waning of local influence using different metrics and inferential techniques. For example, Parrish et al. (2025) estimate a local anthropogenic enhancement to Ozone Design Values (ODVs) in the SW US of typically <6 ppb in recent years. Our direct tagging method quantifies a larger local anthropogenic NOx contribution to average PSO in this region (~16 ppb in 2014-2018, Figure 6h). This quantitative difference likely arises from several factors. First, PSO represents a 6-month seasonal average of MDA8 O3, while ODVs target specific high-percentile episodic conditions, and direct contributions to seasonal averages can be expected to differ from enhancements during specific episodes (although episodic contributions could be expected to have a higher share of local photochemistry than seasonal contributions). Second, and perhaps more fundamentally, inferential methods based on subtracting an estimated 'baseline' from total observed ozone may systematically underestimate the full impact of local anthropogenic emissions. Such approaches often define the baseline based on remote sites or specific statistical filtering, which may not fully account for the ozone produced from local emissions that is then regionally dispersed (as we also see indications of anthropogenic NOx and BVOC interactions in the tagged output) or the non-linear chemical feedbacks that occur when local emissions are present. In contrast, our emissions tagging technique directly attributes ozone formation to its original precursor sources as they undergo transport and chemical transformation within the model's complete and consistent chemical framework. This provides a mechanistic quantification of source contributions to the specific PSO metric under baseline conditions. While inferential methods provide valuable observational constraints, our tagging approach offers a complementary, process-explicit view of how different source categories contribute to the ozone burden, particularly illuminating the partitioning between local, regional, and intercontinental sources in the complex, evolving atmospheric environment"

Multiple other modeling studies have reported contributions to ozone that differ quantitatively from the present results. For example, Mathur et al. (2022b) find that "stratospheric O3 (ranging between 6 and 20 ppb) constitutes 29%–78% of the estimated Spring-time background O3 across the continental United States" while the present paper quantifies significantly smaller impacts: "the stratosphere contributes up to 6–8 ppb in the Southwestern US" (the US region of maximum stratospheric influence) and 4–7 ppb in the Northeastern US. Comparisons and contrasts of quantitative estimates from multiple studies are required in this paper.

We have now included comparisons with reported contributions from published literature.

L979-986: "The springtime (March-May) ozone has seen increases in both foreign NOx contributions (13.16 ppb to 14.81 ppb) as well as stratospheric contributions (12.02 ppb to 12.55 ppb; see Table 3 for a comparison across regions). Springtime mean stratospheric contribution is 12.55

ppb in the recent period (even higher in SW US at 14.25 ppb; Figure S7; Table 3). Previous studies have reported modelled stratospheric contributions in North America during observationally-identified episodes with higher values (e.g., 20-40 ppb; Lin et al., 2012) as well as seasonal mean contributions (6-18 ppb; Mathur et al. 2022b). Our seasonal mean values are lower likely because we do not sample the model output extensively from the mountainous region of western US, where stratospheric contributions are highest, due to lack of TOAR observations in those regions."

A clear and concise summary of what new understanding of the atmosphere has emerged from this study is lacking. The final section of the paper discusses Conclusions, Limitations and Future Outlook; it lists many findings, but it is not clear to me either what is new in this analysis, or which of the findings required the dual tagging system to uncover. That material should be revised to clearly and specifically answer several questions: What new knowledge of atmospheric chemistry emerged from this work? The paper does utilize a relatively novel tagging approach; can the authors provide the reader with a concise summary of when or for what issues the joint NOx and VOC source tagging is required? Or can they at least clearly summarize what additional information was provided by that technique in this study? (After all, the technique does greatly complicate the analysis, and in the end the added benefits are not clear to me.)

We have now significantly updated the Conclusions section to emphasize the new knowledge emerging from this study (L1385-1426) and the role of the TOAST 1.0 dual-tagging system in producing such knowledge (L1431-1445).

**Major issues:**

The format of Figures 5 and 9 should be improved to better illustrate the authors' discussion. Most of the source contributions are so small that their magnitudes and variation are difficult to discern in the present format. Improvements should include a) using a more nearly square format to more clearly show any systematic changes over the two decades, and b) perhaps using a log-scale for the ordinate to more clearly illustrate the magnitude of all source contributions. The nearly square format would be more easily obtained by a) moving the region labels into blank spaces within the graphs and b) by not repeating the years on the abscissa of each graph. (Similar comments apply to Figures 7 and 10.) The log-scale would also be more appropriate for showing the long-term ozone changes due to changes in anthropogenic emissions, since those emissions are expected to decrease in an approximately logarithmic fashion (i.e., linear on the log scale). Figures 6, 8, 11 and 12 are more readable, but could be improved by changing the ordinate scale to 0 to 30 in the 2nd and 3rd column graphs; this would cut off the top of the anthropogenic NOx contribution for the NE US, but it would be useful to duplicate this contribution in all of the 1st column graphs.

We have made several changes to the PSO time series figures for both NAM and EUR (now figures 6 and 12) to address the concerns of the reviewer. First, the NOx and VOC contributions are plotted on separate panels from the total PSO. So, there are 3 columns in the new panelplot. The height:width ratio of the panels has been increased and the maximum ordinate on the y-axis

has been cut off at 40 ppb for NAM and 30 ppb for EUR for the PSO contributions panels to ensure better discernibility of the trends for individual contributions. We have retained the linear-scale instead of the log-scale in line with our decision to fit linear trends to these contributions.

We have also modified the old seasonal cycle figures for NAM and EUR (figures 6 and 10). The height:width aspect ratio has been increased in the new figures (Figures 8 and 14) for improved visibility of the smaller contributions.

For the figures showing seasonal cycle change in NAM and EUR (now fig 9, 10, 15, 16, S6-S10), we have cut off the maximum ordinate on the y-axis for the middle and right panels (those showing source contributions) to 40 ppb to ensure better visibility. For NE US, as recommended by the reviewer, we have duplicated the local anthropogenic NOx contribution to the average seasonal cycle over 2000-2004 on the main panel with total MDA8 O3 seasonal cycle for that region (Figure 10).

In Section 3 the authors discuss the long-term changes in ozone and its components in terms of linear trends derived from a Theil-Sen approach; these trends are collected in Table S1. Several issues should be discussed in this regard. First, the emissions illustrated in Figure 2 appear to be non-linear; thus, at least some of the long-term changes can be expected to be non-linear. The authors should discuss why they employ a technique that can only quantify the linear aspects of the long-term changes. Logan et al. (2012) quantify changes in linear slopes (i.e., trends) over a 3 decade long data record; perhaps such an analysis should be employed in the present discussion?

Logan et al. (2012) and others (e.g., Parrish et al., 2025; Parrish et al. 2020) indeed chose to fit a quadratic function to ozone trends in North America and Europe. We believe that this choice was appropriate given the longer time-series considered in these studies which included both an initial increase in observed ozone in the 1980s and 1990s followed by a decrease in the 2000s. However, for our study, we focus on the 2000-2018 period when the ozone trends in these regions have consistently declined and, based on a visual inspection, concluded that linear trends (along with p-values and 95% CI) as recommended in TOAR statistical guidelines (<a href="https://igacproject.org/sites/default/files/2023-04/STAT\_recommendations\_TOAR\_analyses\_0.pdf">https://igacproject.org/sites/default/files/2023-04/STAT\_recommendations\_TOAR\_analyses\_0.pdf</a>) are appropriate. We have now mentioned this point in the manuscript:

L538-545: "Figure 6 presents the time series of observed and model-simulated total PSO (panels a, d, g, j, m), alongside the attributed contributions from NOx sources (panels b, e, h, k, n) and VOC sources (panels c, f, i, i, o). On a visual inspection of observed and modelled PSO trends (left column panels) we decided to fit Generalized Least Squares (GLS) linear trends to these data points. We note that some previous studies have fitted higher order functions to ozone data over North America as necessitated by their longer period of analysis where ozone concentrations increased, stagnated, and then decreased (Logan et al., 2012; Parrish et al., 2025; Parrish et al. 2020). However, a linear fit is appropriate for the period considered in our study when local emissions have only declined (Figure 2). Quantitative details of the trends and their significance for all contributions are provided in Table 2."

Second, the authors report values they derive for the significance of their derived trends; this significance only informs us regarding whether the trends are significantly different from zero (it

is not clear to me what is implied by a trend of zero with a significance of 1 – that seems nonsensical). The authors should report 95% confidence limits for their derived trends if they indeed judge a linear analysis is adequate to quantify statistically significant long-term changes; these confidence limits are of much greater interest than the significance statistic, as they provide a basis for judging quantitative comparisons such as the authors give on lines 451-453; as presently written, it is not clear that the -0.24 ppb/yr (1.0) trend derived from the observations differs significantly from the -0.35 ppb/yr (0.99) trend derived from the model results.

We have now fitted linear trends based on Generalized Least Squares (GLS) method instead of the Theil-Sen method, as recommended by the TOAR statistical guidelines, and have also included p-values along with 95% confidence intervals. These are included in Table 2.

Finally, and most importantly, the significance values (and potentially any calculated confidence limits) are apparently greatly over-optimistic. If I understand correctly, each region has only a single PSO value each year. Given the limited (i.e., 19) number of PSO values combined with the relatively large interannual variability and autocorrelation that characterize observed time series of ozone concentrations, only modest significance values (and relatively wide calculated confidence limits) are expected. Importantly, for time series of annual PSO values, autocorrelation over multiple year must be considered in deriving reliable confidence limits. Fiore et al. (2022) discuss this issue more fully.

The new GLS linear trend analysis on the new data (with updated TOAR-II data and model co-sampling) yields lower significance values (i.e., higher p-values) and broader 95% confidence intervals. Autocorrelation was considered while deriving these trends and confidence limits.

More generally, the authors provide few confidence limits for the quantitative numbers given. It is generally acknowledged that any scientific paper presenting results of quantitative analysis must include confidence limits for the quantitative findings. This last comment applies to all of the quantitative results presented in the paper. I realize that it is difficult to quantify confidence limits for model results; nevertheless such quantification is essential. Developing such confidence limits could come both from the additional statistical analysis indicated as needed in the first Critical Issue discussed above, better quantitative treatment of the seasonal cycle and its shifts as suggested in the second Critical Issue, also discussed above, and further bolstered from quantitative comparisons of the results from the modeling study presented in this paper with other observational and modeling results, again indicated as needed in the second Critical Issue.

We have now provided 95% confidence limits for both observed and modelled PSO as well as modelled contributions to PSO (see Table 2). We have also taken a much more quantitative approach in model evaluation as well as analysis of seasonal cycles (sections 3.2.2, 3.2.3, 3.3.2, 3.3.3) by breaking down both observed and modelled seasonal cycles into two harmonics and comparing their various parameters, as discussed in previous responses.

The sentences on lines 496-499 are contradictory and obscure an important point: "The model reproduces the 19-year average seasonal cycle over different parts of North America very well. For western regions, we see a consistent systematic positive bias of 2-4 ppb. For eastern regions

we see a *very good reproduction* of the seasonal cycle during winter and spring but *a notable overestimation during summertime* (italics added)." The italicized words are where the contradiction arises. The authors further discuss the summertime overestimate, in multiple places. I suggest a single, consistent discussion of this feature over all of the North American regions.

We have now rewritten this section with a quantitative description and the supposedly contradictory sounding sentences have been removed.

Lines 532-533 state that "Figures 6e and f show the average seasonal cycle of MDA8 O3 in Southwestern US which is similar to that for the Northwestern US ...." I agree that the shapes are somewhat similar between the two regions, but the minimum-maximum difference is significantly smaller in the Southwestern US (~12 ppb) than in the Northwestern US (~22 ppb). This mis-judgement, evidently based on a qualitative assessment of Figure 6, emphasizes the need for the utilization of a quantitative analysis of the ozone seasonal cycles, as detailed above in the discussion of second Critical Issue. There is a similar but smaller mis-judgement in the following comparison of the Southeastern US and Northeastern US seasonal cycles.

These seasonal cycles are now discussed in terms of their Fourier parameters which avoids such visual mis-judgement.

**Minor issues:**

Line 31: The phrase "... especially towards the end of the 20th century" would be more accurate if changed to "... especially during the last half of the 20th century". Substantial ozone increases in the troposphere have been documented over that entire period.

We have now modified this sentence accordingly.

Line 56: The authors state that "This is due to the long-enough atmospheric lifetime of ozone (about 3-4 weeks) which allows it to traverse intercontinental distances and affect the air quality of regions far from the location of its chemical production or the location of the emission of its precursors." This statement is true as written, but should be discussed a bit further. The loss processes leading to that lifetime are dominated by loss in warmer, more humid tropical regions. Further, this lifetime refers to the total photochemical loss processes integrated over the entire globe; considering net ozone tendency, the effective lifetime of ozone in an air parcel transported in the free troposphere at northern midlatitudes (the zone of focus of this paper) is on the order of several months. This is long enough that the free troposphere can be considered a reasonably well-mixed reservoir, further emphasizing the importance of transport over intercontinental distances within this latitude zone.

L62-70: We have now modified the text to the following: "This is due to the long-enough atmospheric lifetime of ozone which allows it to traverse intercontinental distances and affect the

air quality of regions far from the location of its chemical production or the location of the emission of its precursors. While the global average tropospheric lifetime of ozone is often cited as approximately 3-4 weeks, a figure largely influenced by more rapid photochemical loss in warmer, humid tropical regions (e.g., Stevenson et al., 2006; Young et al., 2013), the effective lifetime of ozone in air parcels transported within the cooler, drier free troposphere at northern midlatitudes is considerably longer, on the order of several months (e.g., Jacob, 1999; Wang and Jacob, 1998; Fiore et al., 2009). This extended lifetime in the primary transport pathway for intercontinental pollution allows ozone to traverse vast distances and enables the northern mid-latitude free troposphere to act as a relatively well-mixed reservoir (Parrish et al., 2020)."

Lines 63-64: The authors correctly note that studies have identified "increasing trends in wintertime and background ozone concentrations at many sites in North America, particularly at the US west coast". For completeness, it would be useful to further point out that such increases have also been identified throughout the background troposphere at northern midlatitudes including in the free troposphere, but that ozone in this latitude zone reached a maximum in the first decade of the 2000s (e.g., Parrish et al., 2020; Derwent et al., 2024).

We have now added the following sentence in L79-81: "Such increases in ozone have also been identified throughout the background troposphere at northern midlatitudes including in the free troposphere, with a peak attained in the first decade of the 2000s (e.g., Parrish et al., 2020; Derwent et al., 2024)"

Lines 79-81: The authors could expand this sentence for completeness. As Derwent et al. (2024) discuss, a hierarchy of models is required to fully understand tropospheric ozone. We require not only statistical interpretations of observational data and well-evaluated atmospheric chemical transport models, but also conceptual models that simplify and capture the essence of the most salient physical and chemical processes that control observed ozone abundances.

We have now added the following text in L101-103: "Together, observational analyses and model-generated results can aid the theoretical development and improvement of simpler conceptual models that capture the essence of the most salient physical and chemical processes that control observed ozone abundances (Derwent et al., 2024)."

Lines 88-102: It seems to me that this paragraph overemphasizes the shortcomings of the *perturbation*. Is it not possible to simply limit the approach to such small perturbations that the atmosphere processes are not significantly changed, and the results approach perfect accuracy?

You raise a pertinent point regarding the theoretical limit of small perturbations. Indeed, for infinitesimally small changes in emissions where the atmospheric chemical regime remains essentially linear and unperturbed, the calculated sensitivity might more closely approximate a direct fractional contribution. However, practical source apportionment using the perturbation method typically involves substantial reductions (e.g., 20–100%) of an emission source or sector to obtain a clear and robust signal above model noise and inherent atmospheric variability. Such large perturbations inevitably alter the chemical regime (e.g., by changing NOx/VOC ratios or HOx cycling), meaning the difference between the perturbed run and the baseline reflects the *impact of that source's removal* rather than its *contribution within the original atmosphere*.

To make this clearer, we added an additional line in the text:

L124-126: "On the other hand, tagging techniques, which track the fate of emissions from designated sources as they undergo transport and chemical transformation within the unperturbed baseline atmosphere, allow us to assess the contribution of various sources under a baseline scenario when no policy intervention has been made."

Table 1 lists 9 regional oceanic tagged regions. However, these regions are not mapped in either the paper or the Supplement. Please include such a map; for example, it is of interest to understand how the North Atlantic Ocean is divided into 3 separately tagged regions.

Thanks for pointing this out. We have now added a new global map of the tagged oceanic regions. See figure S12.

Lines 264-274 describe the derivation of MDA8 values from the model output, and lines 287-292 describe the derivation of MDA8 values from the TOAR rural observations. Please discuss if these methods are completely compatible, or if differences in the procedures may possibly be important.

The procedure for calculating MDA8 O3 is essentially the same for both observations and model output. The fact that MDA8 O3 is only calculated for days where at least 18 of the 24 hourly values are available in the observations allows us to minimize any discrepancies between the observed and model-derived MDA8 O3. In other words, the model-derived MDA8 O3 is only sampled into further analyses (i.e. spatial averaging over receptor regions) if the corresponding TOAR-MDA8 O3 has >18 hourly data points. We have emphasized this in the text by adding a line: "This allows us to minimize any discrepancies between the observed and model-derived MDA8 O3 values". (L342-343)

Lines 278-279 state: "We use these receptor regions to perform area-weighted spatial averaging of MDA8 O3 values before analysing the trends and contributions. Please explain the process of "area-weighted spatial averaging", and why it is used rather than simple averaging.

We have now added the following text in L323-330: "Area-weighted spatial averaging is needed because different model grid cells cover different areas on the ground based on the rectangular lat-long coordinate system, with high-latitude grid cells covering smaller areas and low-latitude and equatorial grid cells covering larger areas. So, a simple spatial averaging will overrepresent the concentrations of high-latitude gridcells and underrepresent lower-latitude gridcell concentrations in the receptor region average. So, we derive dimensionless coefficients ranging for all grid cells within each receptor region based on their relative size to the average grid cell area in that region. We scale the gridded MDA8 O3 with these area-coefficients before averaging, ensuring a proportionate representation of the MDA8 O3 value over the entire receptor region."

Lines 405-406 note "that for all regions in North America, the observed PSO exceeds the WHO guidelines throughout the 2000 -2018 period." It should be emphasized that the WHO guideline is based on the highest tail of the MDA8 distribution, while mean values are discussed in this

paper. Thus, the exceedance of the WHO guideline over North America is indeed profound, as reflected in the difference in the WHO (~50 ppb) and US EPA (70 ppb) guidelines.

We do not analyze daily exceedances in our study but focus on Peak Season Ozone (PSO) and the WHO guideline for PSO (~31 ppb) is based on a maximum 6-month *average* MDA8 O3 which is exactly what we use in this study.

Line 450 contains a typo – the WHO guideline is 51 ppb.

By the WHO guideline, here, we are referring to the PSO guideline by the WHO, which is indeed 60 ug/m3 or  $\sim$ 31 ppb. More details are available on page 102 of this report: https://iris.who.int/bitstream/handle/10665/345329/9789240034228-eng.pdf

Each panel of Figures 5 and 9 illustrates a time series of a "Residual O3" contribution. I have not found that quantity defined or discussed in the manuscript. Can this be eliminated? If not please define and discuss.

Thanks for pointing this out. This "residual ozone" is ozone which can not be clearly attributed to either a NOx or a VOC precursor, and which is not associated with the photolysis of molecular oxygen in the stratosphere,, for example the ozone formed from O atoms resulting from the self-reaction of hydroxyl radicals. We have added a sentence to the discussion of the tagging system in the methodology section (lines 239-243) describing this. This residual ozone can not be eliminated, but typically only makes a very small contribution to ozone mixing ratio at the surface (about 1 ppb in this study).

Lines 727-728 state that "the increasing contribution of natural NOX emissions we find in our study, especially during the summertime, is most likely due to the increasing ozone productivity of these emissions." This appears to be speculative; if this statement is to be included in the Conclusion section, it should be shown to be true through quantitative analysis.

L1394-1397: We have added further reasoning to the text to support the statement: "the increasing trend in modelled contribution of natural NOX emissions, especially during the summertime, suggests increasing ozone productivity of these emissions since there is no noticeable increasing trend in natural NOx emissions and a slight decreasing trend in Lightning NOx emissions (Figure  $3\,a,\,c,\,e,\,q$ ).".

Lines 729-730 state that there is "... a smaller effect in the springtime, when long-range transport of ozone produced from foreign anthropogenic NOX emissions is more important." I believe that in the literature there is ongoing discussion regarding whether ozone produced from foreign anthropogenic NOX emissions or ozone of stratospheric origin is of most importance for springtime surface ozone over North America (if not also Europe); I suggest that the authors mention both of these sources in this context.

Thank you for this important point. We have now modified the text as "when long-range transport of ozone produced from foreign anthropogenic NOX emissions and stratosphere is more important". We have also now quantified the springtime foreign NOx and stratospheric contributions to all regions in Table 3.

**Community Comment (by Owen Cooper)**

General comments:

There are some data issues that need to be addressed:

1) Figure S8 (top panel). Please check the North American data availability for 2012. This figure indicates that very few sites are available, but I went to the TOAR-II database and using the plotting tool I found plenty of data for that year. Please see the example figure pasted below.

We have updated the TOAR data (please also see the response to reviewer 2 in this regard) and now there is no significant drop in the no. of North American stations in 2012. All model data used in subsequent analyses is also updated based on new co-sampling corresponding to the updated TOAR-II data.

2) Line 370. Here you mention that you compared the model to observations in regions with limited data. But are there enough data available for a meaningful comparison? For example, the TOAR map tool doesn't show any data in Belarus or Ukraine, and only urban data in Russia. The same applies for North Africa. Please indicate the number of rural stations with ozone data in all of the regions analyzed in this study, and explain how representative you believe the data to be.

We have now updated the text with the number of TOAR stations as available in the TOAR-II gridded dataset. The model-observation representativeness is maintained by co-sampling the model output from only those gridcells where TOAR stations are present. We have highlighted this in the updated text: "We have also included the Belarus & Ukraine region (Figure 4j; with 1-2 valid stations) in our evaluation and here too we see a good simulation of MDA8 O3 for the entire period (with a small mean bias of 0.56 ppb and r value of 0.83), barring a couple of years (2014 and 2017) when the model overestimates the values. We have also evaluated the model for MDA8 O3 against rural observations from the TOAR-II database in other regions including Mexico (11-14 stations), North Africa (1-3 stations), Southern Africa (1 station), Southern Latin America (1-2 stations), and European Russia (2 stations; see Figure 3 for region definitions), where the model has also captured the trends well, however, since we do not discuss these regions in further analyses, they are presented in the supplement (see, Figure S1). Here too, the model output is extracted only from those grid cells where at least one TOAR station exists, ensuring representative co-sampling."

In the introduction it would be helpful to cite some of the key papers from the first phase of TOAR as they are highly relevant to the background information on the importance of ozone for health, vegetation and climate: Fleming and Doherty et al. (2018), Mills et al. (2018), Gaudel et al. (2018). Another key reference for ozone's impact on climate is IPCC AR6 (Szopa et al., 2021).

Fleming and Doherty et al. (2018), Mills et al. (2018), and (Szopa et al., 2021) have been included in the text with appropriate context. See lines 47-52.

**Lines 63-67**

Regarding the observed increase of background ozone at northern mid-latitudes, some of the cited references are out of date and other recent studies have not been cited. IPCC AR6 (Gulev et al., 2021) assessed an increase of free-tropospheric ozone at northern mid-latitudes through the year 2016. Follow-up studies by Wang et al. (2022) and Christiansen et al. (2022) used the updated IAGOS and ozonesonde records, and showed similar results to IPCC AR6. New papers that have emerged from the TOAR-II effort show ozone increases through 2019, especially over East Asia (Eshorbany et al., 2024; Lu et al., 2024). It is also worth mentioning several recent studies that show ozone increased through at least 2019, but since 2020 ozone has decreased slightly, or levelled off, in association with the COVID-19 economic downturn (Miyazaki et al., 2021; Chang et al., 2022; 2023; Ziemke et al., 2022). A good review of U.S. background ozone is provided by Jaffe et al., 2018.

We thank the reviewer for providing this comprehensive list of recent papers with related research. On a careful reading of the above papers, we found the findings of Jaffe et al., 2018; Christiansen et al., 2022; Elshorbany et al. 2024; and Lu et al., 2024 to be directly relevant to the discussion in our study and have included them in the text now. Christiansen et al., 2022 has been added to the list of citations in L79, and the following line is added mentioning Jaffe et al., 2018: "Jaffe et al., (2018) performed a comprehensive knowledge assessment of background ozone in the US and emphasized its growing relative importance and advocated for, among other things, a more strategic observational network and new process-based modelling studies to better quantify background ozone in the US to support informed clean air policies."

Elshorbany et al., 2024 and Lu et al., 2024 are included later in the results section (section 3.4) "These results are consistent with findings of Elshorbany et al., (2024) and Lu et al., (2024) who report increasing ozone trends in Asia both in the troposphere and at the surface which stabilize around 2013".

Table S1, and wherever trends are reported in the paper

As described in the TOAR-II Recommendations for Statistical Analyses, all trends need to be reported with 95% confidence intervals and p-values. It's not clear what is meant by "and their significance (shown in brackets)."

All trends are now reported along with p-values in parentheses and 95% confidence intervals in square brackets at all instances in the text as well as in Table S1. The trends were also revised based on Generalized Least Squares (GLS) fit in line with the TOAR statistical guidelines.

**Line 349**

For context, a very good review of ozone across the southwestern USA is provided by Sorooshian et al., 2024. Also, Simon et al., 2024 provide an update on current impacts of emissions on U.S. surface ozone.

Since we have substantially rewritten this section, this citation didn't fit well at this place into the new text, however, we have included many references from within Soorooshian et al., 2024 (which is indeed a rich review) in the results section (section 3.2.3; L975-977).

We have also cited Simon et al. (2024) in section 3.2.1 L563-566: "These results are consistent with findings from Simon et al. (2024) who analysed observational trends over 51 sites in the US over roughly the same period (2002-2019) and found the marked impact of clean air policies across the US such that the difference between the weekend (lower NOx) and weekday (higher NOx) MDA8 O3 has diminished and become negative in recent years reflecting a transition from NOx-saturated to NOx-limited ozone formation regime."

and in L575-577: "This lack of correlation between local NOx emissions and observed MDA8 O3 has been reported by Simon et al. (2023) for rural California even at a higher temporal frequency through disappearing day-of-week activity patterns indicating an increasing role of transported ozone in this region."

**Line 393**

It would be helpful to briefly mention how your PSO values compare to previous studies, such as Becker et al., 2023.

We have now included a reference to Becker et al. (2023) when discussing spatial PSO results for NAM and EUR (these are new maps in the revised manuscript).

L531: "The spatial features of PSO for both years are very similar to bias-corrected maps of PSO for 2000 and 2017 presented in Becker et al. (2023)."

**And L1113-1116**

"However, this decline is not obviously reflected in the spatial patterns of total PSO (Figure 11b, e), which generally decreased in the southern regions but not in northern regions, especially over areas with the largest emission cuts, as also seen in bias-corrected PSO maps by Becker et al. (2023)"

**Line 594**

It's worth pointing out that the change in ozone seasonal cycle is expected to continue to shift with future emissions changes, as discussed by Clifton et al. (2014).

We have included this reference and included an extra line "This transition in the ozone seasonal cycle in the NE US, towards a springtime maximum, is expected to continue with future emissions changes, as discussed by Clifton et al. (2014)" in section 3.2.3 at L1032.

The submitted paper provides few references regarding the important role that methane plays in ozone production. A recent assessment of the impact of methane on tropospheric ozone is provided by the UN Climate and Clean Air Coalition:

United Nations Environment Programme and Climate and Clean Air Coalition (2021). Global Methane Assessment: Benefits and Costs of Mitigating Methane Emissions. Nairobi: United Nations Environment Programme. ISBN: 978-92-807-3854-4. Job No: DTI/2352/PA <a href="https://www.ccacoalition.org/sites/default/files/resources//2021\_GlobalMethane\_Assessment\_full\_0.pdf">https://www.ccacoalition.org/sites/default/files/resources//2021\_GlobalMethane\_Assessment\_full\_0.pdf</a>

We have now included this reference in the introduction at L42.

---

## Referee Report (RR1)

$2^{nd}$  review of "Explaining trends and changing seasonal cycles of surface ozone in North America and Europe over the 2000-2018 period: A global modelling study with NOX and VOC tagging" by Ansari et al.

MS Number: egusphere-2024-3752

**Summary:**

This paper is a revision of an earlier manuscript presenting a global modelling study with an innovative dual tagging analysis; the paper's focus is on surface ozone over North America and Europe, particularly with regard to the causes of long-term changes in mean concentrations and the seasonal cycle and its changes. In my view, this manuscript represents a very significant improvement over the previous version, and addresses all issues that I raised in my first review, although some are still of concern. I recommend acceptance for publication when some remaining issues are addressed. Below I review the responses to the critical issues I identified in my previous review, discuss several remaining and new major issues, and list a few new minor issues.

**Review of Critical Issues previously raised:**

• Section 3.1 is devoted to **evaluation of the CAM-Chem model** used in the chemical-transport simulations. The more robust evaluation now included in the paper is a significant improvement. I believe this section now gives a more comprehensive evaluation of the model performance, an approach that should be emulated by other studies presenting global modeling results. Nevertheless, a few lingering issues remain, specifically:

Line 378 suggests that "... all HTAP tier 2 receptor regions for North America" were used for model evaluation. However, only 5 are included, while Fig. 3 of their reference (Galmarini et al., 2017) identifies 6 – Western Canada is excluded. I believe that this (at least below 60 deg N) is a particularly important region to include, both in the model evaluation and in the later discussion of results, since it provides a useful contrast with the more urbanized NW US and SW US inflow receptor regions lying to the south. Further, it may provide insight into reasons for the model overshoot of the maxima and undershoot of the minima for the earlier years of 2000-2006 in Eastern Canada. Please provide the comparison for Western Canada, or fully discuss the reason(s) for its being excluded. Importantly, please note that rural data are indeed available from this region - see Fig. 5 of Galmarini et al. (2017) and the figure that Owen R. Cooper included in his comment on this manuscript (https://doi.org/10.5194/egusphere-2024-3752-CC1).

In my previous review I raised a subtle issue associated with model-observation comparisons such as the authors present in Section 3.1 and utilized in many such comparisons in the published literature, i.e., a degree of circular reasoning that results from models developed to agree with observations, so that such agreement cannot be taken as independent confirmation that models perform properly for the correct reasons. In their response to the reviews, the authors acknowledge a general concern with this issue, which they discuss in detail that generally discounts any large effects on their comparison. Some of this discussion is persuasive, but I could dispute other parts in further discussion; for example, nearly all persuasive model-measurement comparisons, whether for purposes of model "tuning" or evaluation of results, focus on metrics that are less sensitive to "nighttime ozone and avoid any large nighttime biases, which often arise due to improper simulation of the nighttime

boundary layer". Rather than push this issue further, I suggest that the authors simply describe this issue and note their concern in one or two sentences in Section 3.1, perhaps at the end of the paragraph beginning on line 368.

I commend the authors for their illuminating discussion of the treatment of ship emissions in chemical transport models, now included in Section 3.1. In their response, the authors do attempt the first zeroth-order check that I suggested, and note a problem in comparing emissions in units of moles NOx/month with monthly average ozone concentrations in units of moles. However, the numbers derived in their comparison for July 2018 would be correct if the total global O3 attributed to ship NOx had a lifetime relative to total gross destruction of one month. This lifetime in the summertime northern mid-latitudes is likely shorter than 1 month. For example, in July in the marine boundary layer (where the ozone derived from ship emissions is primarily formed) that lifetime is estimated to be  $\sim 10$  days (see discussion in Mims et al., 2022 and their Figures S4 and S5). However, some of the ship derived ozone is transported to the cooler and drier free troposphere before destruction, so the effective overall lifetime for ship-derived ozone in summer at northern mid-latitudes might be best estimated as ½ month; thus this first zeroth-order check would indicate a model overestimate of ship derived  $O_3$  on the order of a factor of (7.04/4.25\*2 =) 3.3. The authors also perform the second zeroth-order check that I suggested, and have included a discussion of this issue in Section 3.1 and the Supplement. Overall, I judge these responses to be an adequate response to this issue; however, I suggest inclusion of a discussion of the first zeroth-order check in the Supplement for interested readers to peruse, and in discussion of ship emissions in the paper, the authors should consider a likely overestimate of ship emissions by this factor.

- I am pleased that the issues of **possible corruption of observational data** have been resolved.
- The discussion in Sections 3.2 and 3.3 now include more systematic and **quantitative** analysis approaches; it is now much improved, with the new Figures 5, 7, 11, and 13 adding significant additional information and clarity. However, several issues remain as detailed in many of the Major Issues below.
- Section 4. Conclusion, Limitations and Future Outlook now provides a clear and reasonably
  concise summary of the new understanding of the atmosphere that has emerged from this
  study.

**Major Issues:**

- 1) I note that the current manuscript contains 17 figures composed of nearly 110 separate graphs, most with multiple traces that tend to overwhelm the reader. Some comments below suggest combining or revising manuscript figures and/or moving figures to the Supplement. Any changes the authors can make in this regard would improve the paper.
- 2) In my first review I noted that the modeled contribution of local anthropogenic NOx to PSO decreases along western North America from the Southwestern US (with many large urban areas) to the Northwestern US (with few large urban areas); in this regard, I suggested that it would be useful to include the Western Canada receptor region in the analysis in Section 3.2 Ozone in North America. This inclusion would extend the western North American contrast to a region without large urban areas. The authors responded "We have not included discussion for Eastern Canada due to the unavailability of TOAR-II data from rural stations

in this region which prevents model evaluation and co-sampling for this region." This response is inaccurate; first, my comment referred to Western, not Eastern, Canada, and second, in fact there are TOAR-II data from rural stations in Western (as well as Eastern) Canada as I note in discussion above, and as is readily apparent in the authors' Figures 5 and 7 which include results for that Western Canada region. In my view, this western North American contrast is one of the more exciting prospects for improved understanding of surface ozone that could emerge from this manuscript, and should be exploited as fully as possible. It may help illuminate the comparison between the present work and recent observation-based studies that the authors discuss in the paragraph beginning on line 608. Please provide and fully discuss the model calculations for Western Canada as a 6th North American receptor region, or fully discuss the reason(s) for its exclusion.

3) I suggest that Figure 6 be replaced with simpler, more informative figures.

For discussion of observed and model-simulated total PSO (paragraphs beginning on lines 524 and 534) I suggest replacing the 5 graphs a, d, g, j, m and the potential 6th graph for Western Canada with a two graph figure – one with all observed and the other with all model-simulated total PSO time series. Include the linear fits in both panels, and expand the ordinate to just the PSO range spanned by the time series (35 to 65 ppb?). For these graphs linear fits are appropriate, as the derived slopes quantify the scientifically interesting quantity of average PSO trends over the 6 receptor regions during the 2000-2018 period. Such a figure will allow improved visual comparisons of a) regional differences in PSO within the same graph, b) observed and model-simulated total PSO in side-by-side graphs, and c) the quality of the linear fits superimposed on the fitted data in each graph.

For discussion of the local anthropogenic NOx contribution (paragraph beginning on line 541) I suggest a figure with a single graph illustrating the time series from all 6 North American receptor regions. The 0 to 40 ppb ordinate range in the present Figure 6 should be retained. However, here I (again) suggest that the time series of this contribution in each receptor region be fit to an exponential function, rather than the linear analysis the authors currently employ. The rationale for this suggestion is discussed fully below in the 4th Major Issue. Such a figure will allow improved visual comparisons of a) regional differences in this PSO component within the same graph, and b) the quality of the exponential fits superimposed on the fitted data.

For discussion of other NOx tagged contributions and all VOC tagged contributions, the time series plots in the current Figure 6 provide little information, and the discussion of their trends is based on the derived linear trends from Table 2, while the plots are not directly discussed in the relevant paragraphs (lines 574-607). Thus I suggest those plots be moved to the Supplement.

I suggest a similar treatment of Figure 12.

4) In the paragraph beginning on line 534 the authors discuss linear trends in PSO quantified by means of linear fits to time series. They note regional differences in the trends derived from the observations, and comment that the "model generally captures these decreasing trends and the interannual variability reasonably well, though with some regional differences in magnitude ...." Additional discussion should quantify what is meant by "reasonably well" (i.e., quantify differences between modeled and observed trends and provide correlation coefficient between them), and discuss likely reasons for regional differences in trends (e.g.,

are larger trends found in regions with larger local NOx emissions?).

- 5) I must strongly argue that the most interesting science question to address with regard to the local anthropogenic NOx contribution to PSO is quantification of the average *relative* rate of decrease. A particular *fractional* decrease in local anthropogenic NOx emissions is expected to give a larger *absolute* decrease in a region of large emissions compared to a region of small emissions; however the *relative* decrease may well be similar in the two regions. Similar *relative* rates of decrease of local anthropogenic NOx contributions to PSO would suggest similar *fractional* emission reductions and similar photochemical environments in the different regions, even if the *absolute* emissions differ, and thereby causing the *absolute* local anthropogenic NOx contribution to PSO to differ between those regions. An exponential fit to a time series of a quantity provides a means to quantify the average *relative* rate of decrease of that quantity. This is the rationale for my strong recommendation that the time series of the local anthropogenic NOx contribution be fit to an exponential function, since it gives a quantification that is of much greater scientific interest than the quantification of the *absolute* rate of decrease provided by a linear fit. Alternatively, the authors could employ a different technique to quantify the average *relative* rate of decrease.
- 6) The authors' have added a very useful paragraph (beginning on line 609) that compares their results to the recent observation-based studies of Parrish et al. (2025). That comparison focuses on the apparent difference between the relatively small (<6 ppb in recent years) local anthropogenic enhancement to Ozone Design Values (ODVs) found in the observation-based study and the larger (~16 ppb) local anthropogenic NOx contribution to average PSO found in the authors' model-based analysis. Despite some differences (discussed by the authors) between the two quantities derived to characterize the local anthropogenic contribution, it is clear that the authors have identified an important quantitative difference between the results of the two analyses.

The authors suggest that perhaps the fundamental cause of the difference in results is that the observational-based method may systematically underestimate the full impact of local anthropogenic emissions, thereby overestimating the background contribution. However, they do not provide robust, quantitative analysis to support this suggestion. Importantly, Parrish et al. (2025) and references therein thoroughly discuss multiple lines of quantitative analysis to demonstrate that their observation-based approach does provide quantitatively accurate estimates of the ODV contributions from background ozone (US background ODV) and the local anthropogenic enhancement to ODVs. For example, there is strong evidence that zeroing out anthropogenic emissions in at least some global modeling systems lead to underestimates of the US background ODV; Parrish et al. (2025) discuss the work of Hosseinpour et al. (2024), who report the 4th highest US background (USB) MDA8 ozone concentration across the US from a global modelling system (CMAQ-CAMx) different from that used by the authors. The graph at left below shows that the model gives results that are biased (mean absolute bias = 11 ppb) from those of Parrish et al. (2025), much as the authors site for the current study. However when the model output is adjusted by a machine learning algorithm that regresses observed ozone on the simulated background and anthropogenic ozone fields, much improved agreement (mean absolute bias = 1.5 ppb) emerges, as the graph at right shows. It should be emphasized that this comparison is based on two completely independent analyses.

Given the "apples vs. oranges" aspects of the comparing the overall results from the authors analysis with the observation-based approach, it is unlikely that a definitive comparison can be given in the current manuscript. However, two straight forward comparisons would be informative and should be included:

- First, Parrish et al. (2025) and papers cited therein have shown that local anthropogenic enhancements of surface ozone in North American regions have decreased exponentially with a time constant of 21.8 ± 0.8 years, and they utilize fits to this characteristic temporal change as the basis for quantifying the local anthropogenic enhancement to Ozone Design Values (ODVs). Derwent and Parrish (2022) report similar exponential time constants 18 ± 4 years over the United Kingdom and 37 ± 11 years over continental Europe. Comparison of the present model-derived results for the temporal evolution of the local anthropogenic NOx contribution to those observationally derived results would be quite useful; hence, my continued insistence that the authors include exponential fits to the temporal evolution of the local anthropogenic NOx contribution over both North America and Europe.
- Second, the observation-based studies also show that the background contributions vary in a manner well-described by a quadratic polynomial (Equations 1 and 3 of Parrish et al., 2025) over the period of this study; this polynomial provides a means to calculate the overall change in background concentrations over that period. The authors report linear fits to all NOx-tagged NOx-contributions; these linear trends also provide a means to calculate the overall change in concentrations of all NOx-tagged species over that period. The sum of the changes of all background species (foreign anthro. NOx, natural NOx, and ship NOx) would provide an estimate for the total background O3. This latter quantity should be compared with that from Parrish et al. (2025) (and from other published estimates of observation-based estimates of background ozone over this period, if the authors wish).
- 7) The authors' have added very useful analyses (Section 3.2.2, 3.2.3, 3.3.2 and 3.3.3) that use Fourier analysis to quantify the ozone seasonal cycle and its contributions in both North American and European receptor regions. This analysis and associated discussion are quite informative; however to follow those discussions I had to construct a table that collected material from Tables S1-S9 (see below). I suggest that or a similar table be included in the manuscript. (Tables S1-S9 of the Supplement could then be reduced to a single table

including the results of the 5 year periods for receptor regions not included in this table; tables of the results for individual years are not needed.)

| Region              | y0 (ppb) |       | A1 (ppb) |       | Φ1 (radians) |       | A2 (ppb) |       | Φ2 (radians) |       |
|---------------------|----------|-------|----------|-------|--------------|-------|----------|-------|--------------|-------|
|                     | obs      | model | obs      | model | obs          | model | obs      | model | obs          | model |
| Eastern Canada      | 36.9     | 37.7  | 5.9      | 9.9   | 4.82         | 5.36  | 1.9      | 2.1   | 3.4          | 1.6   |
| NW US               | 40.8     | 44.1  | 5.9      | 7.1   | 5.16         | 5.38  | 1.0      | 1.7   | 2.3          | 1.7   |
| NE US               | 39.5     | 41.5  | 9.3      | 14.9  | 5.24         | 5.53  | 1.6      | 3.1   | 2.9          | 1.4   |
| sw us               | 48.5     | 52.3  | 11.3     | 10.7  | 5.44         | 5.47  | 1.4      | 1.9   | 2.3          | 2.1   |
| SE US               | 41.8     | 44.4  | 8.0      | 11.4  | 5.36         | 5.53  | 3.3      | 3.8   | 2.6          | 2.2   |
| Western Europe      | 35.4     | 34.7  | 8.6      | 11.1  | 5.05         | 5.20  | 1.8      | 1.9   | 3.4          | 3.2   |
| Southern Europe     | 41.2     | 42.4  | 11.6     | 12.4  | 5.39         | 5.45  | 1.8      | 2.7   | 2.2          | 2.3   |
| C&E Europe          | 38.1     | 36.6  | 11.3     | 15.2  | 5.24         | 5.39  | 1.8      | 2.2   | 2.6          | 2.3   |
| SE Europe           | 39.9     | 47.4  | 10.4     | 12.5  | 5.55         | 5.60  | 2.8      | 2.7   | 1.4          | 1.7   |
| NW US 2000-2004     | 41.4     | 43.5  | 6.5      | 9.0   | 5.22         | 5.43  | 1.1      | 1.8   | 2.4          | 1.9   |
| NW US 2014-2018     | 40.6     | 43.9  | 5.2      | 5.3   | 5.18         | 5.33  | 0.3      | 1.1   | 1.9          | 1.3   |
| NE US 2000-2004     | 40.4     | 41.1  | 11.9     | 20.0  | 5.40         | 5.59  | 1.3      | 3.9   | 2.4          | 1.2   |
| NE US 2014-2018     | 38.3     | 41.1  | 6.7      | 9.3   | 5.01         | 5.44  | 2.2      | 2.3   | 3.4          | 1.7   |
| W. Europe 2000-2004 | 35.8     | 34.1  | 10.1     | 13.2  | 5.18         | 5.32  | 1.6      | 1.5   | 3.0          | 2.7   |
| W. Europe 2014-2018 | 35.9     | 35.3  | 8.0      | 9.5   | 5.09         | 5.18  | 1.2      | 1.9   | 3.5          | 3.1   |
| S. Europe 2000-2004 | 40.8     | 42.1  | 13.0     | 15.2  | 5.43         | 5.48  | 1.6      | 2.4   | 2.1          | 2.1   |
| S. Europe 2014-2018 | 41.8     | 42.8  | 10.6     | 9.9   | 5.41         | 5.47  | 1.7      | 2.6   | 2.2          | 2.3   |

Paragraph beginning on line 649: Please more simply and clearly quantify the "tendency for overestimation"; e.g. the 2nd sentence could read: "The model generally captures these mean levels, though with a tendency for overestimation of 0.7 - 2.6 ppb in the eastern and 3.3 - 3.8 ppb in the western regions." The following 2 sentences could then be eliminated, and the final 2 sentences of the paragraph eliminated since they are mostly speculative. The paragraphs beginning on lines 659, 670, and 681 should be reviewed for similar opportunities for simplifying and clarifying the quantification and discussion, and removing speculative statements, unless quantitative analysis is added to support the speculation. (See the next Major Issue in this regard). The final 3 paragraphs of this section compare the Fourier analysis with Figure 8 and give an overall summary; they strike me as largely speculative, without firm quantitative analysis. I suggest shortening and clarification.

8) To more fully inform the readers (and this reviewer) the mathematical definition and the physical significance of  $\varphi_1$ , the phase of the fundamental harmonic (not really of the annual cycle, but close if A2 << A1) must be more fully explained. In the authors' reference (Parrish et al., 2016), the first term included in Fourier Analysis for the fundamental harmonic is (in the authors' notation)  $A1*\sin(\chi+\varphi_1)$ . In this approach, when  $\varphi_1$  is zero, the peak of the fundamental is at  $\pi/2$  radians, which corresponds to 1/4 of the year or roughly the end of March. Importantly, a larger value of  $\varphi_1$  gives an earlier (not later) peak; e.g. if  $\varphi_1=\pi/2$  radians the peak is on January 1. If the authors followed this approach, then their discussion of derived values of  $\varphi_1$  is incorrect, because that discussion assumes a larger value of  $\varphi_1$  gives a later peak. However, I imagine it would be possible to do the Fourier analysis with a negative sign rather than a positive sign in the fundamental term; if the authors followed this approach, then their discussion is correct. A full discussion of the approach actually followed is required, and the discussion corrected if necessary.

If the authors followed the approach of Parrish et al. (2016) then it may be clearer to give values of  $\varphi_1$  after subtracting  $2\pi$ , so that more negative  $\varphi_1$  values correspond to later peaks. This is valid since the phase angle repeats after it advances by  $2\pi$ ,

- 9) In my judgement the most interesting feature of the  $\varphi_1$  values is that for all but one receptor region in North America and Europe, both the modeled and observed  $\varphi_1$  values fall within  $\pm$  0.3 radian (or 17 days) of a mean value of 5.37 (or -0.91) radians, which corresponds to a seasonal maximum of the fundamental on Julian day 144 or May 24. The discussion of this quantity might best further emphasize this close regional and model-observation agreement, before discussing the relatively small differences.
- 10) Line 728: Please specify that Figure S2 shows modeled seasonal cycle envelopes. It would be illuminating to include a similar figure showing observed seasonal cycle envelopes. To my eye, there are evident, but small, seasonal cycle changes, with significant variability about consistent systematic changes. Thus, I would expect difficulty in quantifying the systematic changes, and this difficulty should be carefully considered before making firm conclusions. In this regard, Figure S2 indicates that seasonal changes appear to be clearer and more systematic in NE US compared to SW US. Thus, it may make sense to give a clear, statistically significant analysis of NE US first, and then address the SW US second.
- 11) Section 3.2.3 is well organized, but I think the discussion could be simplified and clarified, and in a few places corrected; in particular:
  - Line 731: Inclusion of parameters of the Fourier analysis of the 5-year averaged periods should be included in a table in the manuscript for the two example regions, as suggested in Major Issue 7).
  - Lines 741-742: Note that the shifts in  $\varphi_1$  of 0.04 and 0.10 radians correspond to shifts of only 2.3 and 5.8 days, respectively quite small shifts. And as noted in Major Issue 8) the peak of the fundamental shifts in the opposite direction from the phase shifts.
  - Lines 773-774: A φ1 shift from 5.40 to 5.01 radians actually indicates a shift of the seasonal maximum by 23 days, but from spring towards summer. Those values correspond to peak value shifting from Julian Day 143 to 165 or May 23 to Jun 14. I do not see how this is consistent with Figure 10. An explanation is required (perhaps in the Supplement) so that the reader can fully follow the discussion. An example showing how the 1st and 2nd harmonics combine to approximate the seasonal cycle in the two 5-year periods in the NE US would be quite helpful to include in the Supplement.

Given the issues identified above, I suggest that this Section be completely rethought, with the concluding paragraph revised as needed.

- 12) I have not attempted to critically review Section 3.3 as carefully as I did Section 3.2. The discussion in these sections is similarly organized for both continents. Please seek to include any manuscript improvements made to the former sections in the latter sections where appropriate. And please similarly review all major and minor comments that refer to Section 3.2 when revising Section 3.3.
- 13) Section 3.4 raises an entirely new area of discussion that raises new questions in my mind, specifically:
- Its introduction is somewhat confusing. I suggest changing the phrase "in these regions" to "in the receptor regions", assuming this is correct.
- The 2nd sentence in the 2nd paragraph is also confusing. "It is noteworthy that this NO2\_FOREIGN, locally recovered from foreign ozone titration, is separately tagged in our modelling system than the NO2 directly flowing from foreign regions (which we do not discuss here)." First, "... separately tagged in our modelling system than ..." is not clear to

me. Second, it raises the question of what exactly is and what is not included in the tagging. NO2 directly flowing from foreign regions is generally considered to be small due to the short lifetime of NOx in the troposphere, but what about PAN and other organic nitrates? They have been considered reservoirs of sequestered NOx that can be transported over intercontinental distances in the free troposphere. However, it is not clear to me how the model treats ozone produced by foreign NOx transported as an organic nitrate to a receptor region, where it produces NOx after release from the reservoir species.

I suggest that the authors remove Section 3.4 from this paper, which is already quite long, and then more fully discuss the NOx-tagging system in the Introduction or Section 2.1 so that the reader is aware of issues such as tagging of NO2 directly flowing from foreign regions, and NOx reservoir species transported from foreign regions.

**Minor Issues:**

- 1) Figures S3 and S4 present scatterplots for the parameter values derived from the Fourier analyses. The derived r values annotated in the figures quantify how well the model reproduces the interannual variability in the parameter values in the respective regions. It would be useful to also give the r value for the entire 95 (North America) or 76 (Europe) set of values; this (generally significantly larger value) would quantify how well the model reproduces both the spatial variability and the interannual variability of the respective parameter throughout all regions on each continent. From inspection of the figures the model performance for some parameters appears to be quite impressive indeed. Note that the caption to Figure S4 should give 76 (not 95) as the number of markers.
- 2) In lines 403 and 413 the y0 parameter is described as representing annual average MDA8 O3 derived from detrended data. However, since the authors derive values for only a single year, no detrending has been performed, and the y0 parameters thus represent actual annual averages, and thus, still include the interannual variability. I suggest removing the references to "detrended" data.
- 3) Unless the authors have a particular reason for including Tables S2-S9, I suggest they be removed, or at least shortened to only the summary values spanning the multi-year periods.
- 4) Line 396: I suggest that the correlation coefficients at the annual average timescale be explicitly stated (i.e., 0.34 to 0.95). Add a similar statement to the paragraph for Europe beginning on line 428 and for the Belarus & Ukraine region on line 451.
- 5) Line 430: Modify final phrase to "..., except SE Europe and RBU."
- 6) Lines 531-32: This statement should be more forcefully stated, something like "... the observed PSO levels consistently exceeded the WHO guideline (31 ppb) throughout the study period by at least 10(?) ppb". Similarly for European regions on line 827.
- 7) Line 532 and elsewhere: When measured or modeled ozone concentrations are compared to the WHO guideline, it should be specified that it is the "WHO long-term guideline" that is being referenced.
- 8) Line 535: Upon the first occurrence of the authors' Quantitative quote of a trend (e.g., (-0.19 (0.01) [-0.32, -0.06] ppb/yr), please define the 4 numbers given.
- 9) Table 2 should include the value of the derived trend (i.e., the most probable value of the trend) even if the 95% confidence intervals include zero. The table would be clarified if the

- column spacing were adjusted so that in each table entry the linear trend appears on the 1st line, p-values (shown in parentheses) on 2nd line, and 95% confidence intervals on 3rd and 4th lines with all negative signs appearing on correct lines.
- 10) Lines 553-57: This sentence requires clarification; it refers to "year-to-year variations in local emissions", which may be taken to indicate interannual variability. However, I certainly expect (and from the discussion the authors seem to agree) that the temporal correlation is largely driven by systematic decreases in local NOx emissions over the 19 year study period.
- 11) Line 568 states that: "... reductions in local NOx emissions translate directly and proportionally to reductions in the ozone ...". It is clear that the translation is direct, but there is no analysis to show that it is proportional (i.e., linearly related). Unless this proportionality can be demonstrated and the proportionality constant quantified, the phrase "and proportionally" should be removed.
- 12) Line 644-645 state "The phase φ₁ indicates the timing of the annual peak, with numerically larger values typically corresponding to a later peak in the year ...." This is not correct; larger phase angle values always correspond to an earlier peak in the year. Please see discussion in Major Point 8).
- 13) Line 646 and elsewhere: "Tables S2-S6" should be "Tables S1-S5".
- 14) Lines 737-738: Discussion could be made more accurate, viz. "The observed annual mean ozone (y0) decreased slightly from 41.4 ppb to 40.6 ppb, while the modeled y0 increased slightly from 43.5 ppb to 43.9 ppb), slightly increasing the positive bias noted earlier."
- 15) In Table 3, certainly only one decimal place in the entries is statistically justified.
- 16) Line 1057: I suggest strengthening perhaps end the sentence with "... in both regions exceeds the long-term WHO guideline by wide margins over the entire study period.

**New references not included in the manuscript under review or in my previous review**

- Mims, C.A., D.D. Parrish, R.G. Derwent, M. Astaneh and I.C. Faloon (2022), A conceptual model of northern midlatitude tropospheric ozone, *Environ. Sci.: Atmos.*, 2, 1303-1313, DOI: 10.1039/d2ea00009a.
- Hosseinpour, F., Kumar, N., Tran, T., and Knipping, E.: Using machine learning to improve the estimate of U.S. background ozone, Atmospheric Environment. 316. 120145, <a href="https://doi.org/10.1016/j.atmosenv.2023.120145">https://doi.org/10.1016/j.atmosenv.2023.120145</a>, 2024.

---

## Author Response (AR2)

Response to the second round of review of the manuscript titled "Explaining trends and changing seasonal cycles of surface ozone in North America and Europe over the 2000-2018 period: A global modelling study with NOX and VOC tagging" by Ansari et al.

MS Number: egusphere-2024-3752

The reviewers' comments are shown in black while our point-by-point responses are shown in blue. All line numbers refer to the track-changes version of the revised manuscript.

**Reviewer 1**

The revised manuscript has incorporated reviewer suggestions and improved the analysis from *the* previous version. The results are presented in a more quantitative way with improved comparison between regions and removed redundancy. The manuscript has now included a Fourier analysis of the seasonal cycle.

I enjoyed reading the manuscript and I would recommend this for publication with one suggestion:

Perhaps the fourier analysis can be introduced in the methods, to give a clear introduction to the different parameters and their meaning. This may help in case the reader needs to refer back to the parameters, and also encourage others to use your analysis method.

We thank the reviewer for once again reviewing our revised manuscript and for their positive comment. We have now included the technical details about the Fourier Transform analysis in the section text S2 of the supplement and have referred the reader to this section in the main manuscript at lines L421-422.

**Reviewer 2**

**Summary:**

This paper is a revision of an earlier manuscript presenting a global modelling study with an innovative dual tagging analysis; the paper's focus is on surface ozone over North America and Europe, particularly with regard to the causes of long-term changes in mean concentrations and the seasonal cycle and its changes. In my view, this manuscript represents a very significant improvement over the previous version, and

addresses all issues that I raised in my first review, although some are still of concern. I recommend acceptance for publication when some remaining issues are addressed.

We once again thank the reviewer for seriously engaging with our work and providing detailed comments and feedback. We are glad that the reviewer appreciates the time and effort we put in in addressing their concerns in the first round of review and that they consider the updated manuscript as a very significant improvement.

Below I review the responses to the critical issues I identified in my previous review, discuss several remaining and new major issues, and list a few new minor issues.

**Review of Critical Issues previously raised:**

Section 3.1 is devoted to evaluation of the CAM-Chem model used in the chemical-transport simulations. The more robust evaluation now included in the paper is a significant improvement. I believe this section now gives a more comprehensive evaluation of the model performance, an approach that should be emulated by other studies presenting global modeling results.

Thank you for the positive comment. We agree that this is a more robust method of model-observation comparison, with less scope for visual misinterpretation.

Nevertheless, a few lingering issues remain, specifically:

Line 378 suggests that "... all HTAP tier 2 receptor regions for North America" were used for model evaluation. However, only 5 are included, while Fig. 3 of their reference (Galmarini et al., 2017) identifies 6 – Western Canada is excluded. I believe that this (at least below 60 deg N) is a particularly important region to include, both in the model evaluation and in the later discussion of results, since it provides a useful contrast with the more urbanized NW US and SW US inflow receptor regions lying to the south. Further, it may provide insight into reasons for the model overshoot of the maxima and undershoot of the minima for the earlier years of 2000–2006 in Eastern Canada. Please provide the comparison for Western Canada, or fully discuss the reason(s) for its being excluded. Importantly, please note that rural data are indeed available from this region – see Fig. 5 of Galmarini et al. (2017) and the figure that Owen R. Cooper included in his comment on this manuscript (<a href="https://doi.org/10.5194/egusphere-2024-3752-CC1">https://doi.org/10.5194/egusphere-2024-3752-CC1</a>).

The model evaluation in this study is performed against a gridded dataset derived from surface observations that are part of the TOAR-II database. This gridded dataset is ultimately based on the MDA8 O3 values reported only from rural stations. The rural/urban classification relies on the *type\_of\_area* station metadata from the original data providers. Unfortunately, we do not have any stations in Western Canada where

this metadata is included from the data providers and therefore they remain unclassified and are not included in the TOAR-II gridded dataset. Due to this reason, we do not discuss the Western Canada receptor region in the main text. However, given the importance of this region for background ozone studies, as pointed out by the reviewer, we have now included the modelled PSO time series with tagged contributions as well as the MDA8 O3 seasonal cycle change between the initial (2000–2004) and recent (2014–2018) periods for this region in the supplement (see Figures S7 and S8). These plots are based on the full sampling of the receptor region rather than co-sampling with TOAR observations due to their unavailability. We have now acknowledged this caveat in the manuscript in lines 396–401 and have also stressed the importance of including station metadata which allows selective evaluation in different contexts.

In my previous review I raised a subtle issue associated with model-observation comparisons such as the authors present in Section 3.1 and utilized in many such comparisons in the published literature, i.e., a degree of circular reasoning that results from models developed to agree with observations, so that such agreement cannot be taken as independent confirmation that models perform properly for the correct reasons. In their response to the reviews, the authors acknowledge a general concern with this issue, which they discuss in detail that generally discounts any large effects on their comparison. Some of this discussion is persuasive, but I could dispute other parts in further discussion; for example, nearly all persuasive model-measurement comparisons, whether for purposes of model "tuning" or evaluation of results, focus on metrics that are less sensitive to "nighttime ozone and avoid any large nighttime biases, which often arise due to improper simulation of the nighttime boundary layer". Rather than push this issue further, I suggest that the authors simply describe this issue and note their concern in one or two sentences in Section 3.1, perhaps at the end of the paragraph beginning on line 368.

We have now added the following text in lines L514-517.

"We note that agreement between models and observations does not in itself demonstrate that the models represent all processes correctly, since models are necessarily simplified representations of reality and can reproduce certain features for the "wrong" reasons. As Box (1976) succinctly put it, "all models are wrong, but some are useful"; our comparisons should therefore be viewed in this light."

I commend the authors for their illuminating discussion of the treatment of ship emissions in chemical transport models, now included in Section 3.1. In their response, the authors do attempt the first zeroth-order check that I suggested, and note a problem in comparing emissions in units of moles NOx/month with monthly average ozone concentrations in units of moles. However, the numbers derived in their comparison for July 2018 would be correct if the total global O3 attributed to ship NOx had a lifetime relative to total gross destruction of one month. This lifetime in the

summertime northern mid-latitudes is likely shorter than 1 month. For example, in July in the marine boundary layer (where the ozone derived from ship emissions is primarily formed) that lifetime is estimated to be  $\sim$ 10 days (see discussion in Mims et al., 2022 and their Figures S4 and S5). However, some of the ship derived ozone is transported to the cooler and drier free troposphere before destruction, so the effective overall lifetime for ship-derived ozone in summer at northern mid-latitudes might be best estimated as  $\frac{1}{2}$  month; thus this first zeroth-order check would indicate a model overestimate of ship derived O3 on the order of a factor of (7.04/4.25\*2=)3.3. The authors also perform the second zeroth-order check that I suggested, and have included a discussion of this issue in Section 3.1 and the Supplement. Overall, I judge these responses to be an adequate response to this issue; however, I suggest inclusion of a discussion of the first zeroth-order check in the Supplement for interested readers to peruse, and in discussion of ship emissions in the paper, the authors should consider a likely overestimate of ship emissions by this factor.

We thank the reviewer for their insightful comment. Based on their suggestion, we have calculated an ship-tagged inferred ozone production rate by using an ozone lifetime of 0.5 months which yields a potential overestimation of ozone production from ships by a factor of 3.3. We have now included the details of our attempt at performing the first-order sanity check, its issues, and the potential fix as suggested by the reviewer, in Text S1 in the supplement in lines L17-62. We have also mentioned this in the manuscript in lines L481-485: "We also performed a zero-order sanity check by comparing the inferred ozone production rate from ship NOx within the marine boundary layer of the northern hemisphere midlatitude region in the model with observational values. We found a potential overproduction of ozone by ships in the model by a factor of 3.3 when compared to the data from previous observational studies. We refer the reader to Text S1 in the supplement for a detailed discussion on these calculations."

• I am pleased that the issues of possible corruption of observational data have been resolved.

We once again thank the reviewer for pointing at the anomalous drop in stations which led us to track down this technical problem with data retrieval scripts. We are glad that it is sorted now.

• The discussion in Sections 3.2 and 3.3 now include more systematic and quantitative analysis approaches; it is now much improved, with the new Figures 5, 7, 11, and 13 adding significant additional information and clarity. However, several issues remain as detailed in many of the Major Issues below.

We are glad that the reviewer acknowledges the newer approach adopted for analyses of seasonal cycles and the additional spatial analyses based on local anthropogenic NOx

contribution to PSO in North America and Europe. We have addressed many of the newly raised *major issues* in the subsequent section.

• Section 4. Conclusion, Limitations and Future Outlook now provides a clear and reasonably concise summary of the new understanding of the atmosphere that has emerged from this study.

Thank you for the positive comment.

**Major Issues:**

1) I note that the current manuscript contains 17 figures composed of nearly 110 separate graphs, most with multiple traces that tend to overwhelm the reader. Some comments below suggest combining or revising manuscript figures and/or moving figures to the Supplement. Any changes the authors can make in this regard would improve the paper.

The increase in the number of figure panels is a direct consequence of the reviewers' earlier requests. For example, the figures showing PSO for North America and Europe (Figures 6 and 12) now have an extra column of panels to accommodate the reviewer's request to shorten the y-axis scales on the tagged contributions time series. We believe that the new figures look good and present the results in a clear manner. The addition of new spatial maps showing PSO-relevant local anthropogenic NOx emissions alongside their contribution to PSO and the total PSO for both continents (figures 5 and 11) were added based on reviewer 1's comments - and they seem to be satisfied with them. The second reviewer has also acknowledged in their earlier comment that they add "significant additional information and clarity". Similarly, the new figures showing the correlations between NOx emissions versus their contribution to PSO on a gridcell level for both continents (figures 7 and 13) were added based on this reviewer's request. This was a significant amount of added effort and as the reviewer notes, adds significant additional information and clarity, which we agree with. Therefore, we have decided against moving some of these figures into the supplement (which already has plenty of figures for other receptor regions).

2) In my first review I noted that the modeled contribution of local anthropogenic NOx to PSO decreases along western North America from the Southwestern US (with many large urban areas) to the Northwestern US (with few large urban areas); in this regard, I suggested that it would be useful to include the Western Canada receptor region in the analysis in Section 3.2 Ozone in North America. This inclusion would extend the western North American contrast to a region without large urban areas. The authors responded "We have not included discussion for Eastern Canada due to the unavailability of TOAR-II data from rural stations in this region which prevents model

evaluation and co-sampling for this region." This response is inaccurate; first, my comment referred to Western, not Eastern, Canada, and second, in fact there are TOAR-II data from rural stations in Western (as well as Eastern) Canada as I note in discussion above, and as is readily apparent in the authors' Figures 5 and 7 which include results for that Western Canada region. In my view, this western North American contrast is one of the more exciting prospects for improved understanding of surface ozone that could emerge from this manuscript, and should be exploited as fully as possible. It may help illuminate the comparison between the present work and recent observation-based studies that the authors discuss in the paragraph beginning on line 608. Please provide and fully discuss the model calculations for Western Canada as a 6th North American receptor region, or fully discuss the reason(s) for its exclusion.

We indeed made a typographical mistake in our earlier comment: we meant "Western" and not "Eastern" Canada. The reviewer is right in noting that results are available over this region as shown in Figures 5 and 7, however these are only modelled results and emissions, not observations. We do not discuss Western Canada in our study because we did not find any station categorized as rural over this region in the TOAR-II database. We would also like to point out that while this study stands on its own feet independently, it is submitted to a TOAR community special issue of ACP and one of its aims is to validate the TOAR-II observational database in its current form. The TOAR-II database relies on the type\_of\_area metadata from the original data providers and many of the data providers did not include this information, which led to only a single rural station in Western Canada which is so close to the US border that it gets engulfed into the NW US receptor region in the gridded observational dataset (see here: https://toar-data.fz-juelich.de/gui/v2/dashboard/?zoom=3&center=48.71,-87.14&count ry=CA&type\_of\_area=Rural&variable\_id=5&data\_start\_date=2000-01-01T00:00&data\_ end\_date=2018-12-31T11:59). We agree that there must be more rural stations in Western Canada but we did not have the appropriate metadata at the time of performing the study.

However, to address the reviewer's concerns, we have now added new figures for PSO and seasonal cycle change along with tagged contributions for Western Canada in the supplement; see figures S7 and S8. Here, due to lack of TOAR observations, the entire receptor region was sampled to produce the regional averaged metrics. We hope that these plots can provide value for those interested in understanding the contrast between Western Canada and Western US regions especially in terms of the role of background ozone.

3) I suggest that Figure 6 be replaced with simpler, more informative figures.

For discussion of observed and model-simulated total PSO (paragraphs beginning on lines 524 and 534) I suggest replacing the 5 graphs a, d, g, j, m and the potential 6th

graph for Western Canada with a two graph figure – one with all observed and the other with all model-simulated total PSO time series. Include the linear fits in both panels, and expand the ordinate to just the PSO range spanned by the time series (35 to 65 ppb?). For these graphs linear fits are appropriate, as the derived slopes quantify the scientifically interesting quantity of average PSO trends over the 6 receptor regions during the 2000–2018 period. Such a figure will allow improved visual comparisons of a) regional differences in PSO within the same graph, b) observed and model-simulated total PSO in side-by-side graphs, and c) the quality of the linear fits superimposed on the fitted data in each graph.

Here, the reviewer is asking us to override their previous request (which was to modify the aspect ratio of the panels to a nearly square format, and to make appropriate changes in the graph to better show the tagged contributions time series, rather than completely getting rid of the tagged contributions), which we have already accommodated in the first revised version of the manuscript. We separated the total PSO time series from the tagged contributions panel which allowed us to curtail the y-axis to a lower value thereby allowing better representation of the various tagged contributions (we did not change the y-axis to a log-scale but now we have separately fitted exponential functions to the local NOx contributions to PSO and reported them in Table S10). Now, the reviewer is asking us to only retain two panels: one with all observed and the other with all modelled PSO time series for all receptor regions. This approach will seriously compromise the reader's ability to view the trend in total PSO as a sum of trends in various NOx- and VOC-tagged contributions for different regions, which is a key result made possible by our TOAST 1.0 tagging methodology (as also mentioned in lines L237-238) that we want to communicate through this figure.

For discussion of the local anthropogenic NOx contribution (paragraph beginning on line 541) I suggest a figure with a single graph illustrating the time series from all 6 North American receptor regions. The 0 to 40 ppb ordinate range in the present Figure 6 should be retained. However, here I (again) suggest that the time series of this contribution in each receptor region be fit to an exponential function, rather than the linear analysis the authors currently employ. The rationale for this suggestion is discussed fully below in the 4th Major Issue. Such a figure will allow improved visual comparisons of a) regional differences in this PSO component within the same graph, and b) the quality of the exponential fits superimposed on the fitted data.

We have decided against merging the results of different receptor regions into a single graph for the same reasons as noted above – and therefore retained the figure in its previous (first revised) form. However, we have addressed the reviewer's concern of fitting exponential functions to the local NOx contribution to PSO for each receptor region. We have presented these results in the supplement (Figure S16 and Table S10) and have added a few lines of text discussing these in the main manuscript (lines L593–602).

"Several previous observational-based studies have inferred the magnitude and temporal decline of local contributions to ozone in North America based on curve fitting the observed ozone time series data and have reported these magnitudes and e-folding times of the local ozone enhancements for various stations and regions (Parrish & Ennis, 2019; Derwent & Parrish, 2022; Parrish et al., 2025 among others). In order to facilitate a comparison with these observational studies, we also fitted an exponential function of the form shown in eq. 1 to our model-derived local anthropogenic NOx contributions to PSO for various receptor regions (see Figure S16) and have tabulated the derived e-folding times against those found in literature (see Table S10). Here, A represents the magnitude of local NOx contribution to PSO for the initial year (2000) in ppb and  $\tau$  represents the e-folding time of these contributions. We find  $\tau$  = ~25-38 years from the model and ~22 years from the literature for various US receptor regions.

$$y = A \exp(-\frac{1}{\tau}t) \dots (1)$$
"

For discussion of other NOx tagged contributions and all VOC tagged contributions, the time series plots in the current Figure 6 provide little information, and the discussion of their trends is based on the derived linear trends from Table 2, while the plots are not directly discussed in the relevant paragraphs (lines 574-607). Thus I suggest those plots be moved to the Supplement.

I suggest a similar treatment of Figure 12.

It is possible that on a first read it may appear to the reviewer that the discussion of results is purely based on the derived linear trends from Table 2. However, in fact, the first author of this manuscript wrote the description entirely while looking at Figure 6 and only inserted the linear trends within the text later to comply with the TOAR guidelines. So, we can only say that the assertion that plots are not directly discussed in the paragraphs is highly subjective and based on the reviewer's unique perspective and interpretation. We find it unreasonable to shift the time series of NOx and VOC-tagged contributions to PSO – a unique result from our modelling technique – to be relegated to the supplement.

The same arguments apply for Figure 12. However, we have also fit exponential functions to the local anthropogenic NOx contribution to PSO for Europe (Figure S16 and Table S10).

4) In the paragraph beginning on line 534 the authors discuss linear trends in PSO quantified by means of linear fits to time series. They note regional differences in the trends derived from the observations, and comment that the "model generally captures these decreasing trends and the interannual variability reasonably well, though with some regional differences in magnitude ...." Additional discussion should quantify what is meant by "reasonably well" (i.e., quantify differences between modeled and observed

trends and provide correlation coefficient between them), and discuss likely reasons for regional differences in trends (e.g., are larger trends found in regions with larger local NOx emissions?).

We have now computed and reported the correlation coefficients between observed and modelled PSO values for all receptor regions considered along with the differences between observed and modelled PSO trends. We have added the following text in lines L553-556 in the manuscript: "r-values between observed and modelled PSO are 0.89, 0.78, 0.89, 0.93, 0.93, and 0.95 and the difference in modelled and observed trends are -0.09 ppb/yr, -0.02 ppb/yr, 0.07 ppb/yr, -0.16 ppb/yr, and -0.17 ppb/yr for E Canada, NW US, SW US, NE US, and SE US, respectively. These regional differences in PSO trends are driven by regionally different local and remote contributions to PSO as revealed in Figure 6."

We have also added the following text for European PSO at lines L871-873: "The model generally also captures the interannual variability in PSO for Western Europe and C&E Europe successfully (r=0.75 and 0.69 respectively) and to a lesser extent in Southern Europe (r=0.37; Figure 12)."

We have also updated Figures 6 and 12 where correlation coefficients (r values) have been printed over the left panels for each receptor region.

5) I must strongly argue that the most interesting science question to address with regard to the local anthropogenic NOx contribution to PSO is quantification of the average relative rate of decrease. A particular fractional decrease in local anthropogenic NOx emissions is expected to give a larger absolute decrease in a region of large emissions compared to a region of small emissions; however the relative decrease may well be similar in the two regions. Similar relative rates of decrease of local anthropogenic NOx contributions to PSO would suggest similar fractional emission reductions and similar photochemical environments in the different regions, even if the absolute emissions differ, and thereby causing the absolute local anthropogenic NOx contribution to PSO to differ between those regions. An exponential fit to a time series of a quantity provides a means to quantify the average relative rate of decrease of that quantity. This is the rationale for my strong recommendation that the time series of the local anthropogenic NOx contribution be fit to an exponential function, since it gives a quantification that is of much greater scientific interest than the quantification of the absolute rate of decrease provided by a linear fit. Alternatively, the authors could employ a different technique to quantify the average relative rate of decrease.

To address the reviewer's request, we have now fitted exponential functions of the form  $y = A \exp\left(-\frac{1}{\tau}t\right)$  to our model-derived (i.e. tagged) local anthropogenic NOx contribution to regional PSO for all the nine receptor regions considered in this study.

These fits are presented in Figure S16 in the supplement along with the fit parameters **A** (initial year contribution on the curve, in ppb) and  $\tau$  (e-folding time, in years). We have also included an additional table in the supplement Table S10 wherein we have compared the model-derived  $\tau$  values from this study to the observations-derived  $\tau$  values in previously published studies. We find a broad agreement in  $\tau$  values from this study and published literature, i.e., smaller  $\tau$  values for North American regions (~25-38 years from the model and ~22 years from observations) suggesting a relatively faster decline and larger values for European regions (~37-63 years from the model and ~37-44 years from observations) suggesting a slower decline in local contributions to ozone in Europe.

We have added extra text describing this in the manuscript in section 3.2.1 in lines L571-580:

"Several previous observational-based studies have inferred the magnitude and temporal decline of local contributions to ozone in North America based on curve fitting the observed ozone time series data and have reported these magnitudes and e-folding times of the local ozone enhancements for various stations and regions (Parrish & Ennis, 2019; Derwent & Parrish, 2022; Parrish et al., 2025 among others). In order to facilitate a comparison with these observational studies, we also fitted an exponential function of the form shown in eq. 1 to our model-derived local anthropogenic NOx contributions to PSO for various receptor regions (see Figure S16) and have tabulated the derived e-folding times against those found in literature (see Table S10). Here, A represents the magnitude of local NOx contribution to PSO for the initial year (2000) in ppb and  $\tau$  represents the e-folding time of these contributions. We find  $\tau = \sim 25-38$  years from the model and  $\sim 22$  years from the literature for various US receptor regions.

$$y = A \exp(-1/\tau t) \dots (1)$$
"

**And in section 3.3.1 in lines L896-901:**

"As for North America, we have also fitted exponential curves (based on eq. 1) to the local anthropogenic NOx contributions to PSO in European regions in order to facilitate the comparison of the e-folding time ( $\tau$ ) with observationally-derived values in published literature (see Figure S16 and Table S10). We find a broad agreement with the observationally-derived values in that they are larger than those for North America (~37-63 years from the model and ~37-44 years from observations), suggesting a relatively slower decline in local contributions in Europe."

6) The authors' have added a very useful paragraph (beginning on line 609) that compares their results to the recent observation-based studies of Parrish et al. (2025). That comparison focuses on the apparent difference between the relatively small (<6 ppb in recent years) local anthropogenic enhancement to Ozone Design Values (ODVs) found in the observation-based study and the larger (~16 ppb) local anthropogenic NOx

contribution to average PSO found in the authors' model-based analysis. Despite some differences (discussed by the authors) between the two quantities derived to characterize the local anthropogenic contribution, it is clear that the authors have identified an important quantitative difference between the results of the two analyses.

We agree that this is an interesting and important observation and thank the reviewer for pointing us earlier to look deeper into such comparisons.

The authors suggest that perhaps the fundamental cause of the difference in results is that the observational-based method may systematically underestimate the full impact of local anthropogenic emissions, thereby overestimating the background contribution. However, they do not provide robust, quantitative analysis to support this suggestion. Importantly, Parrish et al. (2025) and references therein thoroughly discuss multiple lines of quantitative analysis to demonstrate that their observation-based approach does provide quantitatively accurate estimates of the ODV contributions from background ozone (US background ODV) and the local anthropogenic enhancement to ODVs.

Parrish et al. (2025) and references therein rely on deriving long-term trends for US background ozone which they describe by fitting a parabolic curve of the form a + bt + ct2 over long-term historical time-series data (1987-2018) and an additional exponential term of the form  $\mathbf{A} \exp(-\mathbf{t}/\tau)$  to describe the changes in local anthropogenic enhancements. The idea of fitting the parabolic curve finds justification from the results of Parrish et al. (2020) where ozone data from only "background" (west-coastal, MBL, alpine) stations is considered. This methodology implicitly assumes that the local anthropogenic contribution will not be present in the ozone recorded at these sites. However, the local anthropogenic emissions from North America and Europe can be expected to contribute to some fraction of this background ozone too. To ascertain this, we sampled the PSO and the contribution of local anthropogenic NOx to PSO over the model grid cells corresponding to Trinidad Head and Mace Head stations. We have tabulated these values in Table S11. Indeed we found significant local (i.e., North American for Trinidad Head and European for Mace Head) anthropogenic NOx contributions to PSO sampled at these so-called background sites. In recent years (2014-2018) we find the local contribution to PSO at Trinidad Head to range between 4-6.6 ppb. Counting this portion of ozone as background would inevitably lead to an overestimation of background ozone and an underestimation of local enhancement. If this portion of ozone is added to the statistically-inferred local enhancement, it would bring the values reported in Parrish et al. (2025) much closer to our tagging-based results (i.e., around 12ppb vs 16 ppb).

We have now added this quantitative evidence in the manuscript text at lines L656: "To ascertain this claim, we sampled the model output from the grid cells corresponding to these background stations (Trinidad Head for North America and Mace Head for Europe)

and calculated the site-specific PSO and local anthropogenic NOx contributions to PSO. These are reported in table S11 in the supplement. As expected, we found that a significant portion of PSO at these background sites contains contributions from local NOx emissions. For 2014-2018, we find the local contribution to PSO at Trinidad Head grid cell to be 4.0-6.6 ppb, which if added to the statistically-inferred local enhancement in SW US by Parrish et al. (2025) (6 ppb) would bring their values much closer to our findings (16 ppb)."

For example, there is strong evidence that zeroing out anthropogenic emissions in at least some global modeling systems lead to underestimates of the US background ODV; Parrish et al. (2025) discuss the work of Hosseinpour et al. (2024), who report the 4th highest US background (USB) MDA8 ozone concentration across the US from a global modelling system (CMAQ-CAMx) different from that used by the authors. The graph at left below shows that the model gives results that are biased (mean absolute bias = 11 ppb) from those of Parrish et al. (2025), much as the authors site for the current study. However when the model output is adjusted by a machine learning algorithm that regresses observed ozone on the simulated background and anthropogenic ozone fields, much improved agreement (mean absolute bias = 1.5 ppb) emerges, as the graph at right shows. It should be emphasized that this comparison is based on two completely independent analyses.

Given the "apples vs. oranges" aspects of comparing the overall results from the authors analysis with the observation-based approach, it is unlikely that a definitive comparison can be given in the current manuscript.

It is interesting to note this raw as well as bias-corrected ozone attribution result from a different modelling system. However, this example cited by the reviewer pertains to a

different kind of attribution system wherein local anthropogenic emissions are zeroed-out to infer the background concentrations. Such a large emission perturbation, in effect, fundamentally changes many aspects of the atmospheric chemistry within the model which no longer reflects real-world conditions where all emission sources are at play. Such perturbation-based estimates often lead to underestimation of background concentrations (see, Prather 2007; Grewe et al., 2010; Ansari et al. 2019 for further discussions). Here, in our tagging-based system, there are no perturbations to emissions and all emission sources are active at their realistic magnitudes when ozone formed from different sources is separately tagged and tracked within a non-perturbed, realistic, chemical environment reflecting real-world atmospheric conditions. We have already discussed this in the Introduction section (lines L111-127). A part of the "underestimation" of US background ozone in Hosseinpour et al. (2024) can be attributed to these nonlinear changes in atmospheric chemistry but the rest of it could also be realistic (i.e., the modelled background ozone isn't underestimated but rather the observationally-derived background ozone is overestimated as demonstrated in our previous response).

However, two straight forward comparisons would be informative and should be included:

• First, Parrish et al. (2025) and papers cited therein have shown that local anthropogenic enhancements of surface ozone in North American regions have decreased exponentially with a time constant of 21.8 ± 0.8 years, and they utilize fits to this characteristic temporal change as the basis for quantifying the local anthropogenic enhancement to Ozone Design Values (ODVs). Derwent and Parrish (2022) report similar exponential time constants 18 ± 4 years over the United Kingdom and 37 ± 11 years over continental Europe. Comparison of the present model-derived results for the temporal evolution of the local anthropogenic NOx contribution to those observationally derived results would be quite useful; hence, my continued insistence that the authors include exponential fits to the temporal evolution of the local anthropogenic NOx contribution over both North America and Europe.

As discussed in the previous response to major comment 5, we have now fitted exponential functions to our model-derived local anthropogenic NOx contributions to PSO time series for all the nine receptor regions considered in this study. These are shown in Figure S16 and Table S10.

• Second, the observation-based studies also show that the background contributions vary in a manner well-described by a quadratic polynomial (Equations 1 and 3 of Parrish et al., 2025) over the period of this study; this polynomial provides a means to calculate the overall change in background concentrations over that period. The authors report linear fits to all NOx-tagged NOx-contributions; these linear trends also provide a

means to calculate the overall change in concentrations of all NOx-tagged species over that period. The sum of the changes of all background species (foreign anthro. NOx, natural NOx, and ship NOx) would provide an estimate for the total background O3. This latter quantity should be compared with that from Parrish et al. (2025) (and from other published estimates of observation-based estimates of background ozone over this period, if the authors wish).

We have now tabulated the background contributions to PSO (i.e., Foreign anthropogenic NOx, natural NOx, and ship NOx contributions and their sum) for SW US in Table S12 in the supplement. Similar to Parrish et al. (2025) and previous observational studies we find a steady increase in background contributions to PSO in this region. We also fitted a quadratic curve of the form  $a + bt + ct^2$  to the background contributions similar to Parrish et al. (2025). This curve fitting leads to parameter values of  $a=26.6\pm0.94$  ppb,  $b=0.35\pm0.24$  ppb/yr, and  $c=0.0005\pm0.013$  ppb/yr2.

We have added the following lines in the manuscript at lines L662-665: "To facilitate better comparison with previous observational studies, we have also fitted a quadratic curve of the form a + bt + ct2, where t represents time in years starting at 2000, similar to Parrish et al. (2025), to the background contribution (sum of foreign anthropogenic NOx, natural NOx, and shipping NOx) to PSO for SW US (see table S12 in the supplement). We obtain parameter values of  $a=26.6\pm0.94$  ppb,  $b=0.35\pm0.24$  ppb/yr, and  $c=0.0005\pm0.013$  ppb/yr² as compared to observationally derived values of  $a=71.5\pm0.8$  ppb,  $b=0.07\pm0.13$  ppb/yr, and  $c=-0.015\pm0.005$  ppb/yr² in Parrish et al. (2025)"

7) The authors' have added very useful analyses (Section 3.2.2, 3.2.3, 3.3.2 and 3.3.3) that use Fourier analysis to quantify the ozone seasonal cycle and its contributions in both North American and European receptor regions. This analysis and associated discussion are quite informative; however to follow those discussions I had to construct a table that collected material from Tables S1-S9 (see below). I suggest that or a similar table be included in the manuscript. (Tables S1-S9 of the Supplement could then be reduced to a single table including the results of the 5 year periods for receptor regions not included in this table; tables of the results for individual years are not needed.)

| Region              | y0 (ppb) |       | A1 (ppb) |       | Φ1 (radians) |       | A2 (ppb) |       | Φ2 (radians) |       |
|---------------------|----------|-------|----------|-------|--------------|-------|----------|-------|--------------|-------|
|                     | obs      | model | obs      | model | obs          | model | obs      | model | obs          | model |
| Eastern Canada      | 36.9     | 37.7  | 5.9      | 9.9   | 4.82         | 5.36  | 1.9      | 2.1   | 3.4          | 1.6   |
| NW US               | 40.8     | 44.1  | 5.9      | 7.1   | 5.16         | 5.38  | 1.0      | 1.7   | 2.3          | 1.7   |
| NE US               | 39.5     | 41.5  | 9.3      | 14.9  | 5.24         | 5.53  | 1.6      | 3.1   | 2.9          | 1.4   |
| sw us               | 48.5     | 52.3  | 11.3     | 10.7  | 5.44         | 5.47  | 1.4      | 1.9   | 2.3          | 2.1   |
| SE US               | 41.8     | 44.4  | 8.0      | 11.4  | 5.36         | 5.53  | 3.3      | 3.8   | 2.6          | 2.2   |
| Western Europe      | 35.4     | 34.7  | 8.6      | 11.1  | 5.05         | 5.20  | 1.8      | 1.9   | 3.4          | 3.2   |
| Southern Europe     | 41.2     | 42.4  | 11.6     | 12.4  | 5.39         | 5.45  | 1.8      | 2.7   | 2.2          | 2.3   |
| C&E Europe          | 38.1     | 36.6  | 11.3     | 15.2  | 5.24         | 5.39  | 1.8      | 2.2   | 2.6          | 2.3   |
| SE Europe           | 39.9     | 47.4  | 10.4     | 12.5  | 5.55         | 5.60  | 2.8      | 2.7   | 1.4          | 1.7   |
| NW US 2000-2004     | 41.4     | 43.5  | 6.5      | 9.0   | 5.22         | 5.43  | 1.1      | 1.8   | 2.4          | 1.9   |
| NW US 2014-2018     | 40.6     | 43.9  | 5.2      | 5.3   | 5.18         | 5.33  | 0.3      | 1.1   | 1.9          | 1.3   |
| NE US 2000-2004     | 40.4     | 41.1  | 11.9     | 20.0  | 5.40         | 5.59  | 1.3      | 3.9   | 2.4          | 1.2   |
| NE US 2014-2018     | 38.3     | 41.1  | 6.7      | 9.3   | 5.01         | 5.44  | 2.2      | 2.3   | 3.4          | 1.7   |
| W. Europe 2000-2004 | 35.8     | 34.1  | 10.1     | 13.2  | 5.18         | 5.32  | 1.6      | 1.5   | 3.0          | 2.7   |
| W. Europe 2014-2018 | 35.9     | 35.3  | 8.0      | 9.5   | 5.09         | 5.18  | 1.2      | 1.9   | 3.5          | 3.1   |
| S. Europe 2000-2004 | 40.8     | 42.1  | 13.0     | 15.2  | 5.43         | 5.48  | 1.6      | 2.4   | 2.1          | 2.1   |
| S. Europe 2014-2018 | 41.8     | 42.8  | 10.6     | 9.9   | 5.41         | 5.47  | 1.7      | 2.6   | 2.2          | 2.3   |

We thank the reviewer for their positive comment. We have now included a more concise table within the main manuscript (Table 2) which includes the Fourier parameters for the 9 receptor regions for 19-year and 5-year averaged seasonal cycles.

Paragraph beginning on line 649: Please more simply and clearly quantify the "tendency for overestimation"; e.g. the 2nd sentence could read: "The model generally captures these mean levels, though with a tendency for overestimation of 0.7 - 2.6 ppb in the eastern and 3.3 - 3.8 ppb in the western regions." The following 2 sentences could then be eliminated, and the final 2 sentences of the paragraph eliminated since they are mostly speculative.

We have now shortened this paragraph and used the sentence suggested by the reviewer at line L693. However, we have not removed the subsequent statements as we believe they can be very valuable for further analyses in future studies.

The paragraphs beginning on lines 659, 670, and 681 should be reviewed for similar opportunities for simplifying and clarifying the quantification and discussion, and removing speculative statements, unless quantitative analysis is added to support the speculation. (See the next Major Issue in this regard). The final 3 paragraphs of this section compare the Fourier analysis with Figure 8 and give an overall summary; they strike me as largely speculative, without firm quantitative analysis. I suggest shortening and clarification.

We have removed and softened some of these supposedly speculative statements and mentioned the need for further assessment via perturbation experiments which could be done in future studies. However, more fundamentally, we believe it's important to point the reader to the features in the tagged contributions when interpreting the Fourier parameters. We also performed some arbitrary tests, for example, by artificially lowering local contribution numbers in some regions which modified the shape of the simulated ozone seasonal cycle and affected the resulting fourier parameters which confirmed many of our suggestive statements in the text. However, we did not include these little arbitrary tests in the text because we believe that these suggestive statements currently included in the text aren't very far-fetched and should be obvious to the reader. We have made similar modifications in section 3.3.2 for European seasonal cycle discussion.

8) To more fully inform the readers (and this reviewer) the mathematical definition and the physical significance of  $\varphi_1$ , the phase of the fundamental harmonic (not really of the annual cycle, but close if A2 < <A1) must be more fully explained. In the authors' reference (Parrish et al., 2016), the first term included in Fourier Analysis for the fundamental harmonic is (in the authors' notation) A1\*sin(c +  $\varphi_1$ ). In this approach, when  $\varphi_1$  is zero, the peak of the fundamental is at p/2 radians, which corresponds to ¼ of the year or roughly the end of March. Importantly, a larger value of  $\varphi_1$  gives an earlier (not later) peak; e.g. if  $\varphi_1$  = p/2 radians the peak is on January 1. If the authors followed this approach, then their discussion of derived values of  $\varphi_1$  is incorrect, because that discussion assumes a larger value of  $\varphi_1$  gives a later peak. However, I imagine it would be possible to do the Fourier analysis with a negative sign rather than a positive sign in the fundamental term; if the authors followed this approach, then their discussion is correct. A full discussion of the approach actually followed is required, and the discussion corrected if necessary.

If the authors followed the approach of Parrish et al. (2016) then it may be clearer to give values of  $\varphi_1$  after subtracting 2p, so that more negative  $\varphi_1$  values correspond to later peaks. This is valid since the phase angle repeats after it advances by 2p,

We thank the reviewer for pointing out this inconsistency with the phase angles. We had utilized the *fourier\_info* function of the NCAR Command Language (NCL) to perform the fourier transforms for the seasonal cycles (https://www.ncl.ucar.edu/Document/Functions/Built-in/fourier\_info.shtml). On a closer look, we found that this function returns the phases not in terms of phase angles in radians but in terms of the actual abscissa (x-coordinate) where the peaks occur for each harmonic. In our case, the supplied time series are of length 12 each, representing the 12 months, and therefore the phases can be reported within the range of 0-12 months. For example, a phase of 5.0 would mean the end of May while a phase of 2.5 would mean mid-March. We have now clarified this in the manuscript and added a new text section in the supplement Test S2. We have also revised the figures and tables to replace  $\varphi 1$  and  $\varphi 2$  with p1 and p2 respectively which are expressed in months rather than radians.

9) In my judgement the most interesting feature of the  $\varphi_1$  values is that for all but one receptor region in North America and Europe, both the modeled and observed  $\varphi_1$  values fall within  $\pm$  0.3 radian (or 17 days) of a mean value of 5.37 (or -0.91) radians, which corresponds to a seasonal maximum of the fundamental on Julian day 144 or May 24. The discussion of this quantity might best further emphasize this close regional and model-observation agreement, before discussing the relatively small differences.

With the updated definition of the phases, this broader point about model-observation agreement is still true but it now means that the agreement is even closer, within  $\pm$  0.3 months, i.e., 9 days of a mean value of 5.37 months or June 12th.

10) Line 728: Please specify that Figure S2 shows modeled seasonal cycle envelopes. It would be illuminating to include a similar figure showing observed seasonal cycle envelopes.

Figure S2 in fact shows the observed seasonal cycle envelopes. The caption was misleading – we have corrected it now, and have also included a similar envelope figure for modelled data (Figure S3). We have now changed the text in the manuscript accordingly, "(see Figure S2 and S3 for observed and modelled seasonal cycle envelopes over the entire period)."

To my eye, there are evident, but small, seasonal cycle changes, with significant variability about consistent systematic changes. Thus, I would expect difficulty in quantifying the systematic changes, and this difficulty should be carefully considered before making firm conclusions. In this regard, Figure S2 indicates that seasonal changes appear to be clearer and more systematic in NE US compared to SW US. Thus, it may make sense to give a clear, statistically significant analysis of NE US first, and then address the SW US second.

We agree with the reviewer that NE US shows a distinct and clearer shift in the seasonal cycle over the two decades than other sub-regions. Therefore, we have moved the discussion of NE US before the NW US discussion in the text and Figures 9 and 10 have been flipped. We have also made other small tweaks to the text, for example, referring the reader to the envelope plots to fully appreciate the variability in these 5-year averaged changes. Additionally, we have now included regionwise ASCII files containing observed and modelled monthly mean MDA8 O3 seasonal cycles along with their tagged contributions for each year along with the manuscript.

- 11) Section 3.2.3 is well organized, but I think the discussion could be simplified and clarified, and in a few places corrected; in particular:
- Line 731: Inclusion of parameters of the Fourier analysis of the 5-year averaged periods

should be included in a table in the manuscript for the two example regions, as suggested in Major Issue 7).

We have now included a new table (Table 2) in the manuscript which shows the Fourier parameters for observed and modelled seasonal cycles for the entire, initial, and recent periods for the receptor regions discussed in the manuscript.

• Lines 741-742: Note that the shifts in  $\varphi_1$  of 0.04 and 0.10 radians correspond to shifts of only 2.3 and 5.8 days, respectively – quite small shifts. And as noted in Major Issue 8) the peak of the fundamental shifts in the opposite direction from the phase shifts.

Based on the clarified definition of the phases used in our analyses, these shifts mean 0.04 and 0.10 months, i.e., 1.2 and 3 days respectively which are indeed very small. We have noted this in the text at line L842. However, the sign of these peaks is in the same direction of the shifts in time as discussed earlier in our response to major comment 8.

• Lines 773-774: A  $\phi_1$  shift from 5.40 to 5.01 radians actually indicates a shift of the seasonal maximum by 23 days, but from spring towards summer. Those values correspond to peak value shifting from Julian Day 143 to 165 or May 23 to Jun 14. I do not see how this is consistent with Figure 10. An explanation is required (perhaps in the Supplement) so that the reader can fully follow the discussion. An example showing how the 1st and 2nd harmonics combine to approximate the seasonal cycle in the two 5-year periods in the NE US would be quite helpful to include in the Supplement. Given the issues identified above, I suggest that this Section be completely rethought, with the concluding paragraph revised as needed.

With the clarified definitions of the phases, a  $\varphi_1$  (now, p1) shift from 5.40 to 5.01 refers to a summer-to-spring backward shift from 12 June to 1 June. We have now discussed the Fourier Transform process in the supplement in Text S2.

12) I have not attempted to critically review Section 3.3 as carefully as I did Section 3.2. The discussion in these sections is similarly organized for both continents. Please seek to include any manuscript improvements made to the former sections in the latter sections where appropriate. And please similarly review all major and minor comments that refer to Section 3.2 when revising Section 3.3.

We have implemented many of the changes suggested in the comments to European analyses where relevant. For example, we have also fitted exponential functions to European receptor regions (see Figure S16 f-i; lines L947-952). We have also computed correlation coefficient r for observed vs modelled PSO for European receptor regions which are annotated in Figure 12 and mentioned in the text at lines L933-934. We have also included new text on the exponential fits to the local anthropogenic NOx contributions to PSO in European regions in order to facilitate the comparison of the

e-folding time ( $\tau$ ) with observationally-derived values in published literature (see Figure S16 and Table S10). We have also reported local contributions at a background site in Europe (Mace Head at the west coast of Ireland) in Table S11. We have also removed/clarified some of the speculative statements in section 3.3.2 which relate Fourier parameters to tagged contributions.

- 13) Section 3.4 raises an entirely new area of discussion that raises new questions in my mind, specifically:
- Its introduction is somewhat confusing. I suggest changing the phrase "in these regions" to "in the receptor regions", assuming this is correct.

**We have now changed the text accordingly.**

• The 2nd sentence in the 2nd paragraph is also confusing. "It is noteworthy that this NO2\_FOREIGN, locally recovered from foreign ozone titration, is separately tagged in our modelling system than the NO2 directly flowing from foreign regions (which we do not discuss here)." First, "... separately tagged in our modelling system than ..." is not clear to me. Second, it raises the question of what exactly is and what is not included in the tagging. NO2 directly flowing from foreign regions is generally considered to be small due to the short lifetime of NOx in the troposphere, but what about PAN and other organic nitrates? They have been considered reservoirs of sequestered NOx that can be transported over intercontinental distances in the free troposphere. However, it is not clear to me how the model treats ozone produced by foreign NOx transported as an organic nitrate to a receptor region, where it produces NOx after release from the reservoir species.

I suggest that the authors remove Section 3.4 from this paper, which is already quite long, and then more fully discuss the NOx-tagging system in the Introduction or Section 2.1 so that the reader is aware of issues such as tagging of NO2 directly flowing from foreign regions, and NOx reservoir species transported from foreign regions.

We have decided to retain this section because we believe it provides a crucial piece of knowledge on the reasons for the increase in wintertime ozone in NAM and EUR, especially the distinction between actual increase in the inflow of foreign Ox and the weakened titration due to a drop in local NOx. However, based on the reviewer's suggestion we have now further improved the text to make it clearer and have also added further details in section 2.1:

"The TOAST system differentiates NO2 into two distinct chemical families: NOy and Ox, with separate tracers for NO2 as members of each of these families. NO2 as a member of the NOy family tracks NOx which is directly emitted or produced in the atmosphere (e.g. by lightning), while NO2 as a member of the Ox family tracks NO2 which is formed chemically through reactions of NO with either ozone or peroxy radicals and subsequently

undergoes photolysis to ultimately form ozone. Further details are given in Butler et al. (2018)."

**Minor Issues:**

1) Figures S3 and S4 present scatterplots for the parameter values derived from the Fourier analyses. The derived r values annotated in the figures quantify how well the model reproduces the interannual variability in the parameter values in the respective regions. It would be useful to also give the r value for the entire 95 (North America) or 76 (Europe) set of values; this (generally significantly larger value) would quantify how well the model reproduces both the spatial variability and the interannual variability of the respective parameter throughout all regions on each continent. From inspection of the figures the model performance for some parameters appears to be quite impressive indeed. Note that the caption to Figure S4 should give 76 (not 95) as the number of markers.

Thanks for this suggestion. We have now included the correlation coefficient r values for the entire 95 and 76 data points for NAM and EUR respectively in the updated figures. We have also corrected the names of the phase parameters (p1 and p2) and their units (months).

2) In lines 403 and 413 the y0 parameter is described as representing annual average MDA8 O3 derived from detrended data. However, since the authors derive values for only a single year, no detrending has been performed, and the y0 parameters thus represent actual annual averages, and thus, still include the interannual variability. I suggest removing the references to "detrended" data.

Thanks for pointing this out. y0 was calculated for each year before detrending the full 19-year data that was used to perform the fourier decomposition. We have therefore removed these references to detrended data.

3) Unless the authors have a particular reason for including Tables S2-S9, I suggest they be removed, or at least shortened to only the summary values spanning the multi-year periods.

We have now included the fourier parameters for 19-year and 5-year averaged periods in a new table (Table 2) in the main manuscript. However, we have retained the tables S1-S9 in case readers are interested in a quantitative characterization of individual year seasonal cycles.

4) Line 396: I suggest that the correlation coefficients at the annual average timescale be explicitly stated (i.e., 0.34 to 0.95). Add a similar statement to the paragraph for Europe beginning on line 428 and for the Belarus & Ukraine region on line 451.

We have now included the r values for annual averaged timescale for all these regions in the text.

5) Line 430: Modify final phrase to "..., except SE Europe and RBU."

**We have modified the text accordingly.**

6) Lines 531-32: This statement should be more forcefully stated, something like "... the observed PSO levels consistently exceeded the WHO guideline (31 ppb) throughout the study period by at least 10(?) ppb". Similarly for European regions on line 827.

We have now modified this sentence to: "Crucially, across all North American regions, the observed PSO levels consistently exceeded the WHO long-term guideline (31 ppb) by at least 10 ppb throughout the study period."

7) Line 532 and elsewhere: When measured or modeled ozone concentrations are compared to the WHO guideline, it should be specified that it is the "WHO long-term guideline" that is being referenced.

We have modified all instances of "WHO guideline" to "WHO long-term guideline" in the text.

8) Line 535: Upon the first occurrence of the authors' Quantitative quote of a trend (e.g., (-0.19 (0.01) [-0.32, -0.06] ppb/yr), please define the 4 numbers given.

We have now added the following description after the first occurrence: "[here and henceforth the trends are reported in the following format (trend (p-value) [95% confidence lower limit, 95% confidence upper limit])]"

9) Table 2 should include the value of the derived trend (i.e., the most probable value of the trend) even if the 95% confidence intervals include zero. The table would be clarified if the column spacing were adjusted so that in each table entry the linear trend appears on the 1st line, p-values (shown in parentheses) on *the* 2nd line, and 95% confidence intervals on 3rd and 4th lines with all negative signs appearing on correct lines.

**The entries in Table 2 follow the format as described by the reviewer.**

10) Lines 553-57: This sentence requires clarification; it refers to "year-to-year variations in local emissions", which may be taken to indicate interannual variability. However, I certainly expect (and from the discussion the authors seem to agree) that the temporal correlation is largely driven by systematic decreases in local NOx emissions over the 19 year study period.

We have now included the point about the systematic decline in the sentence: "...year-to-year variations in local emissions (i.e., their systematic decline) significantly drive the variability (decline) in total PSO levels."

11) Line 568 states that: "... reductions in local NOx emissions translate directly and proportionally to reductions in the ozone ...". It is clear that the translation is direct, but there is no analysis to show that it is proportional (i.e., linearly related). Unless this proportionality can be demonstrated and the proportionality constant quantified, the phrase "and proportionally" should be removed.

Thanks for picking up this subtle point. We have removed the phrase "and proportionally" from this sentence.

12) Line 644-645 state "The phase  $\varphi_1$  indicates the timing of the annual peak, with numerically larger values typically corresponding to a later peak in the year ...." This is not correct; larger phase angle values always correspond to an earlier peak in the year. Please see discussion in Major Point 8).

After the clarified definition of the phase, this statement is valid. We have changed all instances of  $\varphi$ 1 and  $\varphi$ 2 to  $\varphi$ 1 and  $\varphi$ 2, respectively, to avoid any confusion.

13) Line 646 and elsewhere: "Tables S2-S6" should be "Tables S1-S5".

**Now corrected.**

14) Lines 737-738: Discussion could be made more accurate, viz. "The observed annual mean ozone (y0) decreased slightly from 41.4 ppb to 40.6 ppb, while the modeled y0 increased slightly from 43.5 ppb to 43.9 ppb), slightly increasing the positive bias noted earlier."

We have modified this sentence accordingly.

15) In Table 3, certainly only one decimal place in the entries is statistically justified.

We have updated the entries in Table 3 (now, Table 4) to only one decimal place.

16) Line 1057: I suggest strengthening – perhaps end the sentence with "... in both regions exceeds the long-term WHO guideline by wide margins over the entire study period.

We have modified this sentence accordingly.

**References:**

Mims, C.A., D.D. Parrish, R.G. Derwent, M. Astaneh and I.C. Faloon (2022), A conceptual model of northern midlatitude tropospheric ozone, Environ. Sci.: Atmos., 2, 1303–1313, DOI: 10.1039/d2ea00009a. Hosseinpour, F., Kumar, N., Tran, T., and Knipping, E.: Using machine learning to improve the estimate of U.S. background ozone, Atmospheric Environment. 316. 120145, https://doi.org/10.1016/j.atmosenv.2023.120145, 2024.

 $Prather\ Michael\ J\ 2007 Lifetimes\ and\ time\ scales\ in\ atmospheric\ chemistry Phil.\ Trans.\ R.$

Soc. A.3651705-1726

http://doi.org/10.1098/rsta.2007.2040